# A data-based predictive model for spatio-temporal variability in stream water quality

Danlu Guo[1], Anna Lintern[1,2], J. Angus Webb[1], Dongryeol Ryu[1], Ulrike Bende-Michl[3], Shuci Liu[1], Andrew William Western[1]

[1] Department of Infrastructure Engineering, The University of Melbourne, Parkville, VIC Australia;

[2] Department of Civil Engineering, Monash University, Clayton, VIC Australia

[3] Bureau of Meteorology, Parkes, ACT Australia.

Corresponding author's email: danlu.guo@unimelb.edu.au

## Abstract

Our current capacity to model stream water quality is limited particularly at large spatial scales across multiple catchments. To address this, we developed a Bayesian hierarchical statistical model to simulate the spatio-temporal variability in stream water quality across the state of Victoria, Australia. The model was developed using monthly water quality monitoring data over 21 years, across 102 catchments, which span over 130,000 km$^2$. The modelling focused on six key water quality constituents: total suspended solids (TSS), total phosphorus (TP), filterable reactive phosphorus (FRP), total Kjeldahl nitrogen (TKN), nitrate-nitrite (NO$_x$), and electrical conductivity (EC). The model structure was informed by knowledge of the key factors driving water quality variation, which had been identified in two preceding studies using the same dataset. Apart from FRP, which is hardly explainable (19.9%), the model explains 38.2% (NO$_x$) to 88.6% (EC) of total spatio-temporal variability in water quality. Across constituents, the model generally captures over half of the observed spatial variability; temporal variability remains largely unexplained across all catchments, while long-term trends are well captured. The model is best used to predict proportional changes in water quality in a Box-Cox transformed scale, but can have substantial bias if used to predict absolute values for high concentrations. This model can assist catchment management by (1) identifying hot-spots and hot moments for waterway pollution; (2) predicting effects of catchment changes on water quality e.g. urbanization or forestation; and (3) identifying and explaining major water quality trends and changes. Further model improvements should focus on: (1) alternative statistical model structures to improve fitting for truncated data, for constituents where a large amount of data below the detection-limit; and (2) better representation of non-conservative constituents (e.g. FRP) by accounting for important biogeochemical processes.

## Keywords

stream water quality; spatio-temporal variability; sediments; nutrients; statistical modeling; Bayesian hierarchical model

## 1. Introduction

Deteriorating water quality in aquatic systems such as rivers and streams can have significant environmental, economic and social ramifications (e.g. Whitworth et al., 2012;Vörösmarty et al., 2010;Qin et al., 2010;Kingsford et al., 2011). Reducing these impacts requires effective management and mitigation of poor water quality; however, high variability in water quality both across space and time reduces our ability to accurately assess the status of water quality and to develop effective management strategies. Thus, improved modelling frameworks to predict and interpret this variability would be useful for water quality management (Chang, 2008;Ai et al., 2015;Zhou et al., 2012).

Water quality conditions can vary across individual events, as well as at daily, seasonal and inter-annual scales at an individual location (Arheimer and Lidén, 2000; Kirchner et al., 2004; Larned et al., 2004; Pellerin et al., 2012; Saraceno et al., 2009). Water quality conditions also typically differ substantially across locations (Meybeck and Helmer, 1989;Chang, 2008;Varanka et al., 2015;Lintern et al., 2018a). These variabilities in stream water quality are driven by three key mechanisms: (1) source, which defines the total amount of constituents being available in a catchment; (2) mobilization, which detaches constituents (both in particulate and dissolved forms) from their sources via processes such as erosion and biogeochemical processing; and (3) delivery of mobilized constituents from catchments to receiving waters via multiple hydrologic pathways including surface and subsurface flow (Granger et al., 2010).

Spatial variability in stream water quality is driven by human activities within catchments (e.g., land use and management, vegetation cover etc.) (Lintern et al., 2018a;Carey and Migliaccio, 2009;Giri and Qiu, 2016;Heathwaite, 2010), along with natural catchment characteristics such as climate, geology, soil type, topography and hydrology (Hrachowitz et al., 2016;Poulsen et al., 2006;Sueker et al., 2001;Onderka et al., 2012). At the same time, temporal shifts in water quality are also influenced by changes in pollutant sources, such as land use and land management including urbanization, agriculture and vegetation clearing (Ren et al., 2003;Smith et al., 2013;Ouyang et al., 2010). In addition, water quality can also vary in time with variations in the mobilization and delivery processes, which are largely driven by the hydro-climatic conditions at a catchment, such as streamflow (Ahearn et al., 2004;Mellander et al., 2015;Sharpley et al., 2002;Zhang and Ball, 2017), the timing and magnitude of rainfall events (Fraser et al., 1999;Miller et al., 2014) and temperature (Bailey and Ahmadi, 2014).

As abovementioned, we have good understanding of the key controls for variations in water quality,
albeit in an isolated, idealized context. We still lack a sound understanding of how relationships between
specific landscape characteristics and water quality can shift with influences from other landscape
characteristics, and how the drivers of temporal variability in water quality can interact and vary across
large spatial scales (Musolff et al., 2015;Lintern et al., 2018a;Ali et al., 2017). In contrast, current
detailed understanding have been primarily based on field studies at small scales with detailed
information on specific temporal drivers ranging from hydrologic conditions to detailed management
decisions such as fertilizer rates and application timing (Smith et al., 2013;Poudel et al., 2013;Adams et
al., 2014). While operational weather observation networks, stream gauging networks and remote
sensing can provide some of this information, developing a large-scale understanding of water quality
patterns across catchments would ideally also involve an extensive suite of management information
that substantially exceeds what is currently available.
Due to the limited understanding of large-scale water quality patterns, we currently lack the capacity to
model spatio-temporal variabilities in water quality at large scales across multiple catchments. This
hinders our ability to inform the development of effective policy and mitigation strategies over large
regions. Specifically, conceptual or physically-based water quality models are typically limited by the
simplification of physical processes such as flow pathways (Hrachowitz et al., 2016). Furthermore,
practical implementation of these models can be also limited by the intensive data requirements for
calibration and validation, particularly for large regions with highly heterogeneous catchment conditions
(Fu et al., 2018;Abbaspour et al., 2015). In contrast, when performed over large geographical regions,
statistical water quality models are generally more capable of simulating water quality variability while
requiring less detailed information and thus effort for implementation. However, existing statistical
models often focus only on either the spatial variation of time-averaged water quality conditions
(Tramblay et al., 2010;Ai et al., 2015) or the temporal variation at individual locations (Kisi and Parmar,
2016;Kurunç et al., 2005;Parmar and Bhardwaj, 2015), which often limits their value as practical
management tools. Modelling the spatio-temporal variability simultaneously remains challenging over
long time periods and large regions.
Accordingly, this research attempts to bridge the gap between fully-distributed physically-based water

quality models and data-driven statistical approaches. We aim to develop a process-informed, data-driven model to predict spatio-temporal changes in stream water quality over a large region consisting of multiple catchments. Specifically, this model was established using long-term (21 years) stream water quality observations across 102 catchments in Australia, with an aggregate catchment area of 130,000 $km^2$. To obtain the necessary understanding of process drivers required to develop this model, two preceding studies were conducted on the same dataset to identify the key drivers for the spatial and temporal variability of water quality, respectively (Lintern et al., 2018b; Guo et al., 2019). The aim of this study is to develop an integrated spatio-temporal model using the previously-identified spatial and temporal predictors, and to then assess the performance of this model. Spatio-temporal variability of water quality was modelled using a novel Bayesian hierarchical approach which can jointly account for both variability components, including accounting for varying temporal water quality dynamics between catchments. This modelling approach also has relatively low requirement for input data, which keeps the modelling detail commensurate with the level of data availability. During the model development, we also obtained additional understanding on the patterns of spatial variations in the effects of each temporal predictor. The model can potentially provide useful information for large-scale catchment management, assessment and policy making, such as testing major changes in land use patterns, informing pollution hot-spots, as well as identification and attribution of water quality trends and changes over time.

## 2.   Method

We first discuss the process used to develop the integrated spatio-temporal model (Section 2.1). Sections 2.1.1 and 2.1.2 introduces the statistical modelling framework and the data used for model development, respectively. The approaches to determine model structure was then introduced, which include the choice of key predictors (Section 2.1.3) and the calibration for model parameters (Section 2.1.4). Finally, the approaches to evaluate model performance and robustness are described in Section 2.2.

**2.1 Model development**

2.1.1 Spatio-temporal modelling framework

A Bayesian hierarchical approach was used to model the spatio-temporal variability in stream water quality. The Bayesian approach enables the inherent natural stochasticity of water quality to be

incorporated into the model (Clark, 2005). A key strength of applying the hierarchical model structure
to analyze spatio-temporal variability is that this structure enables the key controls of temporal
variability in water quality to vary across locations (Webb and King, 2009;Borsuk et al., 2001). This
variability has been found to be important in other study regions where the (temporal) solute export
regime varies with catchment characteristics such as climate and land use (Musolff et al., 2015;Poor and
McDonnell, 2007).
The structure of the Bayesian hierarchical model is presented below in Eq. 1 to 6. Eq. 1 formulates the
transformed constituent concentration (see Section 2.1.2 for justification) at time $i$ and site $j$ ($C_{ij}$) as a
normally distribution with a mean $\mu_{ij}$ and standard deviation $\sigma$ representing inherent randomness.

$$C_{ij} \sim N(\mu_{ij}, \sigma) \tag{1}$$

To represent spatio-temporal variability, $\mu_{ij}$ is modelled as the sum of the site-level mean constituent
concentration ($\bar{C}_j$) and the deviation from that mean at time $i$ ($\Delta_{ij}$) (Eq. 2).

$$\mu_{ij} = \bar{C}_j + \Delta_{ij} \tag{2}$$

To describe spatial variability, the site-level mean concentration at site $j$ ($\bar{C}_j$) is modelled as a linear
function of a global intercept ($intC$), and the sum of $m$ catchment characteristics $S_{1,j}$ to $S_{m,j}$ (e.g. land
use, topography) weighted by their relative contributions to spatial variability ($\beta S_1$ to $\beta S_m$) (Eq. 3).

$$\bar{C}_j = intC + \beta S_1 \times S_{1,j} + \beta S_2 \times S_{2,j} + \cdots + \beta S_m \times S_{m,j} \tag{3}$$

The temporal variability, represented by the deviation from the mean ($\Delta_{ij}$), is a linear combination of $n$
temporal variables, $T_{1,ij}$ to $T_{n,ij}$ (e.g., climate condition, streamflow, vegetation cover) (Eq. 4), at time
$i$ and site $j$.

$$\Delta_{ij} = \beta T_{1,j} \times T_{1,ij} + \cdots + \beta T_{n,j} \times T_{n,ij} \tag{4}$$

The selection of key spatial and temporal predictors for the model has been performed in our two
preceding studies (Lintern et al., 2018b; Guo et al., 2019) and is briefly described in Section 2.1.3. Eq.
1 to 4 enable the model to separately represent the spatial and temporal variability in water quality;
however, there is still a further step required to make the model fully spatio-temporal (i.e. being able to
predict over both time and location). Specifically, in Guo et al. (2019), clear spatial variation was
observed in the relationships between water quality and its key temporal predictors (i.e. in the $\beta T_{N,j}$ in
Eq. 4). To be able to model multiple catchments across a large spatial area simultaneously, we must
account for differences in these temporal influences across sites. To do this, the effect of each temporal
variable at site $j$ ($\beta T_{N,j}$ with $N$ in 1,2, … $n$) is drawn from a distribution with a mean of $\mu\beta T_{N,j}$ (Eq. 5),
which is then modelled with a linear combination of two additional chatchment characteristics, $ST_{N1,j}$
and $ST_{N2,j}$ (Eq. 6). Details of the selection for these two additional predictors are presented in Section

147 2.1.3.

$$\beta T_{N,j} \sim N(\mu\beta T_{N,j}, \sigma\beta T), for\ N\ in\ 1, 2, … n \qquad (5)$$

$$\mu\beta T_{N,j} = int\beta T_N + \beta ST_{N1} \times ST_{N1,j} + \beta ST_{N2} \times ST_{N2,j} \qquad (6)$$

2.1.2 Data collection and processing
The Bayesian hierarchical model was developed with 21 years of monthly stream water quality
observations at 102 catchments in the state of Victoria, Australia (aggregate catchment area > 130,000
km$^2$). The collection and processing of the data are detailed in previous publications that worked with
the same dataset (Lintern et al., 2018b; Guo et al., 2019). Briefly, stream water quality data were
extracted from the Victorian Water Measurement Information System (Department of Environment
Land Water and Planning (DELWP) Victoria, 2016b), which contains monthly grab samples of water
quality at approximately 400 sites across Victoria. Water quality data sampled between 1994 and 2014
at 102 sites were used to develop the model (Fig. 1). These sites and time period were chosen because
they provided the longest consistent period of continuous records over the greatest number of monitoring
sites. The catchments corresponding to these water quality monitoring sites were delineated using the
Geofabric tool (Bureau of Meteorology, 2012), and have areas ranging from 5 km$^2$ to 16,000 km$^2$. The
water quality parameters of interest were: total suspended solids (TSS), total phosphorus (TP), filterable
reactive phosphorus (FRP), total Kjeldahl nitrogen (TKN), nitrate-nitrite (NO$_x$) and electrical
conductivity (EC). These parameters represent sediments, nutrients and salts, which are some of the key
concerns for water quality managers in Australia and around the world. These water quality samples
were collected following standard DELWP protocols (Australian Water Technologies, 1999) and
analysed in National Association of Testing Authorities accredited laboratories. Note that in the
sampling protocol, FRP is defined as '*Reactive Phosphorus for a filtered sample to a defined filter size*
*(e.g. RP(<0.45 μm))*', which is equivalent to the more widely-used terminology, SRP i.e. Soluble
Reactive Phosphorus (Jarvie et al., 2002).

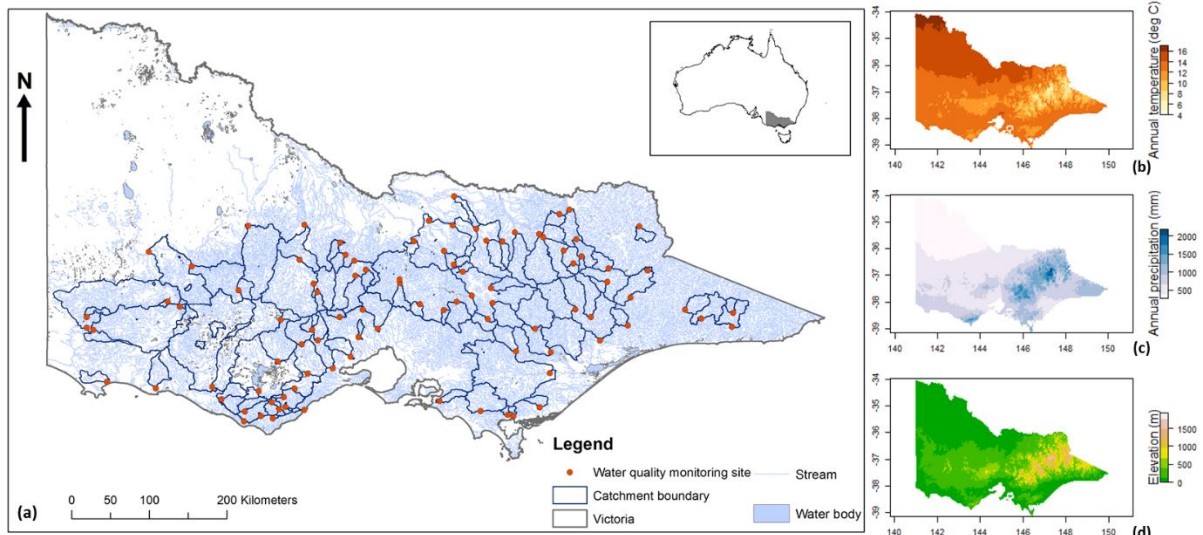


**Figure 1. Map of (a) the 102 selected water quality monitoring sites and their catchment boundaries, with inserts showing the location of the state of Victoria within Australia; (b) annual average temperature and (c) annual precipitation and (d) elevation across Victoria.**

To compile a dataset for the potential spatial explanatory variables (i.e. predictors to explain spatial variability in water quality), a comprehensive literature review was conducted (Lintern et al., 2018a), which summarized the key catchment landscape characterisitics that are widely known to influence water quality. Further, as part of Lintern et al. (2018b), fifty potential explanatory catchment characteristics were selected, which included catchment land use, land cover, topographic, climatic, geological, lithological and hydrological catchment characteristics. These variables were derived using datasets obtained from Geoscience Australia (2004, 2011), the Bureau of Meteorology (2012), the Bureau of Rural Sciences (2010), Department of Environment Land Water and Planning Victoria (2016) and the Terrestrial Ecosystem Research Network (2016) (see Table S1 in the Supplementary Material for detailed variable names and data sources). We used a static set of land use data from 2005-2006 to represent the entire study period, as a preliminary analysis between 1996 and 2011 suggested less than 1% changes in the key land uses in these catchments (i.e. agricultural, grazing, conservation).

Nineteen potential temporal explanatory variables were included. Firstly, data of discharge (originally in ML d$^{-1}$) and water temperature (°C) corresponding to the same timestamps for water quality observations were also extracted for each monitoring site over the study period (Department of Environment Land Water and Planning Victoria, 2016). Discharge was converted to runoff depth (mm d$^{-1}$) for each catchment, and the average streamflows over 1, 3, 7, 14 and 30 days preceding the water quality sampling dates were calculated. In addition, we extracted gridded dataset from the Australian

Water Availability Project (AWAP) (Frost et al., 2016;Raupach et al., 2009, 2012) and Australian Water
Resources Assessment Landscape (AWRA-L) model (Frost et al., 2016). These datasets were used to
calculate catchment averaged values of daily average temperature (°C), daily rainfall (mm), antecedent
rainfall (1, 3, 7, 14 and 30 days preceding sampling), dry spell (> 0.1mm rainfall) length in the antecedent
14 days, daily actual evapotranspiration (ET) (mm), as well as soil moisture for the root-zone and the
deep-zone (averaged volumetric content for shallower and deeper than 1m, respectively). In addition,
catchment averaged monthly NDVI data were extracted from Advanced Very High Resolution
Radiometer (AVHRR) Product (Eidenshink, 1992) and Moderate Resolution Imaging
Spectroradiometer MOD13A3 (NASA LP DAAC, 2017). A summary of these datasets of temporal
variables and their corresponding sources are in Table S2 in the Supplementary Material and details are
provided in Guo et al. 2019.
The raw input data were filtered and transformed to increase the data reliability, continuity and
symmetry, making them more suitable for use in the linear spatio-temporal model structure (Eq. 3, 4
and 6). For the filtering process, we first removed all water quality records with flags indicating quality
issues. We also removed any values below the detection limit (DL), which was defined as the '*minimum*
*concentration detected for which there is 95% confidence of accuracy and therefore is accurate enough*
*to report*' in the monitoring protocols for this dataset (Australian Water Technologies, 1999). This was
because the uncertainty in values below the DL would be amplified after transformation, which would
influence the subsequent model fitting. Furthermore, those undetectable low concentrations were of less
interest for management purposes. Water quality records corresponding to days with zero flows were
also excluded from further analyses.
The transformation process was performed for each of the spatial catchment characteristics, temporal
explanatory variables, as well as each water quality constituent to improve the symmetry of individual
distributions. The log-sinh transformation (Wang et al., 2012) (Eq. 7) was used for all catchment
characteristics, due to its ability to resolve the presence of zero values in several of the catchment
characteristics (e.g., percentage area of individual land uses). The *GA* package in R (Luca Scrucca, 2019)
was used to identify the log-sinh transformation parameters (*a* and *b*) for each spatial explanatory
variable that minimized the data skewness (i.e. symmetry is maximized) across all 102 catchments.

$$y_{log-sinh} = \frac{1}{b}\log(\sinh[a + by_{raw}])$$ (7)

In addition, all observed constituent concentrations and temporal explanatory variables were Box-Cox transformed (Box and Cox, 1964) (Eq. 8).

$$y_{Box-Cox} = \begin{cases} \frac{y_{Raw}^{\lambda}-1}{\lambda}, & for \; \lambda \neq 0 \\ log y, & for \; \lambda = 0 \end{cases}$$ (8)

For each variable, the optimal Box-Cox transformation parameter $\lambda$ was identified using the *car* R package and a maximum likelihood-like approach. We first identified the optimal Box-Cox parameter $\lambda$ using the data at each site (i.e. 21-year time-series). The averaged $\lambda$ across all sites was then used to transform the data across all catchments together. This transformation approach ensured that all sites used a consistent transformation parameter. All transformation parameters used are summarized in Tables S3 and S4 in the Supplementary Material. The transformation process has greatly improved the data symmetry and thus suitability for use in a linear model (the quality of the transformations was assessed via visual inspection in Lintern et al., 2018b; Guo et al., 2019; and summarized in Figures S2, S4 and S6 in the Supplementary Material).

2.1.3 Selection of key model predictors

Key predictors for the model were selected in a process-informed and data-driven manner based on our two preceding studies (Lintern et al., 2018b; Guo et al., 2019). Lintern et al. (2018b) identified the best spatial predictors ($S_1$ to $S_m$ in Eq. 3) for the model, while the best temporal predictors across all sites ($T_1$ to $T_n$ in Eq. 4) have been identified in Guo et al., (2019). In both studies, the best predictors were selected using an exhaustive search approach (May et al., 2011;Saft et al., 2016), which considered all possible combinations of the potential predictors introduced earlier in this section. This selection approach required firstly fitting an individual model to all possible candidate predictor sets, and then comparing all fitted models to select a single best set of predictors. Alternative models were evaluated based on the Akaike Information Criterion (AIC) (Akaike, 1974) and Bayesian Information Criterion (BIC) (Schwarz, 1978) to ensure optimal balance between model performance and complexity.

The best predictors to explain the spatial and temporal variabilities in each constituent are listed in Table 1. Generally speaking, the key factors controlling the spatial variability in river water quality were land-use and long-term climate conditions (Lintern et al., 2018b). Temporal variability was mainly explained

by temporal changes in streamflow conditions, water temperature and soil moisture (Guo et al., 2019).
The potential mechanisms via which these key drivers influence water quality are discussed in details
in these two previous studies.
**Table 1. Key factors affecting the spatial and temporal variability for each of six constituents, as identified**
**in Lintern et al. (2018) and Guo et al. (2019b), respectively.**

Whilst the previous studies (Lintern et al. 2018b, Guo et al. 2019) identified the predictors for spatial
and temporal variability respectively, they did not provide guidance on the predictors for spatial
variability in the relationships between drivers of temporal variability and temporal water quality
response (i.e. $\beta T$ in Eq 4). As such, the final step of the predictor selection process to develop the
combined spatio-temporal model was to identify the key catchment characteristics that affect spatial
variability in the hydroclimatic parameters driving temporal changers in water quality ($\beta T_1$ to $\beta T_n$ in Eq.
4, also right column in Table 1). This is achieved by selecting two spatial characteristics that are most
closely related to the coefficient for each temporal predictor ($ST_{N1}$ and $ST_{N2}$, Eq. 6) across all sites,
where only two spatial characteristics were used to avoid over-fitting. Selection of these two spatial
characteristics were based on a Spearman correlation analysis between the fitted parameter values of
each temporal predictor variable and the fifty potential spatial explanatory variables (as mentioned
earlier in this section), following three steps:
1.   from the 50 candidate spatial predictors, the one with the highest Spearman correlation with $\beta T_N$ is

selected as $ST_{N1}$, provided the correlation is statistically significant ($p<0.05$);

2.   the subset of remaining spatial predictors with spearman correlation with $ST_{N1} < 0.7$ is found; and
3.   from this subset, the spatial predictor with the highest spearman correlation with $\beta T_N$ is selected as

$ST_{N2}$, provided the correlation has $p<0.05$;

Steps 2 and 3 intended to avoid cross-correlations between $ST_{N1}$ and $ST_{N2}$. The selected spatial
characteristics that influence the temporal relationships in our model are presented and interpreted in
Section 3.1. Note that the entire process to select $ST_{N1}$ and $ST_{N2}$ was performed with the fitted
parameters for each predictor of the temporal variability obtained from Guo et al. (2019).
2.1.4 Model calibration
After identifying the spatial and temporal predictors for each constituent, as well as the spatial
characteristics which affect the strengths of each temporal predictor, the Bayesian hierarchical spatio-
temporal model was fitted for each constituent across all monitoring sites simultaneously. To achieve
this, we used the R package *rstan* (Stan Development Team, 2018), which enabled both the sampling of
parameter values from posterior distributions with Markov chain Monte Carlo (MCMC) and model
evaluation. Constituent standard deviation ($\sigma$) was assumed to be drawn from a minimally informative
prior half-normal of $N(0,10)$ distribution truncated to only positive values (Gelman, 2006; Stan
Development Team, 2018). The regression coefficient of each spatial predictor ($\beta S_1, \beta S_2, ..., \beta S_m$ in Eq.
3) was independently drawn from hyper-parameter distributions of $N(\mu\beta S_M, \sigma\beta S_M)$. The site-level
regression coefficients of the temporal predictors ($\beta T_{1,j}, \beta T_{2,j}, ..., \beta T_{n,j}$ in Eq. 4, respectively) were
sampled from the corresponding hyper-parameter distribution of $N(\mu\beta T_N, \sigma\beta T_N)$. The hyper-parameters
were further assumed to be drawn from minimally informative prior distributions, following
recommendations in Gelman (2006) and Stan Development Team (2019): for all the hyper-parameter
means, a normal prior distribution of $N(0,5)$ was used; for all the hyper-parameter standard deviations,
a half-normal prior distribution of $N(0,10)$ was used, which was truncated to only positive values. In
each model run there were four independent Markov chains. A total of 20,000 iterations were used for
each chain. Convergence of the chains was ensured by checking the *Rhat* value (Sturtz et al., 2005),
which is a summary statistic on the convergence of the Bayesian models from the four Markov chains
used in model calibration (Stan Development Team, 2018). Specifically, an *Rhat* value much greater
than 1 indicates that the independent Markov chains have not been mixed well, and a value of below 1.1
is recommended (Stan Development Team, 2018).
**2.2 Model performance evaluation and sensitivity analyses**
Performance evaluation of the model was undertaken on several aspects of the model results (Section.
3.2). Since the model was calibrated in a Box-Cox transformation scale (see justification in Section
2.1.2), the Box-Cox transformation scale was used for model evaluation to enable a clear investigation
on the influences of a wide range of factors that can influence model performance. Detailed performance
evaluations include:
1. *Ability to capture total spatio-temporal variability.* Firstly, the simulations from the fitted model

and the corresponding observed concentrations were compared at 102 sites altogether to

understand how the overall spatio-temporal variabilities were captured. For each constituent, this evaluation was performed with: 1) these above-DL data to focus only on data used for calibration (as detailed in Section. 2.1.2); and 2) the full dataset including the below-DL data (set to half of the DL of the specific constituent), to understand how well the model represents the full distribution of constituent concentrations. A good model performance when including the below-DL data would suggest that the calibrated model is transferable to below-DL data too. All performance assessments were based on both visual inspection of model fitting as well as the Nash-Sutcliffe efficiency (NSE), which quantified the proportion of variability that was explained by the model (Nash and Sutcliffe, 1970).

2. *Proportions of spatial and temporal variability explained.* This involved a decomposition of the total observed variability using Eq. 2., into proportions contributed by spatial variability (variations in all site-mean concentrations from the grand average of site-mean concentrations) and temporal variability (variations in all concentrations from the corresponding site-mean concentrations). The corresponding modelled values were then used to calculate NSE for each variability component of each constituent.

3. *Ability to capture variation in ambient conditions across space, and temporal variation (including trends) across multiple catchments.* These were evaluated by a) comparing all simulated and observed site-averaged long-term mean concentrations; and b) comparing the simulated and observed time-series and long-term trends at representative sites. Further to a), performance was also evaluated on a real measurement scale by first back-transforming all modelled sample concentrations, calculating the back-transformed site-level means and then compared those to the corresponding observations. A further analysis to b) was also performed by comparing the estimated Sen's slope (Akritas et al., 1995) for the observations and simulations at all sites, and then computing the percentage of sites where the observed trends as indicated by the Sen's slope have been correctly represented by the model.

Additional evaluations of model sensitivity were conducted with calibration and validation on subsets of the full data (Section. 3.3), to understand model transferability and stability:

1. *Model sensitivity to the monitoring sites used for calibration.* We randomly selected 80% of the

sites for calibration and used the remaining 20% for validation, and repeated this validation
process 50 times. We compared all calibration and validation performances of these 'partial
models' with each other, as well as with the performance of the full model, to obtain a
comprehensive evaluation of the sensitivity of model performance to calibration sites.
2.  *Model sensitivity to calibration data period.* Since the study region was greatly influenced by a
prolonged drought from 1997 to 2009 – known as the Millennium Drought (van Dijk et al.,
2013), we also investigated model robustness for before, during and after this drought period.
Specifically, we calibrated the model to each pre-, during- and post-drought period (1994-1996,
1997-2009 and 2010-2014, respectively) with model validation on the remaining data. For
example, when calibrating to the pre-drought period (1997-2009), validation was performed on
the merged during and post-drought period (1994-1996 plus 2010-2014). The corresponding
calibration and validation performances were compared with each other as well as against that
of the full model, to identify potential impacts of the drought on model robustness.

## 3.  Results

### 3.1 Spatial variation in the impact of temporal factors

The key controls of the spatial and temporal variations in water quality have been identified in our two
preceding studies (Lintern et al. 2018b, Guo et al. 2019) and briefly summarized in Section 2.1.3. and
are thus not discussed here. As also detailed in Section 2.1.3, to achieve full spatio-temporal predictive
capacity, the model developed in this study considers the spatial variation in the strength of each
temporal predictor by using two additional catchment spatial characteristics ($ST_{N1,j}$ and $ST_{N2,j}$ in Eq.
6). on the Spearman's correlations. Here we focus on the most important temporal predictor for each
constituent, streamflow, where Table 2 shows the two spatial characteristics identified that are most
closely related to the spatial variation of the effects of impact of streamflow on water quality. The full
list of the selected key catchment characteristics for all temporal predictors of each constituent is
summarized in Table S5 and visualized in Figure S4.

**Table 2. The key catchment landscape characteristics that are related to the varying relationships of water quality and same-day streamflow across space, which were selected as the two predictors for the streamflow effect in our model. The corresponding Spearman's correlation ($\rho$ at p<0.05) between the effect of streamflow and each catchment characteristic is presented.**

TSS, TP and TKN show consistent patterns of the spatial variation in the effects of streamflow on water

quality, which are strongly driven by the differences in average rainfall conditions across catchments. Specifically, streamflow generally has a larger effect on water quality in catchments with higher average annual rainfall. Since the streamflow effects are positive for the majority of catchments (as shown in Figure S5), these correlations indicate that for the same increase in transformed streamflow, a greater increase in transformed concentrations of TSS, TP and TKN will occur at a catchment with higher annual average rainfall. Given that the Box-Cox lambda values (Table S4) are close to zero, the transformation is log-like and hence changes in transformed flow and concentration approximately correspond to proportional changes in the real values of flow and concentration. In contrast, for FRP, $NO_x$ and EC, the spatial patterns of streamflow effects are specific to each constituent. This difference in the model results between TSS, TP and TKN against the other constituents might be related to the distinct transport pathways of particulate and dissolved constituents. The former is predominantly related to surface flow and thus relies heavily on rainfall contribution. Dissolved constituents are likely transported along the subsurface pathway. Apart from streamflow, the spatial patterns in other key temporal drivers of water quality (e.g. antecedent streamflow, soil moisture etc.) are less consistent across different constituents (Figure S4).

**3.2 Model performance evaluation**

The spatio-temporal water quality models show varying performances between the constituents. When assessed with only the above-DL data (Fig. 2), the best performing models are those for EC and TKN, which capture 90.7% and 65.8% of the total observed spatio-temporal variability. The modelling performance is lowest for FRP, $NO_x$ and TSS, with NSE values of -1.92, 0.216 and 0.225, respectively. When evaluated against the entire dataset (i.e., including both below- and above DL data), the models explain 19.9% (FRP) to 88.6% (EC) of spatio-temporal variability (Table 3). Model performances for FRP, $NO_x$ and TSS improve notably compared with the previous evaluation of above-DL data, however, they remain as the three constituents that are most difficult to predict. We further discuss the possible factors influencing their model performance in Section 4.1.

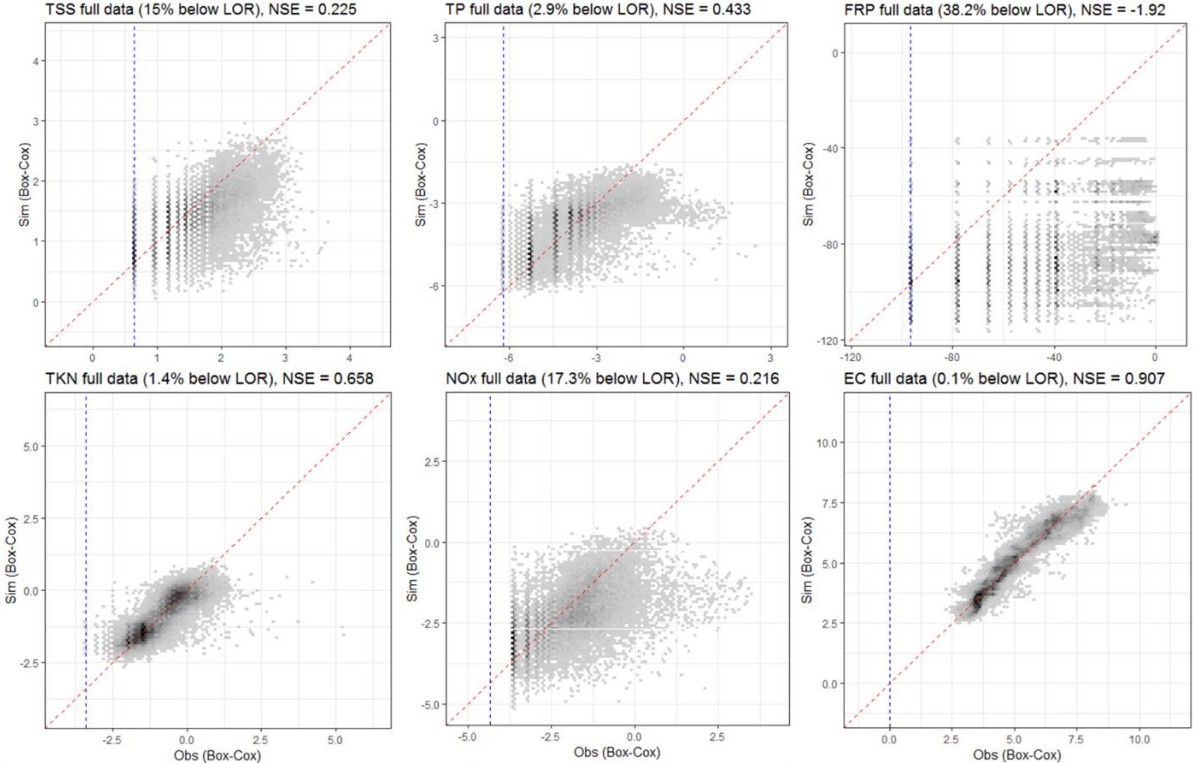

**Figure 2. Performance of the spatio-temporal models for each of the six constituents, represented by the simulated median concentrations and corresponding observations of above-DL records across all 102 calibration sites, in Box-Cox transformed space. Darker regions represent denser distribution of simulation and observation points. Dashed red lines show the 1:1 lines whereas dashed blue lines show the DL levels. For each constituent, the percentage of data below the DL and the model performance (NSE) are also specified.**

**Table 3. Comparison of model performance for all records and only the above-DL records for each constituent.**

The model performance to predict spatial and temporal variability is summarized in Figure 3, which compares the observed and explainable variability for each of the spatial and temporal components (detailed in Section 2.1.4). Regarding the observed variability (lighter colours), EC is strongly dominated by spatial variability (91.8%), highlighting that within-site variation in water quality is minimal compared to between-site variation. To a lesser extent, spatial variability also contributes to major proportions of total variability for TP and TKN (60.8% and 66.6%, respectively). TSS, FRP and NO$_x$ are more influenced by temporal variability (57.4%, 56.6%, 60.5%, respectively).

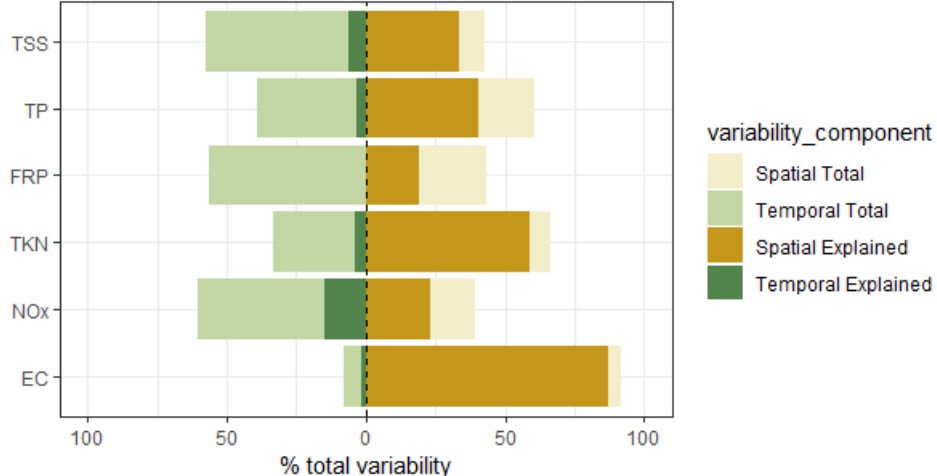

**Figure 3. Observed spatial and temporal variabilities as proportions of the total variability (total width of each bar, 100%). The dashed line differentiates temporal variability (left side) with spatial variability (right side), and the darker colours highlight the proportions of spatial and temporal variabilities that are explainable by the model. All values were estimated in Box-Cox transformed space.**

The explained variability (darker colours) show that, across all catchments, temporal variability is much more difficult to model compared with spatial variability. It also appears that a substantial part of the model's overall performance is driven by its ability to capture spatial variability in ambient water quality conditions. For example, the models for TSS, FRP and NO$_x$ show poorer overall performance (Fig. 2, with NSE values of 0.225, -1.92 and 0.216, respectively)), because the total variability for each of these is dominated by temporal variability (57.4%, 56.6%, 60.5%, respectively), which largely remains unexplained by the model (Fig. 3). In contrast, the EC model shows a very good fit with 90.7% of total variability explained – 91.8% of the total observed variability is due to spatial variability, of which 94.7% is explained by the model. Therefore, although the EC model can only explain a small portion of temporal variability (20% out of 8.2% of total variability), the overall model performance remains high.

As highlighted in Fig. 3, the model has good capacity to capture spatial variability in water quality. This is further evaluated in Fig. 4 by comparing the simulated and observed site-level mean concentrations. The highest model performance is for EC and lowest performance is for FRP (explaining 94.7% and 44.2% spatial variability, respectively). At the back-transformed scale, the model shows greater biases for sites with higher concentrations (approximately the highest 10% sites for each constituent) (Fig. 5). This is not surprising as the model was fitted to a Box-Cox transformed space that reduces focus on high values and increases the focused on low values. This compromised its ability to represent sites with

unusually high concentrations. The implications of the model having higher predictive capacity in the

transformed scale is further discussed in Section. 4.1.

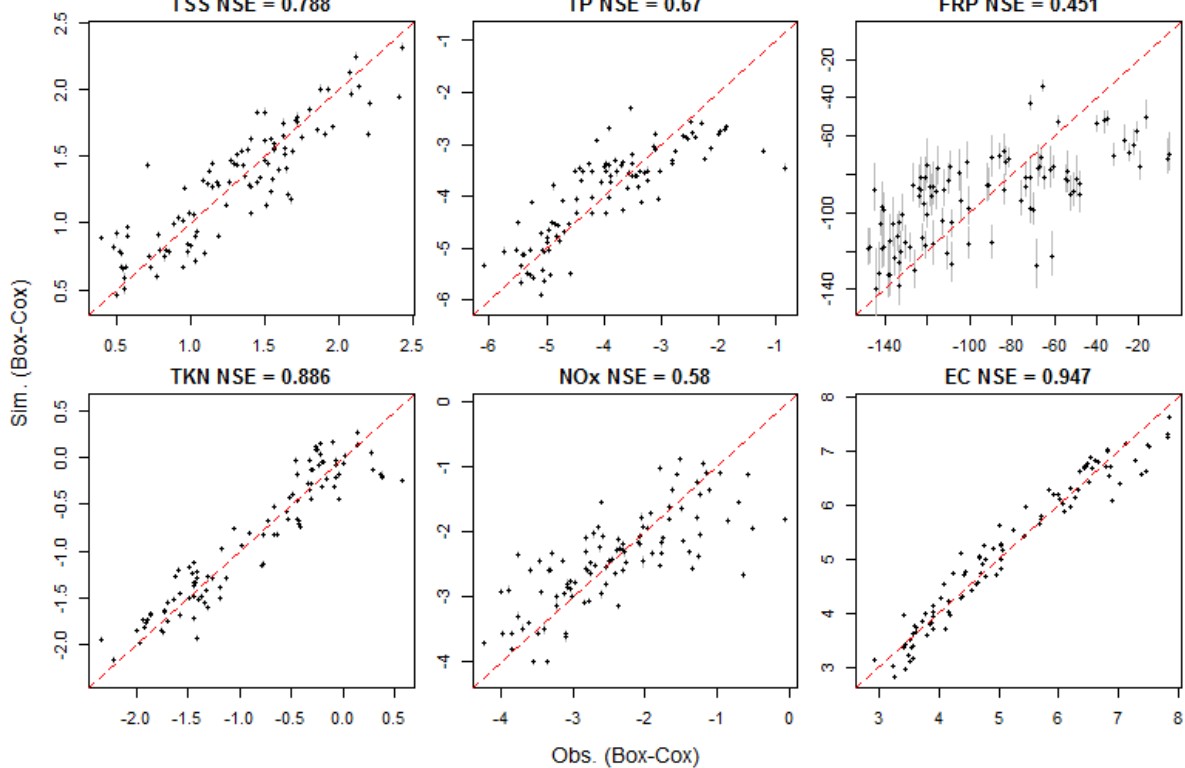

**Figure 4. Model fit for site-level mean concentration at the 102 calibration sites for six constituents, with the 95% lower and upper bounds of posterior simulations shown in vertical grey lines. All simulations and observations are presented in in Box-Cox transformed space. The NSE for each constituent is also shown and red dash lines show the 1:1 lines.**

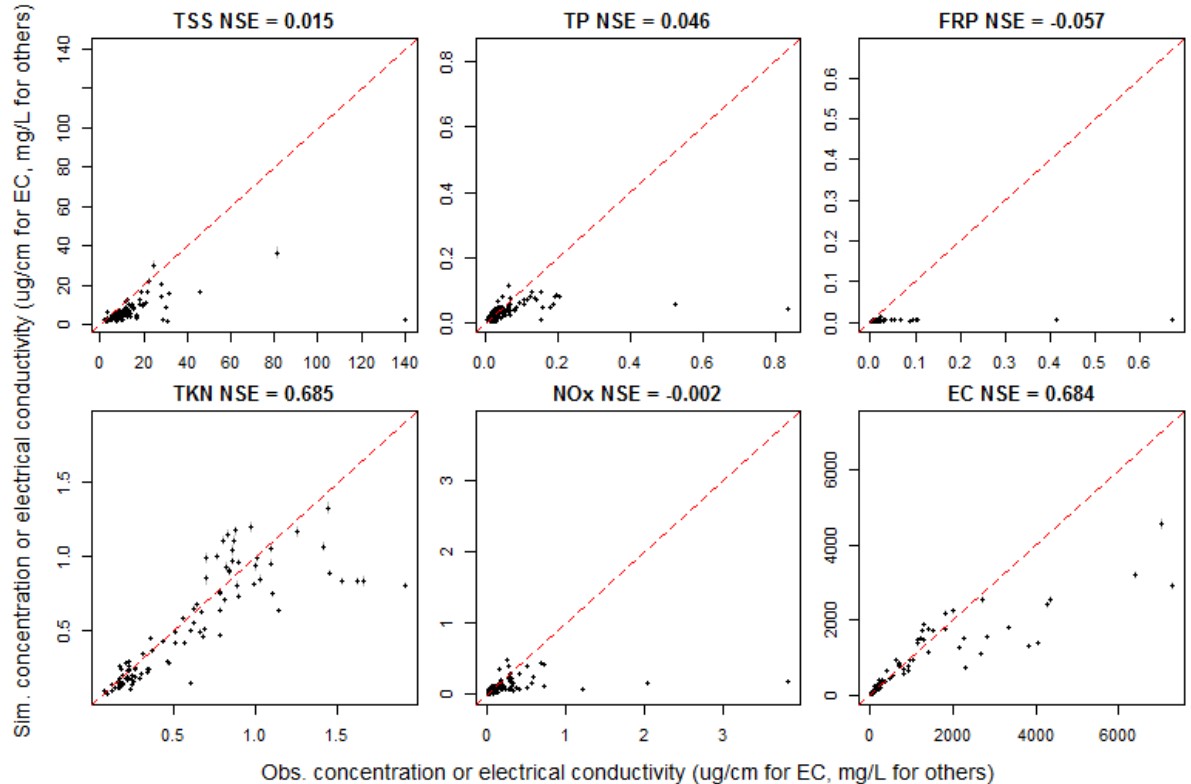

**Figure 5. Back-transformation of the model simulations to the measurement scale emphasizes lack of fit for the highest concentrations, illustrated by simulated against observed site-level mean concentrations of each constituent in a back-transformed scale. The 95% lower and upper bounds of all posterior simulations shown in vertical grey lines. The NSE for each constituent is also shown and red dash lines show the 1:1 lines.**

As also noted in Fig. 3, the ability of the spatio-temporal model to explain temporal variability remains

relatively limited. This is further explored in Fig. 6, where the observed and simulated time-series are

presented for one monitoring site for each constituent, at which the model performance (NSE) was the

highest. These results show that even for catchments where the model has the highest ability to capture

temporal variability, the model consistently underestimated temporal variability for all constituents.

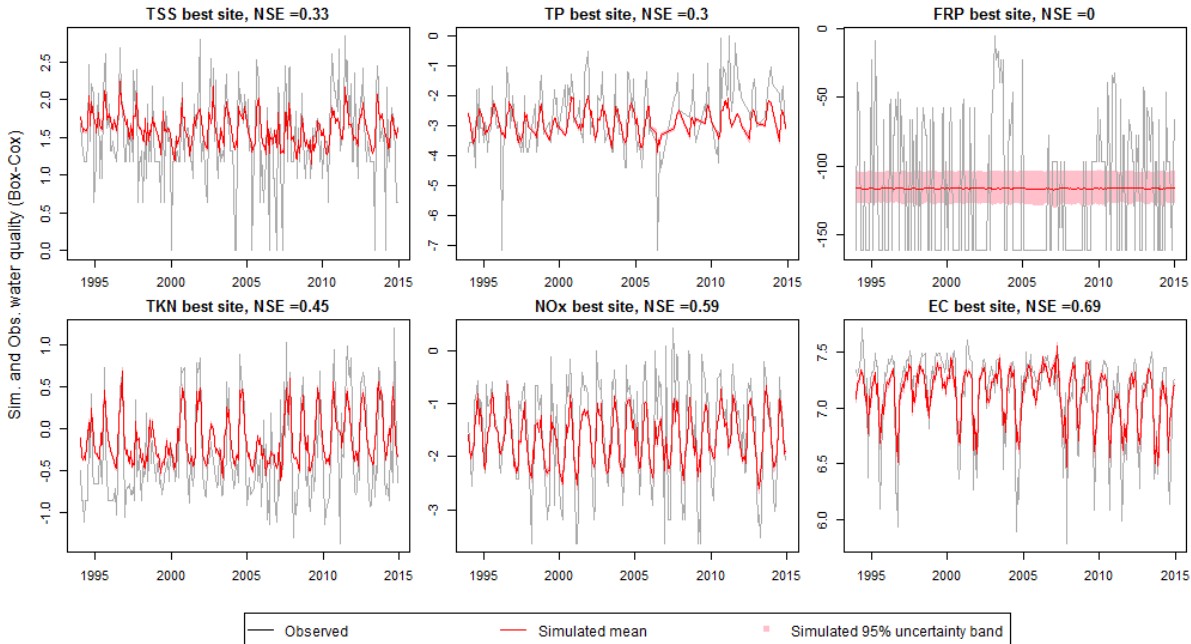

Figure 6. Model fit of the within-site (temporal) water quality variability, illustrated with the observed and simulated time-series for the best-performing site for each constituent. All values are presented in Box-Cox transformed space. The NSE for each constituent is also shown. The red line indicates the corresponding mean of all posterior simulations, while the pink bands show the corresponding 95% lower and upper bounds (only visible for FRP).

Fig. 6 also illustrates that, although the model shows substantial underestimation of temporal variability within site, long-term temporal trends in the time-series are well captured at the best sites (except for FRP). Table 4 summarizes the ability of the model to capture observed trends across all 102 catchments for each constituent. In general, the model is able to capture observed trends in most sites for $NO_x$ and EC and for both positve and negative trends. For TP and TKN, positive trends are well captured while for TSS the negative trends are better captured.

Table 4. Model ability to capture observed water quality trends across all monitoring sites for each constituent. The percentages of sites where observed positive and negative trends are captured by the model are presented separately. Values in brackets indicate numbers of sites where corresponding positive or negative trends are observed. For detailed estimation of these percentages please refer to Sect. 2.2.

### 3.3 Model sensitivity analyses

We first compare the performance of each spatio-temporal model fitted with the full dataset with those obtained from the 50 corresponding "partial" models that were calibrated to only 80% of the monitoring sites. Note that in this comparison, the FRP model was not assessed due to its poor performance (Section 3.2). The calibration and validation results for the 50 partial models are summarized in Table 5 along

with the performance of the full model calibrated to all 102 sites (see Figs. S6 and S7 in the
Supplementary Material for detailed comparison of model residuals of the partial calibration/validation).
Across constituents, the calibration performance of the full model was comparable with the 50 partial
models. Note the slightly higher calibration performance for the partial models of $NO_x$ compared to the
full model. This seems to be related to the generally lower percentages of below-DL data in the 50
randomly-chosen partial calibration datasets (14.1%-17.9%) compared to the full dataset (17.3%) – we
further discuss the impacts of below-DL data on model performance in Section 4.1. In addition, model
performance is highly consistent between corresponding calibration and validation, with most
differences in NSEs less than 0.1. These suggest that the spatio-temporal model performance is highly
robust and unaffected by the choice of calibration sites.
**Table 5. Comparison of model performances (as NSE) of the full model (Column 2) and the 50**
**partial models (Columns 3 to 5) with each calibrated to 80% randomly selected monitoring sites.**
**Columns 3 to 5 summarize the mean, minimum and maximum NSE values across the 50 runs,**
**where for each constituent, the top row showing calibration performance and the bottom row**
**showing the validation performance (i.e. at the 20% sites that were not used for calibration).**

The performance of the full model for each constituent is also compared with that of the three models
calibrated to the pre-, during and post-drought periods. In general, we observe consistent performance
for each constituent, across calibrations to the three periods of contrasting hydrological conditions
(Table 6, see Figs. S8 to S13 in the Supplementary Material for detailed model fittings). One notable
common pattern is that the performance for calibration and validation is more consistent during the
drought period than either the pre- and post-drought periods. However, this is most likely explained by
relative sizes of the calibration data sets, which are 3, 13 and 5 years for the pre-, during and post-
drought periods respectively.
Of all constituents (excluding FRP), TSS shows greater differences in model performances across
periods – especially when comparing the pre-drought calibration with its validation for the site-level
mean concentrations (Fig. 7). Notably, when calibrated to the pre-drought period and validated on both
the during- and post-drought periods, the validated model over-estimates most of the data (Fig. 7 (b));
and when calibrated to the during-drought period, the validated model slightly under-estimates pre-
and post-drought period TSS (Fig. 7 (d)).
**Table 6. Comparison of model performances (as NSE) of the full model and the three models**
**that were calibrated to the pre-drought (1994-1996), drought (1997-2009) and the post-drought**
**(2010-2014) periods. For each of the models, the calibration performance is shown on the top**
**row and the validation performance (i.e. over the periods that were not used for calibration) is**
**shown on the bottom row. See Section 2.1.4 for details of the calibration and validation**
**approach.**

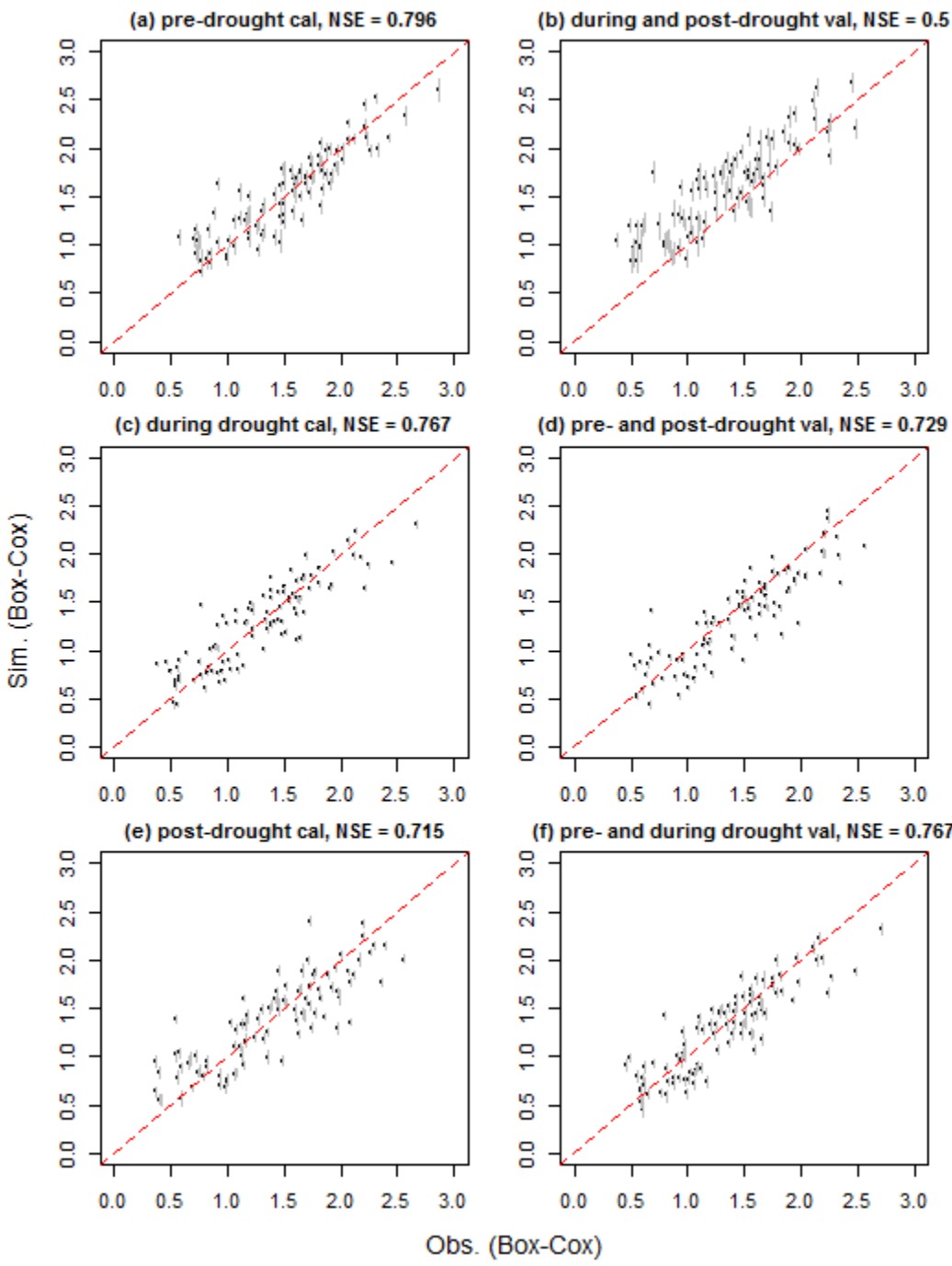

**Figure 7. Comparison of the TSS model performance, as the simulated against observed site-**
**level mean concentrations in Box-Cox transformed space. The left column shows calibration**
**performance for the model calibrated to the pre-drought (1994-1996), drought (1997-2009) and**
**the post-drought (2010-2014) periods, respectively; the right column shows the corresponding**

**validation performance for each period. The 95% lower and upper bounds of simulations shown in vertical grey lines and red dash lines show the 1:1 lines.**

The potential impacts of drought on TSS dynamics are further illustrated with the performance of the spatio-temporal model (calibrated to the full dataset with all sites and all data from 1994 to 2014) over the pre-, during and post-drought periods (Fig. 8). Both the during- and post-drought periods have consistently good performances, while the model underestimates most sites for the pre-drought period. This is consistent with Fig. 7 in suggesting a systematic decrease in TSS concentration since the drought began. The better performance of the full model during and after drought (Fig. 8) can be a result of the calibration period of the full spatio-temporal model – between 1994 and 2014 – which was dominated by the during- and post-drought periods.

In summary, Figs 7 and 8 together with Figs. S13-S17 suggest that whilst model performance for most constituents are not affected by the hydrological periods used for calibration and validation, the calibration period did have notable impact on TSS. Some possible causes are discussed in Section 4.3.

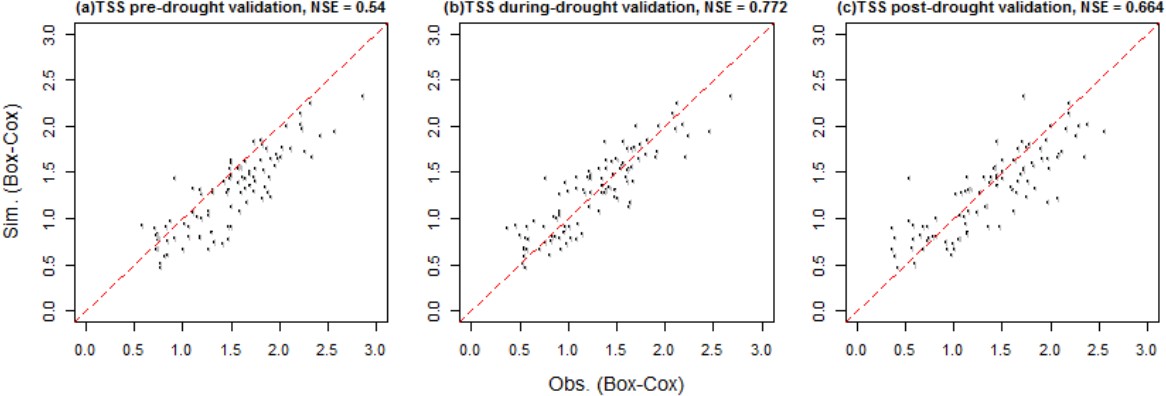

**Figure 8. Comparison of the performance of the full spatio-temporal TSS model calibrated to all data across a) pre-drought (1994-1996), b) during drought (1997-2009) and c) post-drought (2010-2014) periods, as represented by the simulated against observed site-level mean concentrations in Box-Cox transformed space. The 95% lower and upper bounds of simulations shown in vertical grey lines and red dash lines show the 1:1 lines.**

## 4. Discussion

### 4.1 Implications for statistical water quality modelling

In this study, we developed the first process-informed statistical model that is capable of explaining a reasonable proportion of water quality variability for a large spatial area of over 130,000km$^2$. Although the calibration data have relatively low sampling frequency (i.e. monthly), our model generally performs

satisfactorily in explaining the total variability in water quality. This demonstrates the effectiveness of
the Bayesian hierarchical modelling framework in predicting spatio-temporal variability in water quality
across large scales. The Bayesian hierarchical model is: a) more advantageous than other simpler
statistical water quality models with its more comprehensive and process-informed approach, and
capacity to represent varying temporal relationships across large-scale regions; b) less demanding for
input data compared with those required by fully-distributed, processes-based models. From a practical
perspective, this model has the potential to contribute to a number of management activities including
catchment planning, management and policy-making activities, specifically:
1)  The spatial predictive capacity can be used to identify pollution hot-spots and the catchment

conditions that are likely causes of high concentrations. This can be used to help identify target

catchment(s) to prioritize future water quality monitoring and management (Figs. 4 and 5);

2)  Further to 1), since water quality has been linked with catchment characteristics in this model,

it can also be used to assess potential impacts of alternative options of land use and land cover

change, as well as potential effects of climate change, on ambient water quality conditions;

3)  The model's temporal predictive capacity can identify changes in water quality due to changes

in hydro-climatic conditions and vegetation cover, and thus enabling attribution of detected

trends. On the other hand, any 'unexpected' trends can be identified to prompt further

investigation to identify causes (Figure 6 and Table 4). The model could also be used for

assessing the impacts of long-term catchment changes on water quality (Figures 7 and 8).

Despite the opportunities highlighted above, the model's performance also suggests some current

limitations of the modelling framework in the following situations:

*1)*  *High within-site temporal variability.* In Section 3.2 we have identified a general lack of

predictive power for temporal variability. The potential impacts of high temporal variability on

model performance is particularly evident for results of TSS, $NO_x$ and FRP in Fig. 3. Since our

model has already included hydro-climatic conditions and vegetation cover to explain temporal

variability, the unexplained temporal variability is likely due to other uncaptured temporal

drivers. These could be: changes in land use and land management, bio-geochemical processes,

or transit time of water through catchments.

2) *Presence of high proportions of below-DL data.* The full datasets for the three poorly modelled

constituents (FRP, TSS and $NO_x$) all have higher proportions of data below the detection limit

(38.2% 17.3% and 15% of all data, respectively) compared with other constituents. As

illustrated in Fig. 2, for each of these constituents, removal of below-DL data before model

calibration had created clear a truncation on the left-hand side of the distribution. This

substantially increases the degrees of skewness and discontinuity of the data, essentially

violating the assumption of normally distributed residuals and thus limiting model performance.

The model capacity to handle truncated data might be improved by model fitting approaches

explicitly designed for this issue. For example, Wang and Robertson (2011) and Zhao et al.

(2016) illustrated an approach to resolving the discontinuity of the likelihood estimation in

model fitting to data with presence of a lower bound such as zero rainfall values.

3) *Non-conservativeness of constituents.* The results indicate that the reactivity of the constituent

is broadly associated with performance, which suggest that bio-geochemical processes (e.g.

phosphorus cycling, nitrification/de-nitrification) can make water quality dynamics more

difficult for the model to capture. To better capture changes in reactive constituents, the model

may require greater consideration of and more extensive spatial and temporal data to represent

bio-geochemical processes. Examples include improvements on the process representation for

nitrogen cycling and the desorption and adsorption of phosphorus (Granger et al., 2010;Smyth

et al., 2013;Tian and Zhou, 2007).

As previously noted, our model was developed in a Box-Cox transformed scale to ensure the validity of
the statistical assumptions (see details on data transformation in Sect. 2.1.2), which shows limited
performance for high constituent concentrations when simulations are back-transformed to the
measurement scale (Figs. 4 and 5). However, our model approximately represents proportional changes
in water quality[1], which can thus help managers to understand proportional changes to inform practical
catchment management.
For future implementations, the established model structure and parameterization would be best suited
to within the study region. Before performing new simulations (e.g. for new monitoring sites or for
current study sites over a different time-period), the statistical properties of the new input datasets should
be checked to ensure that they are similar to the calibration datasets. To model new catchments outside
of the study region, a re-calibration of the model is required. This would involve extensive selection of
key predictors and model calibration, much as performed in this study and the two preceding ones
(Lintern et al., 2018b; Guo et al., 2019). A sufficiently long record length (e.g. 20 years) is ideal for such
modelling, as it ensures a reasonable understanding of the temporal variability to be obtained.
**4.2 Implications for water quality monitoring programs**
The current spatio-temporal model extracts water quality temporal variability from monthly data.
Utilizing data with higher temporal resolution may further strengthen the model capacity to explain
temporal variability, especially by capturing more information on water quality dynamics during flow
events. This may be possible into the future; however, current high-frequency water quality sensors
(Bende-Michl and Hairsine, 2010;Outram et al., 2014;Lannergård et al., 2019;Pellerin et al., 2016) still
have very high resourcing requirements that limits widespread deployment in operational networks.
Furthermore, changes in land use and management over time are currently not considered here as
predictors of temporal variability in water quality, which include but not limit to land clearing,
urbanization, tillage, fertiliser application and irrigation. This is due to a complete lack, or inconsistency
of available data. However, changes in land use/land management practices can occur over short time
periods, which can lead to increases in pollutant sources and changes to runoff generation processes
(e.g. Tang et al., 2005;DeFries and Eshleman, 2004;Smith et al., 2013). Therefore, our modelling

---

[1]All Box-Cox transformation parameters for water quality constituents are approximately 0 (Table S4), which means that the transformations are similar to a log transformation.

framework can potentially be improved by having additional monitoring data on the temporal patterns
of land use/land management to better capture their impacts on water quality.
**4.3 Potential impacts of long-term drought on water quality dynamics**
Results of model calibration and validation to different time periods suggest a systematic decrease in
TSS concentrations during and after the prolonged drought, in comparison with the pre-drought period
under the same spatial and temporal conditions. Such a shift is not observed for any other five
constituents analyzed (nutrients and salts) (Section 3.3).
A further analysis of the calibrated model parameters for pre-, during and post-drought periods suggest
that the effects of key spatial predictors do not vary much across periods (Figure S14). In contrast, the
effects of key temporal predictors highlight a clear shift in the role of antecedent flow (prior 7-day flow)
across different time periods (Figure 9). Specifically, the antecedent flow effects are mostly positive
across catchments before the drought, and shift to mostly negative during the drought. After the drought,
the antecedent flow effects have mixed directions among different catchments. Considering the limited
performance of the TSS model (i.e. substantial under-estimation of temporal variability in Section 3.1),
these changing relationships suggested in the calibrated parameters might be unreliable. However, this
should not affect the reliability of the observed change in TSS since the drought (Section 3.3), which
was based on the systematic differences of model fitting between different periods, revealing a broad-
scale patterns across the state on the drought influences.

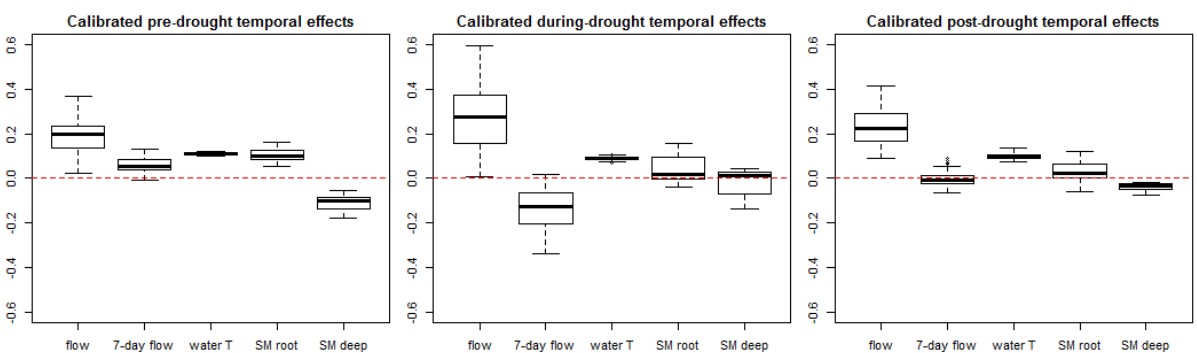


**Figure 9. Effects of the five key predictors for the temporal variability in TSS across 102 sites, summarized by the posterior mean of the calibrated parameter values for each predictor (box shows values across all sites), from left: flow, 7-day antecedent flow, water temperature, root-zone soil moisture and deep soil moisture.**

In the literature, impacts of the Millennium Drought on the hydrology and runoff regimes of south-
eastern Australia are well understood (van Dijk et al., 2013;Leblanc et al., 2012;Saft et al., 2015).
However, less is known about how this major and prolonged drought event has impacted water quality
(Bond et al., 2008). Previous studies on other drought events around the world mainly focused on
changes in water quality as responses to the reduced streamflow during drought. For example, reduction
in sediment levels during drought has been reported and attributed to lower erosion from the contributing
catchment, together with lower rates of solid transport associated with reduced flows (Murdoch et al.,
2000;Caruso, 2002). At a more local scale, increasing sediment concentrations during drought have also
been observed in streams adjacent to land with high densities of livestock and bushland, which both
constantly contribute to sediment load during drought, leading to elevated concentrations with lower
dilution rate (Caruso, 2002). Similar to sediments, the impact of droughts on stream nutrient and salt
concentrations have also commonly been understood as responses to reduced runoff generation and
streamflow. In catchments with no significant point-source pollution, nutrient concentrations typically
decreased during droughts (Mosley, 2015) with less nutrient leaching and overland flow, but may also
increase due to increasing livestock inputs at more local scales (Caruso, 2002). In contrast, catchments
with significant point-source pollution generally experience water quality deterioration during drought
due to reduced dilution (van Vliet and Zwolsman, 2008;Mosley, 2015). For salinity, concentration often
increases during drought with reduced dilution and increased evaporation (Caruso, 2002). This is
particularly evident for catchments that are more influenced by saline groundwater input as the relative
contribution of groundwater increased during drought (Costelloe et al., 2005).
In contrast to these previous studies, our findings suggest additional possible pathways along which
drought can affect stream water quality, that prolonged drought might have altered the relationships
between sediments and its predictors (Figs. 7 and 8). In contrast to sediments, our model suggests no
clear shifts in the dynamics of nutrients and salts in a regional scale. Our findings are in line with a few
previous studies which reported temporal changes in the concentration-discharge relationships for
sediments and nutrients, specifically, when comparing high- and low-flow conditions (Zhang,
2018;Moatar et al., 2017), as well as drought and recovery period (Burt et al., 2015). Our findings
provide extra dimensions to what would be offered by simple trend analyses using approaches such as
Mann Kendall test or Sen's slope (e.g. Smith et al., 1987;Chang, 2008;Hirsch et al., 1991;Bouza-Deaño
et al., 2008). Those approaches are only capable of indicating direction and magnitude of observed
trends. In contrast, our model was able to attribute the consistent upward shift in TSS concentration to
change in relationships between water quality and its key driving factors since the start of drought.
In addition, we also acknowledge that our ability to represent the pre- and post-drought conditions in
this study may be limited by the record length, since only 2 years of pre-drought and 4 years of post-
drought data were available. Once longer records build up, they will enable us to update our
understanding of the impact of this prolonged drought. We would be also able to conduct more
sophisticated investigations, such as comparing the impacts of long-term droughts versus individual dry
and wet years and events (e.g. Saft et al., 2015;Outram et al., 2014;Burt et al., 2015).

### 669    5. Conclusions

This study aims to address the current lack of water quality models that operate at large scales across
multiple catchments. To achieve this, we used long-term stream water quality data collected from 102
sites in south-eastern Australia, and developed a Bayesian hierarchical statistical model to simulate the
spatio-temporal variabilities in six key water quality constituents: TSS, TP, FRP, TKN, $NO_x$ and EC.
The choice of model predictors was guided by previous studies on the same dataset (Lintern et al.,
2018b; Guo et al., 2019). The model generally well captures the spatio-temporal variability in water
quality, where spatial variability between catchments is much better represented than temporal
variability. The model is best used to predict proportional changes in water quality in a Box-Cox
transformed scale, and can have substantial bias if used to predict absolute values for high
concentrations. Cross-validation shows that the spatio-temporal model can predict water quality in non-
monitored locations under similar conditions to the historical period and the calibration catchments that
we investigated. This can assist management by (1) identifying hot-spots and key temporal periods for
waterway pollution; (2) testing effects of catchment changes e.g. urbanization or afforestation; and (3)
identifying and attributing major water quality trends and changes.
Based on the above model evaluations, we discussed potential ways to further enhance the model
performance. In improving the modelling framework, alternative statistical approaches could be
considered to reduce the impact of below detection limit data on model performance. In addition, the
models could be extended to consider some key bio-geochemical processes to better dynamics in non-
conservative constituents (e.g., FRP or $NO_x$). Regarding data availability, the current models could
potentially benefit from improved monitoring of changes in land use intensity and management to be
able to include these drivers in the model. The inclusion of high-frequency water quality sampling data
may also extend the model's ability to represent temporal variability. However, high-frequency water
quality data are also typically highly variable with large noise. Therefore, the implication of such data
for the spatio-temporal modelling framework remains an open question, which needs further
investigation in future applications of this modeling framework.

## Data availability

All data used in this study were extracted from public domain. All stream water quality data were
extracted from the Victorian Water Measurement Information System (via http://data.water.vic.gov.au/,
provided by the Department of Environment Land Water and Planning Victoria). The catchments
corresponding to these water quality monitoring sites were delineated using the Geofabric tool provided
by the Bureau of Meteorology, via ftp://ftp.bom.gov.au/anon/home/geofabric/. We have listed the
sources of all other data for the spatial and temporal predictors of our models in Tables S1 and S2 in the
Supplementary Materials.

## Author contribution

All authors contributed to the conceptualization the models and the design of methodology. A. Lintern
and S. Liu contributed to the data curatiom. D.Guo carried out the formal analyses, visualization and
validation. J.A. Webb, D. Ryu, U. Bende-Michl and A.W. Western contributed to the funding
acquisition. D. Guo, A. Lintern, J.A. Webb, D. Ryu, S. Liu and A.W. Western contributed to the
investigation. D. Guo carried out project administration and coding to run the experiments. J.A. Webb,
D. Ryu, and A.W. Western contributed to the supervision. D.Guo prepared the manuscript with
contributions from all co-authors.

## Competing interests

The authors declare that they have no conflict of interest.

## Acknowledgement

The Australian Research Council, the Victorian Environment Protection Authority, the Victorian
Department of Environment, Land Water and Planning, the Australian Bureau of Meteorology and the
Queensland Department of Natural Resources, Mines and Energy provided funding for this project
through the linkage program (LP140100495). The authors would also like to thank Matthew Johnson,
Louise Sullivan, Hannah Sleeth and Jie Jian, for their assistance in the compilation and analysis of data.
All water quality data used for this project can be found on: Water Measurement Information System
(http://data.water.vic.gov.au/monitoring.htm). Sources of other data are provided in Tables S1 and S2
of the Supplementary Materials.

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

**Tables**

**Table 1. Key factors affecting the spatial and temporal variability for each of six constituents, as identified in Lintern et al. (2018) and Guo et al. (2019b), respectively.**

| Constituent | Key factors that affect spatial variability | Key factors that affect |
|---|---|---|
| TSS | Hottest month maximum temperature | Same-day streamflow |
| | Percentage area covered by grass | 7-day antecedent streamflow |
| | Percentage area covered by shrub | Water temperature |
| | Percentage cropping area | Soil moisture root |
| | Maximum elevation | Soil moisture deep |
| | Dam storage | |
| | Percentage clay area | |
| TP | Erosivity | Same-day streamflow |
| | Percentage area covered by grass | 30-day antecedent streamflow |
| | Percentage area covered by shrub | Water temperature |
| | Percentage area made up of roads | Soil moisture root |
| | Percentage cropping area | Soil moisture deep |
| | Average soil TP content | |
| FRP | Percentage area covered by shrub | Same-day streamflow |
| | Percentage cropping area | Water temperature |
| | Catchment area | Soil moisture deep |
| | Average soil TP content | |
| | Mean channel slope | |
| TKN | Percentage clay area | Same-day streamflow |
| | Warmest quarter mean temperature | 30-day antecedent streamflow |
| | Coldest quarter rainfall | NDVI |
| | Percentage cropping area | Water temperature |
| | Percentage pasture area | Soil moisture root |
| | Average soil TP content | Soil moisture deep |
| NO$_x$ | Annual radiation | Same-day streamflow |
| | Warm quarter rainfall | 30-day antecedent streamflow |
| | Hottest month maximum temperature | NDVI |
| | Average soil TP content | Water temperature |
| | Mean channel slope | Soil moisture root |
| | | Soil moisture deep |
| EC | Annual radiation | Same-day streamflow |
| | Annual rainfall | 14-day antecedent streamflow |
| | Wettest quarter rain | Water temperature |
| | Hottest month maximum temperature | Soil moisture root |
| | Percentage agriculture area | Soil moisture deep |
| | Percentage cropping area | |
| | Percentage area covered by shrub | |
| | Average soil TN content | |

**Table 2. The key catchment landscape characteristics that are related to the varying relationships of water quality and same-day streamflow across space, which were selected as the two predictors for the streamflow effect in our model. Two characteristics were selected to summary the variability of streamflow effects across space for each constituent, see Section 2.3 for details of the selection method. The corresponding Spearman's correlation (R, at p<0.05) between the effect of streamflow and each catchment characteristic is presented.**

| Constituent | Key factors that affect spatial variability in temporal effects | Spearman's $\rho$ (p<0.05) |
|---|---|---|
| TSS | Annual rainfall | 0.722 |
| | Hottest month maximum temperature | -0.575 |
| TP | Annual rainfall | 0.695 |
| | Percentage area used for cropping | -0.556 |
| FRP | Percentage agriculture area | 0.392 |
| | Percentage area underlain by mixed igneous bedrock | 0.314 |
| TKN | Annual rainfall | 0.713 |
| | Hottest month maximum temperature | -0.618 |
| NO$_x$ | Total storage capacity of dams in catchment | -0.493 |
| | Mean soil TN content | 0.458 |
| EC | Percentage area covered by grassland | -0.347 |
| | Percentage area covered by woodland | -0.317 |

**Table 3. Comparison of model performance for all records and only the above-DL records for each constituent.**

| Constituent | Above-DL records only | All records |
|---|---|---|
| TSS | 0.225 | 0.397 |
| TP | 0.433 | 0.445 |
| FRP | -1.920 | 0.199 |
| TKN | 0.658 | 0.630 |
| NO$_x$ | 0.216 | 0.382 |
| EC | 0.907 | 0.886 |

**Table 4. Model ability to capture observed water quality trends across all monitoring sites for each constituent. The percentages of sites where observed positive and negative trends are captured by the model are presented separately. Values in brackets indicate numbers of sites where corresponding positive or negative trends are observed. For detailed estimation of these percentages please refer to Sect. 2.2.**

| Constituent | % positive trends captured | % negative trends captured |
|---|---|---|
| TSS | 33.3 (12) | 85.0 (20) |
| TP | 82.1 (28) | 16.7 (12) |
| FRP | 47.1 (17) | 55.6 (9) |
| TKN | 81.1 (37) | 40.0 (10) |
| NO$_x$ | 68.6 (35) | 66.7 (27) |
| EC | 82.6 (23) | 77.3 (22) |

**Table 5. Comparison of model performances (as NSE) of the full model (Column 2) and the 50 partial models (Columns 3 to 5) with each calibrated to 80% randomly selected monitoring sites. Columns 3 to 5 summarize the mean, minimum and maximum NSE values across the 50 runs, where for each constituent, the top row showing calibration performance and the bottom row showing the validation performance (i.e. at the 20% sites that were not used for calibration).**

| Constituent | Full model | 50 CV mean | 50 CV min | 50 CV max |
|---|---|---|---|---|
| TSS | 0.397 | 0.413 | 0.376 | 0.439 |
|  |  | 0.382 | 0.292 | 0.513 |
| TP | 0.445 | 0.461 | 0.427 | 0.501 |
|  |  | 0.411 | 0.151 | 0.575 |
| FRP | 0.199 | 0.168 | 0.067 | 0.232 |
|  |  | 0.129 | -0.078 | 0.272 |
| TKN | 0.630 | 0.654 | 0.622 | 0.670 |
|  |  | 0.622 | 0.468 | 0.691 |
| NO$_x$ | 0.382 | 0.453 | 0.414 | 0.489 |
|  |  | 0.397 | 0.258 | 0.563 |
| EC | 0.886 | 0.893 | 0.882 | 0.903 |
|  |  | 0.875 | 0.809 | 0.924 |

**Table 6. Comparison of model performances (as NSE) of the full model and the three models that were calibrated to the pre-drought (1994-1996), drought (1997-2009) and the post-drought (2010-2014) periods. For each of the models, the calibration performance is shown on the top row and the validation performance (i.e. over the periods that were not used for calibration) is shown on the bottom row.**

| Constituent | Full model | Pre-drought calibration | During drought calibration | Post-drought calibration |
|---|---|---|---|---|
| TSS | 0.397 | 0.495 | 0.399 | 0.499 |
|  |  | 0.208 | 0.402 | 0.390 |

| | | | | |
|-----|-------|--------|-------|-------|
| TP | 0.445 | 0.477 | 0.438 | 0.525 |
| | | 0.421 | 0.474 | 0.411 |
| FRP | 0.199 | -1.336 | 0.187 | 0.204 |
| | | -1.406 | 0.197 | 0.024 |
| TKN | 0.630 | 0.649 | 0.650 | 0.711 |
| | | 0.566 | 0.648 | 0.610 |
| NOx | 0.382 | 0.443 | 0.426 | 0.509 |
| | | 0.394 | 0.471 | 0.393 |
| EC | 0.886 | 0.854 | 0.901 | 0.901 |
| | | 0.887 | 0.873 | 0.884 |

1001