# Peer review of "A data-based predictive model for spatio-temporal variability in stream water quality"

_Hydrology and Earth System Sciences, 2019_

## Referee Comment (RC1) · Anonymous Referee #1 · 20 Aug 2019

This manuscript presents a Bayesian-based approach to analyze spatio-temporal variability in stream water quality. The approach is demonstrated with an application to a large set of monitoring data in Australia. Overall, I think the manuscript is well written and will become a worthwhile contribution to the hydrological community. The proposed method also has the potential of being applied to monitoring data elsewhere. I do have some major and specific comments for the authors, which I hope can help improve the manuscript. I recommend its publication after the following comments are addressed.

General comments:

1. On model applications: I recommend the authors to add a separate sub-section to provide some guidelines to potential users of the proposed approach, including at least

the computer running time of the model, the required no. of stations and required no. of water-quality samples for running the model, as well as approaches to evaluate if the model does a reasonable job.

2. On calibration/validation analysis: The authors randomly selected 80% of the sites for calibration and used the remaining 20% for validation, and repeated this validation process for five times for each constituent, in order to evaluate the sensitivity of the model to the monitoring sites. Could you justify the use of five times for each constituent? If this cannot be easily justified, I recommend the authors to increase the replicates from five to a larger number (say 30 or 50). The results may be summarized as boxplots instead of Table 2, which can provide an overall evaluation of the model's ability to capture the dynamics of the different constituents.

3. On the below-LOR data: The authors argue that the model performance is related to the proportions of below-LOR data. The results appear to support the argument that model works better when the proportion of below-LOR data is low. Can you further prove this? The authors may quantify the proportion of below-LOR data for each monitoring site and conduct a separate analysis for sites of low proportions vs. sites of high proportions (perhaps 50% of sites for each group?) and see if the performance varies significantly between the two groups. This analysis may be implemented for each constituent.

4. On monitoring data: In this pilot application of the proposed approach, water-quality variability is modeled based on monthly monitoring data. First, I think the authors have made a good point that high-temporal-resolution data can further strength the model capacity to explain temporal variability in water quality. Second, I think the approach's ability to reasonably capture that variability based on just monthly monitoring data is a big strength of the proposed approach. After all, a lot of the monitoring records at many locations are based on a monthly sampling scheme. This aspect should be more emphasized. Third, how about high-flow sampling? Many monitoring programs supplement regular sampling with targeted stormflow sampling to capture concentration variability during storm events (e.g., Chanat et al., 2016; Zhang et al., 2017). It is widely acknowledged that sediment and particulate constituents are heavily affected by storms. However, I cannot find any discussion of this aspect in the manuscript. Would you expect the models to be further improved if the monitoring data contain targeted stormflow samples? References: • Chanat et al. (2016) (URL: http://dx.doi.org/10.3133/sir20155133) • Zhang et al. (2017) (URL: https://doi.org/10.1016/j.jhydrol.2016.12.052)

5. On key controlling variables: Table S5 and Table S6 may be combined to a single table and moved to the main text. I think this information is critical and deserves to be placed in the main text.

Specific comments:

6. The term "filterable reactive phosphorus (FRP)" may be replaced with "soluble reactive phosphorus (SRP)". I think the latter is more widely used.

7. L46: Add a few more references to support the argument "differ significantly".

8. L56: Provide some specific examples on "other catchment conditions". One could be antecedent condition, which is heavily discussed in the manuscript. In this regard, Zhang et al. (2017) (URL: https://doi.org/10.1016/j.jhydrol.2016.12.052) provides a study on how antecedent conditions affect the estimation of riverine constituent concentrations. This is also relevant to your discussion at L430.

9. L103-L107: These sentences can be removed. I think the subsection titles are already very clear.

10. Figure 1: Use a different color or a larger font for the dots to make them more clear.

11. L130: Add a few more references to support the argument "widely known to influence water quality condition".

12. L131: "literature review" is vague. Could you briefly describe how it was conducted?

13. L164: I do think one or two references should be provided for "Box-Cox transformation" to help readers. The meaning of the parameter lambda should be also briefly described.

14. L352: This ranking is roughly consistent with particular constituent vs. dissolved constituent. Any comment in this regard?

15. L366: The authors list here some processes for N. How about processes for P?

16. L206: What is the "Rhat" value? Please clarify.

Editorial comments:

17. L71: Fix the usage of ". . .not only. . .but also. . ." In addition, "limits" should be "limit".

18. L76: The model built. . . –> The model was built. . .

19. Equation 3 and Equation 4: For the betas, consider using subscript instead of dash.

20. L180: "General speaking" –> "Generally speaking"

21. L317: Fix "a results of"

22. L382: Fix "oppourtunities"

23. L417: Fix "droguht"

24. L420: Similarly to –> Similar to

Comments on the SM:

25. Supplementary Materials lack of "title-page" information.

26. Table S4: Change "lambda" to its Greek form.

[Figure]

342, 2019.

---

## Referee Comment (RC2) · Matthias Gassmann (Referee) · 27 Aug 2019

**Review of hess-2019-342: "A predictive model for spatio-temporal variability in stream water quality"**

**by Guo et al.**

**Content**

This paper introduces a Bayesian hierarchical model for spatio-temporal prediction of water quality variables in Australia. After model construction and validation, the results are discussed in terms of influences on prediction accuracy and regarding the influence of a long drought period on average suspended sediment concentrations. The paper concludes with recommendations regarding model improvement.

**General comments**

Generally, the paper is well written and the methods and results are interesting. However, I have some major concerns regarding (i) the statements drawn from the results, (ii) influences on the simulation accuracy and (iii) the focus of the study. These major points need to be clarified before publication.

*Focus of the study*

The study is introduced as a new model for water quality prediction. It is mentioned that the construction of the site-specific model was already published in two preceding papers (Lintern et al., 2018b; Guo et al., 2019). It is not really clear which additional information this paper provides. In the discussion section, there is a long chapter about the influence of a long-term drought to TSS concentrations, which was found as a by-product (?) of the study. The papers ends with conclusions suggesting higher-frequency sampling data, which was not analysed in this study at all. Thus, the study lacks a clear focus and coherent conclusions.

*The influence of LOR on simulation accuracy.*

- First of all: What is LOR (Limit of Reporting)? Is it a limit of detection (LOD) or a limit of quantification (LOQ) or something different? Which value was used for the calculation of Nash-Sutcliff (Neff) efficiency if the measurement was below LOR? Zero? Half the LOR? Please clarify.

- For model construction, the values below LOR were excluded due to statistical reasons and due to the fact that these low concentrations were of less interest. Thus, why were the values below LOR included in model validation at all? Please clarify.

- Later on it is analysed that the fraction of LOR on total measurement values influences model performance, especially the P fractions and TSS. The discussed reasons are mainly methodical/statistical. I think, the effect of LOR on model performance might also be a secondary effect: the parameters with a high proportion of LOR are mainly those with the highest natural concentration variability, since their concentration peaks are event-driven. Thus, monthly grab samples might capture peaks or not. Since some of the catchments are as small as a few km², even the specific time of a day might influence the sampled concentration to a large extent. Thus, the probability of sampling low between-event concentrations is higher for P and TSS than for e.g. Nitrate. Therefore, the low model performance might rather be an effect of the overall lower information content of the samples, which results in models which are based on a lower information content. What do you think?

*Influences of drought on TSS*

During the modelling process, the authors note, that a long-term drought influenced TSS concentration, which is a really interesting observation. However, I do not understand why a model is required for this analysis. Wouldn't simple statistics (such a t-test or Mann-Whitney-U-Test) have done the same job? I don't see that this is a special result of this model application.

*Meaning of factors.*

Since the model is a (multidimensional) statistical model, the explaining variables (factors) not necessarily contain process-based meaning for the target water quality parameters. For example, the water temperature is an explaining variable for temporal variability of TSS (Table S6), which is not really clear to me. In L.15-17 it is stated that the paper addresses the key controls (factors) explaining water quality variability, but an in-depth analysis and discussion is missing in the text. I would encourage the authors to even discuss the factors in more detail or to change the focus of the paper.

**Specific comments**

L. 1-2: The title "A predictive model for spatio-temporal variability in stream water quality "suggests a generic model for different sites and different water quality parameters. However, the described model is very site-specific. Thus, I would suggest to change the title to a more site-specific one, probably including the region or similar, including the applied method.

L. 71: Change "…quality can not…" to "…quality not…"

L. 76: Change "…model built…" to "… model was built…"

L.76-78. It is stated that the model was constructed and published in two previous papers. Please elaborate on the additional information this paper provides.

L. 79: It is stated, that this study aims at bridging the gap between fully distributed and statistical models. Well, what is this model if not a statistical model? Probably, it was meant to bridge the gap between fully/semi-distributed and lumped models.

L. 154-156. During the Box-Cox transformation of the data, the high sampling values lose their significance, especially for goodness-of-fit calculations. This effect can be seen after back-transformation (figure S13), which results in low Neff values. Thus, how is the statement "poor water quality conditions…were our primary concerns…" compatible to the fact that the data was transformed?

L. 159. Insert a blank between "as each"

L. 186. "… via a Spearman correlation analysis" (note the typo "analyses"). Please add the correlation coefficients and the p-values in the supplement.

L. 246. "…in Sect. 4.2." Isn't it section 4.1?

L. 265. Fix "… is also show…"

L. 414. Fix "For examples, …"

L. 418 Fix "adjscent"

L. 449-451. In the beginning, this paper aims at introducing a model. In this lines, the reader has the impression, that the main aim of this paper is the analysis of drought on TSS concentrations. Please think about the focus of the paper.

L. 466-469: "1) collection … in the model". These are not a results/conclusions of this study. Data frequency was not evaluated in this study.

L. 469-470. "These improvements will be very helpful…" How?

---

## Referee Comment (RC3) · Mark Honti (Referee) · 29 Aug 2019

**GENERAL COMMENTS**

The study describes a Bayesian statistical model of selected water quality variables in 102 catchments. The model successfully described both the spatial and temporal variability of certain variables, and performed quite well at describing the site-specific means for all variables. Based on the results, the model can serve as a valuable prediction tool in the calibration region (and potentially adapted elsewhere too).

The main issue with the manuscript is that the otherwise valuable work is presented in an unsuitable (and constantly evolving) context. The title appropriately focuses on the main element of the study, the model and emphasised predictions as the primary

field of utilisation. In the Abstract the motivation for the study is summarised as: "To address this [knowledge gap compromising present water quality models], we developed a Bayesian hierarchical statistical model to analyse the spatio-temporal variability in stream water quality across the state of Victoria, Australia." This shifts from predictions to analysis and promises that the model will cover knowledge gaps presumably by revealing so far unknown relations between water quality and its drivers. Interestingly, this objective is not featured in the Introduction. There it reads: "Our approach aims to bridge the gap between fully-distributed water quality models and statistical approaches to provide useful information for catchment managers, especially for large-scale water quality assessments." This alters the context again, now the model is meant to be a "missing link" between very detailed (deterministic) models and simple statistical tools and the raison d'être is to serve catchment managers. These context shifts do not help to assess the values of the study and generate expectations that are not fulfilled later.

Unfortunately, none of the above alternative contexts is completely followed in the Results and Discussion. The Results consist almost exclusively of performance indicators calculated and plotted in transformed scale. The Discussion focuses on the effects of the drought period on model performance and future development directions without mentioning potential major obstacles and pitfalls (gathering more detailed data and developing more detailed models is an idealistic recipe). The manuscript would greatly benefit from following a clearly defined logical structure, objectives and featuring topics that are truly relevant for the work.

Performance indicators should not occupy all the Results section. There is much more to show about the model, especially considering the ideas that show up in the present Introduction and Abstract. The potential topics include:

- Untransformed comparison of measured and modelled time series for selected catchments

-The needs of catchment managers with respect to predictions and how this model fulfils them (If it does so. If not, management should not be emphasised so much).

-Key controls and mechanisms governing water quality. What do we learn from this study compared to Guo et al. 2019 and Lintern et al 2018a, 2018b?

-The grade of intrinsic randomness (and its compatibility with management), predictability of water quality variables.

-Model limitations: implicit assumptions, conditionality on the calibration set and the present layout of calibration units (what would happen if the model was calibrated on merged catchments?)

-Spatial and temporal distributions of validation errors, their relationship with model development alternatives.

SPECIFIC COMMENTS

Lines 15-16: In my opinion it is not the lack of understanding, but the lack of information. The effects of many key controls on water quality are well understood, albeit in an isolated, idealised context. It is clear, for example what certain polluting sources (like a WWTP effluent, a plot of arable land, etc.) do, how different landcover types affect the transport of pollutants along a specified pathway. The problem with the modelling of stream water quality on the (sub)catchment scale is that numerous key factors and controls act together and in practice there is no hope to get relevant information on all/most of them. That's why detailed and dynamic models fail on all components except those that behave quite simply and are not affected by too many factors. The challenge of modelling is to include the relevant factors AND the necessary information about them. So I would rephrase the sentence to mention that despite the long history of research there are too many key controls and very high complexity in both space and time compared to the available information.

Line 16: Even if there would be a lack of understanding (which I doubt, see previous

comment), how would this issue be addressed by a Bayesian statistical model? Statistical models build on covariance instead of causal relations and therefore are rarely suitable for modelling conditions that are different from the calibration dataset in any significant aspect — which is the primary objective of most modelling exercises.

Line 20: Please mention how FRP relates to the more commonly known Soluble Reactive Phosphorus (SRP).

Line 21: The abbreviation of "NOx" is not the best choice, as this is a widely known name of the air pollutant group of gaseous nitrogen oxides. Why not "NOi" or something else?

Lines 21-22: Yes, the model described variation, but above an improvement of understanding is promised.

Lines 29-30: How would a statistical model include those mechanisms that govern non-conservative constituents? Such a development would indeed be a major step forward, but it is definitely not trivial.

Line 32: High frequency data often reveal phenomena that are typically not parts of models and therefore model performance further declines.

Line 33: Besides the classical landuse, agricultural activities (ploughing, fertiliser/pesticide application, livestock handling practices, etc.) would need to be known too.

Lines 40-42: Unpredictable variability does not preclude management. Robust measures can address issues without having to predict the full dynamics. It is well known that the elimination of pollution sources and artificial hydrological factors improves water quality. If the statement in lines 40-42 was true, water quality management would not exist yet.

Lines 42-46: This is a bit lengthy description of the high variability in both space and time. Please consider compressing.

Lines 46-51: Briefly, there are allochthonous and autochthonous emissions and both are subject to transport. Please consider compressing.

Lines 55-59: This listing is somewhat odd. Emission dynamics are completely missing, others are a bit over-detailed and supported with arbitrary references (is the importance of temperature only known since Robert and Mulholland, 2007?).

Lines 60-62: This sentence contradicts the abstract statement (lines 15-16). Water quality modeling faces high epistemic uncertainty, unpredictable variability stems rather from an information gap than the lack of understanding. And what do you mean here by "larger scales"? And please include why effective policy and mitigation need information on variability.

Lines 66-69: It would be worth to mention that most statistical models have weak explanatory and predictive power and therefore it is difficult to use them for designing management interventions.

Lines 71-72: Please check and fix this sentence, by e.g. deleting "can" or any other way.

Lines 74-80: After mentioning management so many times above, one would expect a brief summary about the requirements of managers against water quality models plus a sentence in the objectives on how the current model would fulfil these.

Line 103: Please fix "Beyesian".

Line 112: Please delete "however". Either you describe data processing or not. The present formulation suggest that you don't want to describe it, but later — reluctantly — still do so.

Line 132: Please briefly mention the forms and indicators of landuse considered among the drivers, because these are non-trivial.

Line 143: You mean "area-specific streamflow"? Streamflow also has the unit of volume

/ time.

Lines 144-149: How did you convert 2D climatic data to soil moisture? This must have included a complete soil hydrological model, but no hints are given in the main text.

Lines 156-157: Low flow days often mean the periods of concern with regard to water quality. What was the case here?

Lines 162-166: This means that you conditioned the transformation on the dataset. Since the predictive nature of the model is emphasised, please explain the procedure of including new catchments. What to do when the new data suggest a different transformation parameter?

Lines 172-174: A random forest approach could have been an alternative for the selection process.

Lines 179-183: Aren't these results? Since management is emphasised in the introduction, how would you reflect on the final set of key factors? Climate is close to impossible to manipulate, temperature, soil moisture and streamflow are difficult. Why no direct human factors other than landuse?

Lines 194-196: What is the rationale behind the half-normal prior? What is the advantage compared to an exponential? The half-normal suggests that relatively small standard deviations are equally likely, while the exponential prioritises as small std. deviation as possible. Please justify your choice.

Lines 212-214: This is a rather extreme test, why do you expect the model to describe the below-LOR data, after excluding all of them from the calibration dataset. Would a good fit mean that below-LOR data follow the same rules as above-LOD data do? Line 217: The verb "suggested" sounds weird to me here.

Lines 238-240: FRP is a subset of TP. TP has complicated relations to TSS. The FRP-TP relationship is governed by several (fast) biochemical processes simultaneously. Consequently, it is no surprise that FRP is hard to model without considering all these

intricate interactions. By the way, a negative NSE suggests that the model entirely failed to capture any of the real dynamics (negative NSE means that a constant model at the mean would perform better).

Figures 2-3: It would be great to see some visualisation beyond 1:1 plots in transformed space (of unknown transformation parameters unless one digs them up from elsewhere).

Lines 268-269: This sentence is not necessary, the section title tells the same.

Lines 269-270, 273: Please delete the "Note that . . . in Sect. .3.1." sentence and add "We exclude the FRP model from the analysis due to its poor performance (section 3.1)." into Line 273 after "monitoring sites.".

Tables 1-2: These tables are all about calibration indicators, and not the subject of the model. These could be moved to the SI. Why not showing something about the factors? The introduction promised filling some knowledge gaps yet we do not learn about anything except performance indicators (and later the influence of drought on them in table 3).

Line 324: The Results section is over, yet the roles of "key controls", the proportions of "inherent randomness" both remain untold. The primary value of such a model is its information content, which is embodied in the relationships that turn inputs to outputs using the parameters. Model performance indicators are important too, but in a secondary sense: they help to assess the quality of information that can be obtained from the model. Here the reader learns about the model performance in various cases, yet the lesson can't be learnt. What governs the different water quality variables? Are there covariations between the variables? Are certain models similar to others? Are errors clustered in certain situations? Which environmental factors influence the variables, how sensitive are they to the most important one? Etc.

Lines 333-334: Would be more positive to start with the opportunities and afterwards

with limitations.

Lines 334-335: Or when the variability is high and explanatory power is weak. Very low FRP values could be much better simulated given that the model knows all the influencing factors and processes.

Line 336: This can also be by chance. TKN and EC are "more conservative" than the others, and have much weaker relations to sediment.

Lines 347-359: It is true that transformation increases the distance between distinct values close to the numerical resolution of data, which violates the linearity assumption. But when you do not transform, linearity is violated by default (as one of the aims of transformation is to reduce nonlinearity). Besides the alternative model structures mentioned, a practical solution is to perturb the data with random small values (small fraction of numerical resolution), which dissolves the discrete bands of the low values without significantly altering the data. This is basically the same as "measurement noise" beyond the resolution of the time-series.

Lines 360-361: Yes, this was obvious from the start. That's why the "positioning" of the model study is not optimal. The applied methodology tested whether temporal / regional differences could be replicated by a simple statistical model that lacks any mechanistic background. The exposition of knowledge gaps, management-relevant factors, general predictive power for ungauged catchments create expectations that simply cannot be fulfilled by this model. A lot of mechanistic knowledge is available for these water quality variables, no single bit of this knowledge is reflected by the model structure. A more realistic context would have been to investigate the overarching patterns in this region of Victoria, emphasising that the model only considers emissions only implicitly, through landuse, which in turn assumes similar human activities in the same landuse type. The results are completely in line with previous experiences, more conservative and less sediment-related variables are easier to predict than the others. The model can be a valuable predictive tool, but only in the region of calibration and

only for those water quality variables, for which have the model performed acceptably.

Lines 364-369: Making the model more detailed can potentially lead to a dead end. Non-linear statistical model structures may perform a bit better, but need more data for a meaningful calibration and still often lack the mechanistic background, and are much more complicated numerically. Adding descriptions of different mechanisms to the model either moves it towards a deterministic direction, which is a wrong way for this spatial and temporal scale because data will anyway appear to be at least partly random due to the lack of information on all relevant drivers, or leads to a stochastic-dynamic model, which is extremely complicated and difficult to calibrate.

Lines 372-373: If this was an issue, why don't we learn about the "real-world" (=non-transformed) model accuracy earlier? The NSE values and the figures are all in trans-formed space, so it is difficult to judge what these mean for the practice.

Lines 375-377: I don't understand this example. Completely usual floods often bring much more sediments in almost pristine mountain catchments. Why would such an event be an alarm for management?

Lines 377-379: How? This should have been the main topic if the logical line of the Introduction was followed. How strong is the predictive power of the calibrated models considering practical needs? Are they suitable for real forecasting either for the far future or for shorter periods during operative management?

Lines 384-386: The references are "too new" for this statement in its present general form. Commercial solutions for online monitoring with <10 minute resolution is available for turbidity (proxy for TSS), temperature, EC, chlorophyll, dissolved oxygen since at least 20 years. Nutrient sensors are indeed newer, yet they are often not sensitive enough to yield meaningful data in surface waters (unless they are heavily polluted).

Lines 386-388: How would you apply remote sensing in stream networks? Except for larger rivers (and of course, lakes and reservoirs), these water surfaces are difficult to

analyse because the number of "clean" pixels without any terrestrial or littoral influence is very low or even zero.

Lines 390-391: There may be better (older, original) references for this. This is known since at least 30 years.

Lines 391-397: Please remove, this is too case-specific.

Lines 398-399: This is the exact reason why models fail despite the rather solid understanding of mechanisms (and this is a data or information gap and not a knowledge gap). Relevant, representative, and accurate data on such activities is close to impossible to obtain, even for smaller regions or shorter periods. Therefore, the temporal and spatial variability of these contribute to apparent "inherent randomness" and undescribed variance (the difference of NSE from 1) and weaken the predictive power of models. At the moment the solution to this issue remains an open question even for the past/present, not to mention the potentially changing practices of the future.

Lines 422-423: Direct livestock input may increase concentrations during drought.

Lines 438-443: As the results of this study showed, this would be a hard job without implementing at least a few mechanistic features in the model. However, more features would require more data, potentially beyond the scope of the presented dataset.

Lines 455-457: This is a crucially important sentence. I would add explicitly that the model is not only bound to the period, but also to the region for which calibration took place.

Supplementary material: Figures could be structured better graphically. When 4x4 panel units are to be seen, please structure the figure so that the units get obvious. Please indicate the contents in the subfigure title. Print Box-Cox or log-sinh transformation parameters on figures or in the caption, because without knowing the strength of transformation it is difficult to judge the quality of fit.

---

## Referee Comment (RC4) · Anonymous Referee #4 · 10 Sep 2019

Review of manuscript "A predictive model for spatio-temporal variability in stream water quality" submitted to HESS (HESS-2019-342).

This manuscript describes a statistical modelling exercise for stream water quality in Victoria, Australia. The manuscript is well written, however, I have some concerns regarding the modelling framework, performance measures, site bias, and results in drought impact. These comments are outlined below, and need to be clarified before publication.

1) Model framework

   While I understand the notion of using catchment spatial variables to represent site-level mean (which is the focus of the published Lintern 18 paper), and using temporal variables to represent deviation from the mean (which is the focus of the published Guo 19 paper), I do not understand equation 6 – why is it necessary to add additional catchment characteristics in the temporal component? Why 2 variables? What's the implication of this equation for the framework overall? i.e. the framework started with distinct spatial and temporal components, but ended with the temporal component also include spatial variables? Wouldn't that means the spatial variables are double counting, i.e. does this lead to the model overly focusing on spatial variability while less representing temporal variations? In any case, this need to be explained better in the manuscript.

2) Model performance measures

   a. The manuscript uses long-term mean concentrations frequently in the result and discussion sections (e.g. Figs3-5). My understanding is, based on equation 2, the long-term mean results would be very close to spatial variability, while the temporal component does not have much role in determining the long-term mean:

   $$\text{Long} - \text{term mean in model results for k time steps} = C_j + \frac{\sum \Delta ij}{k}$$

   Assuming $\sum \Delta ij$ can be close to 0 as the positive and negative derivations more or less cancel each other out.

   If this is the case, then I'm not sure the long-term mean results are representative for both spatial and temporal variability, and the authors may consider using different result measures to better demonstrate the model's ability to represent spatial AND temporal variability.

   b. The NSE values for 4 of the 6 constituents are not great. Based on a widely used classification in water quality model performance measures (Moriasi etal 07), the model performance (i.e. NSE values) for these 4 constituents are "unsatisfactory", while that for TKN is "good", and EC is "very good". While it's perfectly fine to report results even if they are not great, it is questionable to use these 4 poorly performed models for further inference, i.e. change in system response for TSS since drought. Granted, the authors used the long-term mean concentration results for TSS (which have higher NSE values), then it's back to the previous comment regarding the long-term mean concentration may not adequately represent temporal variability.

3) Site bias

a. The areas of sites are highly diverse, from 5km2 to 16,000 km2. It's reasonable to expect that these different sized sites may be dominated by different processes, e.g. smaller sites may be constituent supply driven, while larger sites may be transport driven. These differences may be translated to different explanatory variables for these sites. But in the model, these sites share the same explanatory variables AND model parameters (ie the betas). The implications needs to be discussed, e.g. if there're more sites with large areas, then the model may bias towards representing large catchments, and the explanatory variables selected does not have strong predictively power for smaller catchments, and thus leading to poor model performance.

b. Data transformation: the authors chose to transform observation data, rather than back-transform modelled data. There are a few issues with transforming observation data: 1) the transformation involves additional parameters (such as lambda, instead of a straight transformation, e.g. logx), thus the "observed" data is in effect a "modelled" data, albeit a simple model. 2) The observation data across sites is transformed using the same parameter value (mean), thus the site bias issue in the comment above also applies. 3) the choice of transformation (log) leads to a decrease in the sensitivity of large values due to the log() function, and increase the sensitivity of small values. Thus, it is unclear to me whether using transformed observation data is any better than back-transforming modelled data. These implications need to be pointed out in the manuscript.

4) Results in drought impacts
a. Assuming the model is appropriate for inference (i.e. have good enough performance measure), a better (more insightful) way to demonstrate the impact of drought could be to show what the parameters (beta) for pre and post drought models are. This is because (I assume) these parameters represent the system behaviours, i.e. how strong different explanatory variables are to predict concentrations.

Other comments:

1) Pg 17, L374: please explain why "out models are very useful in representing and predicting proportional changes in concentrations"?
2) Maybe consider putting supplement tables S5 and S6 in to main text as these are important part of the model.

Reference:

Moriasi, D. N., Arnold, J. G., Van Liew, M. W., Bingner, R. L., Harmel, R. D., & Veith, T. L. (2007). Model evaluation guidelines for systematic quantification of accuracy in watershed simulations. Transactions of the ASABE, 50(3), 885-900.

---

## Author Comment (AC1) · 16 Oct 2019

**Responses to Reviewer Comments on "A predictive model for spatio-temporal variability in stream water quality" (RC4)**

Our proposed revisions are underlined.

**General comments**

This manuscript describes a statistical modelling exercise for stream water quality in Victoria, Australia. The manuscript is well written, however, I have some concerns regarding the modelling framework, performance measures, site bias, and results in drought impact. These comments are outlined below, and need to be clarified before publication.

1. Model framework
   While I understand the notion of using catchment spatial variables to represent site-level mean (which is the focus of the published Lintern 18 paper), and using temporal variables to represent deviation from the mean (which is the focus of the published Guo 19 paper), I do not understand equation 6 –

1.1 Why is it necessary to add additional catchment characteristics in the temporal component? Why 2 variables? What's the implication of this equation for the framework overall? i.e. the framework started with distinct spatial and temporal components, but ended with the temporal component also include spatial variables?

*Our modelling framework accounts for spatial variation in the parameter of each predictor that has been selected to explain the temporal variability, which were observed in Guo et al. (2019), as well as in Musolff et al. (2015) from a separate dataset. Therefore, the purpose of Eqn. 6 is to explain these spatial variations and thus enabling spatial prediction of those temporal effects according to catchment characteristics. This equation essentially makes the modelling framework fully spatio-temporal (i.e. being able to predict any location at any time-step). The choice of two variables was mainly due to the consideration of controlling model complexity (i.e. number of parameters).*

- *Musolff, A., Schmidt, C., Selle, B., and Fleckenstein, J. H.: Catchment controls on solute export, Advances in Water Resources, 86, 133-146, https://doi.org/10.1016/j.advwatres.2015.09.026, 2015.*

1.2 Wouldn't that means the spatial variables are double counting, i.e. does this lead to the model overly focusing on spatial variability while less representing temporal variations? In any case, this need to be explained better in the manuscript.

*We do not agree with your opinion that using additional spatial variables to explain temporal variability is redundant in our models. We believe that the reviewer is concerned about considering catchment characteristics twice in both Eqn. 3 and Eqn. 6; however, we acknowledge that these two sets of catchment characteristics served contrasting purposes. In Eqn. 6, the two additional catchment characteristics represent spatial variation in the relationships between temporal variability in water quality and its key predictors (e.g. hydro-climatic conditions, vegetation cover). For example, the impacts of streamflow on temporal changes in water quality are stronger at some catchments than at others, and these differences can be explained with additional catchment properties. This contrasts from the purpose of Eqn. 3, where uses a separate set of catchment characteristics to explain the spatial variation in ambient (average) water quality conditions (with more details in Lintern et al., 2018b). Therefore, both sets of spatial predictors serve unique purposes and are necessary components of the models.*

*To address Comment #1 and improving the clarification of modelling framework, we will:*

1) *add brief description of Eqn. 6 in Section 2.1 to better justify the purpose of including this equation. We will also further emphasize this additional modelling capacity (i.e. modelling temporal variability across catchments) that we gained from Eqn.6, apart from the two preceding studies.*
2) *present additional results and discussions on the key drivers for varying temporal relationships across catchments to illustrate the value of this specific model component.*

2. Model performance measures

2.1 The manuscript uses long-term mean concentrations frequently in the result and discussion sections (e.g. Figs3-5). My understanding is, based on equation 2, the long-term mean results would be very close to spatial variability, while the temporal component does not have much role in determining the long-term mean:

Long–term mean in model results for k time steps= $C_j + \frac{\sum \Delta ij}{k}$

Assuming $\sum \Delta ij$ can be close to 0 as the positive and negative derivations more or less cancel each other out.

If this is the case, then I'm not sure the long-term mean results are representative for both spatial and temporal variability, and the authors may consider using different result measures to better demonstrate the model's ability to represent spatial AND temporal variability.

*Thank you, this is a very good point and we confirm that your interpretation about the spatial and temporal variabilities are all correct. We agree that the existing results presented on model performance are predominantly focused on spatial variability. To improve this, in the revision we will present more results on how the model represent temporal variability in the Results section. This is in line with a number of important additions that we propose to present, to improve a) the presentation of different aspects of model performance and b) the illustration of model utilities that are useful for catchment management. In specific, we propose to include the following topics:*

1) *Modelled and observed time-series at selected catchments, to illustrate model capacity to predict trends and changes over time;*
2) *Proportions of spatial and temporal variabilities explained for each constituent, to illustrate the importance of the two variability components for each constituent, and how they can be explained by our models;*
3) *Extending the existing model cross-validation from 5 replicates to 50 replicates, to provide a more comprehensive summary on model sensitivity to calibrated dataset.*

*We believe that these additions could add more evidences on the model capacity to represent temporal variability.*

2.2 The manuscript The NSE values for 4 of the 6 constituents are not great. Based on a widely used classification in water quality model performance measures (Moriasi etal 07), the model performance (i.e. NSE values) for these 4 constituents are "unsatisfactory", while that for TKN is "good", and EC is "very good". While it's perfectly fine to report results even if they are not great, it is questionable to use these 4 poorly performed models for further inference, i.e. change in system response for TSS since drought. Granted, the authors used the long-term mean concentration results for TSS (which have higher NSE values), then it's back to the previous comment regarding the longterm mean concentration may not adequately represent temporal variability.

*We agree with the reviewer that the NSE achieved in our models is not as high as those recommended in Moriasi et al. (2007). However, this discrepancy should not be a key concern for our models. Firstly, we would like to point out that the water quality models reviewed in Moriasi et al. (2007) were all physically-based, spatially-distributed models (SWAT, HSPT, DRAINMOD-DUFLOW and DRAINMOD-W) which focused on individual catchments within the US, where the key practical implication of modelling is to simulate catchment processes and management activities, and thus to support local catchment management. In contrast, the statistical models that we developed aimed to predict spatio-temporal variabilities over a large Australian region, which has an area of over 200,000 km$^2$ and more than 100 catchments. The key practical implication was to support higher-level catchment management at a state- or even a national-scale. Due to the different model types, contrasting scales and practical implications between our models and the models reviewed in Moriasi et al. (2007), we are not convinced that the performance standards summarized in Moriasi et al. (2007) are directly transferable to our models. Furthermore, due to the extensive spatial and temporal extents that our models cover and the less focus to support local-scale management activities, it is both more difficult and less necessary for our models to achieve the same performance standards as suggested in Moriasi et al. (2007).*

*We understand the second part of your comment as questioning: 1) whether the 4 poorer models, TSS, TP, FRP and NOx, are capable to make further inference from, specifically on exploring TSS changes to drought; 2) the validity of using long-term mean concentration of TSS to represent temporal variability, when exploring the drought effects on TSS. Our response to each question is as follows:*

1) *We completely agree with you that when a model is not performing well, we should be careful on drawing further inferences. However, we understand such 'further inferences' as to making predictions and/or interpreting parameter values with respect to physical processes. What we presented on the responses of TSS to drought was different to such 'inferences', as this analysis was based on a model validation against three distinct periods which are differently affected by a prolong drought in the region. In this experiment, the focus was not the model performances in an absolute perspective, but instead, the relative performances of different calibration/validation periods. Specifically, Figure 4 focused on how performance deteriorated when calibrating to one sub-period and validating on the other. Similarly, Figure 5 focused on the variation of model performance of the 'full-model' when simulating individual sub-periods of the full data period, so the focus is again on the relative model performance. We believe that exploring these 'changes in model performances' is an informative approach to explore any drought effects especially when the absolute model performances are not optimal.*

2) *We would like to clarify that although this analysis explored changes in water quality across different sub-periods of the full dataset, the focus was any consistent shift across three periods, as opposed to the day-to-day variability of water quality (i.e. as how 'temporal variability' has been defined in our modelling framework). We believe that such cross-period changes can be more clearly summarized with the long-term mean concentrations for each period, as currently presented. As in our responses to Comment #2.1, to better illustrate the model capacity to represent temporal variability, in the revised manuscript we will provide additional results and discussions.*

3. Site bias

3.1 The areas of sites are highly diverse, from 5km2 to 16,000 km2. It's reasonable to expect that these different sized sites may be dominated by different processes, e.g. smaller sites may be constituent supply driven, while larger sites may be transport driven. These differences may be translated to different explanatory variables for these sites. But in the model, these sites

share the same explanatory variables AND model parameters (ie the betas). The implications needs to be discussed, e.g. if there're more sites with large areas, then the model may bias towards representing large catchments, and the explanatory variables selected does not have strong predictively power for smaller catchments, and thus leading to poor model performance.

*This is an excellent concern. However, we identified a major misunderstanding of our models which we would like to clarify – the statement 'in the model, these sites share the same explanatory variables AND model parameters (i.e. the betas)' is incorrect. This is because that our Bayesian hierarchical models do allow parameters for the temporal predictors across catchments to vary depending on catchment characteristics (Equation 6, which we explained with more details in response to your Comment #1.1). Such variations in temporal parameter sets are capable to account for differences in the key water quality processes across catchments e.g. different roles of surface and sub-surface flows on water quality due to different scales of catchment processes.*

*Furthermore, in representing these variations of temporal relationships across catchments in our models, we have already considered catchment area as a potential predictor (see Table S1 in the supplementary materials which lists all 50 potential predictors that we considered). However, catchment area has not been identified as a key predictor for variation in these temporal relationships for any constituent, which indicates the less important role that catchment area has on affecting the temporal variability patterns across space.*

*Our choice of the use of a consistent set of model predictors across all catchments was to ensure that models are able to represent key processes and controls in a large-scale perspective, rather than being dominated by catchment-specific patterns that are difficult to generalize and interpret. For example, if we allow 102 catchments to have different numbers of predictors, it would be extremely difficult to obtain a large-scale understanding on the role of streamflow, as well as to understand how the impacts of streamflow vary across catchments.*

*To resolve this comment, we propose to improve our model description in Section 2.1 (Spatio-temporal modelling framework), to further emphasize the point that the temporal parameters were allowed to vary across space, which considered potentially different key processes and controls for water quality across the diverse catchment conditions in our catchments. We will also provide some examples to explain how the key processes can vary across catchments e.g. contrasting processes for larger and smaller catchments.*

3.2 Data transformation: the authors chose to transform observation data, rather than back-transform modelled data. There are a few issues with transforming observation data: 1) the transformation involves additional parameters (such as lambda, instead of a straight transformation, e.g. logx), thus the "observed" data is in effect a "modelled" data, albeit a simple model. 2) The observation data across sites is transformed using the same parameter value (mean), thus the site bias issue in the comment above also applies. 3) the choice of transformation (log) leads to a decrease in the sensitivity of large values due to the log() function, and increase the sensitivity of small values. Thus, it is unclear to me whether using transformed observation data is any better than back-transforming modelled data. These implications need to be pointed out in the manuscript.

*Transforming data for our modelling was a decision informed by previous phases of the same research project, where we used linear statistical models to identify the key drivers of water quality variability across space and time (Lintern et al., 2018b; Guo et al., 2019). To incorporate the previously obtained*

understanding into this study, we used similar linear model structures – which were calibrated with transformed data to satisfy the assumptions of linear modelling or otherwise the untransformed data would be too skewed to work with (see more details in Figure R1 under point 3)). Since the model was calibrated in a transformed scale, we believe that the transformed scale is also most relevant and informative for performance assessments as we presented. We clarify each specific issue you raised as following:

1) Politely disagree. The log transformation, as referred to as a 'straight' transformation in the comment, is a special case of Box-Cox with lambda=0. Therefore, even a log would still introduce an additional parameter (although = 0), which has fundamentally no difference with a parametric Box-Cox transformation. In a more general sense, parametric transformations (e.g. log, Box-Cox, Log-sinh) have been widely applied and recognized as data pre-processing approaches, instead of a step in the modelling process.

2) Using the same parameter value for transformation across all catchments ensured that the results (performance of the calibrated water quality models for each constituent) are in a consistent scale and are thus comparable across catchments. This is an essential requirement to achieve large-scale spatio-temporal modelling capacity as addressed in this paper. Regarding site bias, we have explained in our response to your Comment #3.1 that our Bayesian hierarchical modelling framework can effectively address the concern of site bias, by allowing variation in the temporal parameters to represent potentially different key processes across catchments.

3) The whole purpose of data transformation was to reduce the impacts of the extremely high values on model calibration. This is because that those high values often present in extremely low proportions within the data – as illustrated in Figure R1 in which untransformed data were plotted against corresponding quantiles, for each constituent. If those extreme values (right tails in each panel in Figure R1) were left untransformed, they may cause the models to emphasize too much on rare extreme events, and thus largely affect our ability to represent the overall large-scale patterns in water quality.

[Figure]

*Figure R 1. Untransformed data against their quantiles for each constituent.*

_To address this comment, we will add more discussions in Section 2.2 (Data collection and processing) to improve the justification of data transformation. We will also add explanations in Section 2.4 (Model performance and sensitivity analyses) on why model performance assessments are presented in a transformed scale._

*While we decided to focus this study on the transformed data, we have noted back-transforming modelled data as a possible option, and we would like to explore the differences between these two approaches in future studies.*

4. Results in drought impacts

Assuming the model is appropriate for inference (i.e. have good enough performance measure), a better (more insightful) way to demonstrate the impact of drought could be to show what the parameters (beta) for pre and post drought models are. This is because (I assume) these parameters represent the system behaviours, i.e. how strong different explanatory variables are to predict concentrations.

*Thanks for sharing the interesting idea. Firstly, we have compared the parameter values for the key spatial and temporal predictors of TSS when the model was calibrated to different periods. The effects of key predictors for spatial variability did not vary much across periods (Figure R2). In contrast, the effects of key predictors for temporal variability showed a clear shift in the role of antecedent flow (prior 7-day flow) across different drought periods (Figure R3). Specifically, the flow effects are mostly positive across catchments before the drought, which shift to mostly negative during the drought; after the drought, the flow effects have mixed directions among different catchments.*

[Figure]

*Figure R 2. Effects of the seven key predictors for the spatial variability in TSS across 102 sites, summarized by the posterior mean of the calibrated parameter values for each predictor, to the pre-, during- and post-drought periods (differentiated by colour). The seven key predictors are, from left: hottest month maximum temperature, percentage catchment area as grassland, percentage catchment area as shrub, percentage catchment area as cropping land, maximum catchment elevation, percentage catchment area made up of valley bottoms, and average soil clay content.*

[Figure]

*Figure R 3. Effects of the five key predictors for the temporal variability in TSS across 102 sites, summarized by the posterior mean of the calibrated parameter values for each predictor (box shows values across all sites), from left: flow, 7-day antecedent flow, water temperature, root-zone soil moisture and deep soil moisture.*

*Despite the interesting results and potential discussions, we note that these results are likely not reliable considering the limited performance of the TSS model. This has been explained in our*

*response to your Comment #2.2, point 1), where we agreed with your comment that we should be careful in making further inferences with a model with limited performance. Therefore, for this drought analysis, we propose to not presenting/discussing further results beyond the existing ones in the manuscript (i.e. relative performance of model over different calibration/validation periods as in current Figures 4 and 5).*

**Other comments**

5.  Pg 17, L374: please explain why "out models are very useful in representing and predicting proportional changes in concentrations"?

*The Box-Cox transformation which our models were developed with is essentially similar to log transformation, which is widely used in water quality to represent proportional differences in linear space. We will improve clarification of this argument during revision.*

6.  Maybe consider putting supplement tables S5 and S6 in to main text as these are important part of the model.

*Agreed. The key model predictors shown in Tables S5 and S6 are important parts of this model, although they have been identified from our two preceding studies (Lintern et al., 2018b; Guo et al., 2019). To address this comment, we will move these tables to the main text. In addition, since the second column of Table S6 (which summarizes the key factors relating to the spatial variability in temporal effects) are new findings in this study, we will provide more interpretations and discussions on these results.*

**Reference**

Moriasi, D. N., Arnold, J. G., Van Liew, M. W., Bingner, R. L., Harmel, R. D., & Veith, T. L. (2007). Model evaluation guidelines for systematic quantification of accuracy in watershed simulations. Transactions of the ASABE, 50(3), 885-900.

---

## Author Comment (AC2) · 16 Oct 2019

**Responses to Reviewer Comments on "A predictive model for spatio-temporal variability in stream water quality" (RC2)**

Our proposed manuscript revisions are underlined.

**Context**

This paper introduces a Bayesian hierarchical model for spatio-temporal prediction of water quality variables in Australia. After model construction and validation, the results are discussed in terms of influences on prediction accuracy and regarding the influence of a long drought period on average suspended sediment concentrations. The paper concludes with recommendations regarding model improvement.

**General comments**

Generally, the paper is well written and the methods and results are interesting. However, I have some major concerns regarding (i) the statements drawn from the results, (ii) influences on the simulation accuracy and (iii) the focus of the study. These major points need to be clarified before publication.

*Thank you very much for your comprehensive review and identification of key areas of improvement. We provide detailed response to your comments in the subsequent sections.*

**Focus of the study**

1. The study is introduced as a new model for water quality prediction. It is mentioned that the construction of the site-specific model was already published in two preceding papers (Lintern et al., 2018b; Guo et al., 2019). It is not really clear which additional information this paper provides. In the discussion section, there is a long chapter about the influence of a long-term drought to TSS concentrations, which was found as a by-product (?) of the study. The papers ends with conclusions suggesting higher-frequency sampling data, which was not analysed in this study at all. Thus, the study lacks a clear focus and coherent conclusions.

*Great points. We acknowledge that the two preceding papers (Lintern et al., 2018b; Guo et al., 2019) focused on identifying the key controls for spatial and temporal variabilities of stream water quality, and understanding the effects of these controls. In contrast, this study presents the integrated model developed based on the previous understanding. Although the model structure was informed by the preceding studies, this study established, for the first time, a spatio-temporal model which is capable to predict across multiple catchments in a regional scale. In addition, in this study we have also developed new understanding on how the temporal drivers of water quality vary spatially, which is a key component of spatio-temporal predictive capacity. To address the comment on the innovations that this study brings, we will revise the Introduction and relevant sections in the Discussion to highlight how this study differs from its preceding works. We will also add more results on how the temporal effects vary spatial as a new understanding obtained from this study.*

*In addition, to improve the linkage between study objectives and results, we propose to present additional results to highlight several model capabilities, with further discussions on how these can benefit catchment management. These include the following potential topics:*

*1) Modelled and observed time-series at selected catchments, to illustrate model capacity to predict and interrogate trends and changes over time;*

2) *Proportions of spatial and temporal variabilities explained for each constituent, to illustrate the importance of the two variability components for each constituent, and how they can be explained by our models;*

3) *Extending the existing model cross-validation from 5 replicates to 50 replicates, to provide a more comprehensive summary on model sensitivity to calibrated dataset.*

*We disagree that the effects of long-term drought on TSS is a by-product of the study, but rather consider it as an illustration of model utility – to identify potential changes in water quality processes associated with major catchment changes. The same approach can also be applied to simulate catchments with regions which experienced significant changes in land use and dam development, etc.. and assess corresponding impacts on water quality. To address this comment on the purpose of analyzing drought effects on TSS, we will revise the Discussion section to make this point clearer.*

*We consider the use of high-frequency monitoring data to be an interesting discussion point, as also reflected in the discussions in Comment #4 from Reviewer 1 and Comment #9 from Reviewer 3 and our responses to those comments. Although we have not analyzed high frequency data in our study (since only a few sites in Victoria have such data), we would like to discuss the potential benefits and additional requirements of using high-frequency data in our modeling framework, and thus to provide useful guidance to future studies. To address this comment on discussing high-frequency sampling data in the conclusion, we will clarify that these are recommendations based on our experiences with the models. We will also thoroughly revise the Conclusion to better align it with the rest of paper.*

**The influence of LOR on simulation accuracy**

2. First of all: What is LOR (Limit of Reporting)? Is it a limit of detection (LOD) or a limit of quantification (LOQ) or something different? Which value was used for the calculation of Nash-Sutcliff (Neff) efficiency if the measurement was below LOR? Zero? Half the LOR? Please clarify.

*Our use of the term 'LOR' actually refers to the concept 'detection limit' as defined in the Victorian Water Quality Monitoring Network and State Biological Monitoring Programme (1999), as:*

- *'minimum concentration detected for which there is 95% confidence of accuracy and therefore is accurate enough to report. Detection limits are based on a minimum of 10 replicates of a sample or standard of low concentration of the analyte, taken through the whole procedure (including digestion if required by the method).'*

*This is different to either of LOD and LOQ, which have been defined as (Armbruster & Pry, 2008):*

- *LOD: 'the lowest analyte concentration likely to be reliably distinguished from the LoB and at which detection is feasible. LoB is the highest apparent analyte concentration expected to be found when replicates of a blank sample containing no analyte are tested.'*
- *LOQ: 'the lowest concentration at which the analyte can not only be reliably detected but at which some predefined goals for bias and imprecision are met.'*

*Regarding the second part of your question, for the calculation of NSE, when the measurement was below LOR (below LOR values were used only for model evaluation in Section 3.1), the value of half of LOR was used.*

*To minimize confusion and keep consistency with our monitoring dataset, we will replace the term 'LOR' with 'detection limit' in the revised manuscript. We will clarify the definition of 'detection limit'*

*where this term is first introduced (L153). We will also add clarification on how the data below detection limit were used when describing the relevant model performance assessments (L213).*

- *Australian Water Technologies: Victorian Water Quality Monitoring Network and State Biological Monitoring Programme: Manual of Procedures, 1999.*
- *Armbruster, D. A., and Pry, T.: Limit of blank, limit of detection and limit of quantitation, Clin Biochem Rev, 29 Suppl 1, S49-S52, 2008.*

3. For model construction, the values below LOR were excluded due to statistical reasons and due to the fact that these low concentrations were of less interest. Thus, why were the values below LOR included in model validation at all? Please clarify.

*The only place which we considered below-LOR data in model evaluation was to understand the ability of this model (which was calibrated to truncated data) to simulate the full distribution of observations (as justified in Section 2.4), which was not a validation strictly speaking (where independent dataset should be used). Due to the exclusion of below-LOR data for our model calibration, readers may question how much the model performance would be affected by including the below-LOR data. If inclusion of the below-LOR data leads to a good fit, then the models calibrated to above-LOR data is transferable to below-LOR data too.*

*To address your comment, we will add these discussions to Section 2.4 to better highlight the purpose of this specific model performance evaluation and the fact that subsequent evaluations excluded the below-LOR data.*

*We acknowledge that all model cross-validations (shown in Section 3.2) were performed without below-LOR data.*

4. Later on it is analysed that the fraction of LOR on total measurement values influences model performance, especially the P fractions and TSS. The discussed reasons are mainly methodical/statistical. I think, the effect of LOR on model performance might also be a secondary effect: the parameters with a high proportion of LOR are mainly those with the highest natural concentration variability, since their concentration peaks are event-driven. Thus, monthly grab samples might capture peaks or not. Since some of the catchments are as small as a few km², even the specific time of a day might influence the sampled concentration to a large extent. Thus, the probability of sampling low between-event concentrations is higher for P and TSS than for e.g. Nitrate. Therefore, the low model performance might rather be an effect of the overall lower information content of the samples, which results in models which are based on a lower information content. What do you think?

*Thank you for sharing this very interesting point. We understand that you suggest another possible explanation for the influences of high proportions of below-LOR samples on our model performance, that is, constituents with large number of below-LOR samples are often also driven by high streamflow events, which are otherwise insufficiently captured by the monthly monitoring data.*

*To check whether the high LOR issue is more observed in event-driven constituents we referred to Figure 5 in Guo et al. (2019) (included below), in which the impacts of flow on concentrations across all 102 sites within same dataset were summarized by the 'Q same day' boxes for each constituent (i.e. panel). We focus on the two constituents that are most affected by the high LOR issue (TSS and FRP), and found that TSS is highly event-driven while FRP is relatively less influenced by flow, compared with other constituents. These suggests that the high LOR issue might be related to lack of*

*sampling high flow events for TSS, but there is no systematic pattern across constituents. We will add some discussions on this point when highlighting the high LOR issue for TSS.*

[Figure]

*Figure R 1. Figure 5 in Guo et al. (2019): Effects of hydro-climatic predictors on the temporal variability of each constituent across 102 sites, summarized by the posterior mean of the calibrated parameter values for each predictor. Y-axis shows the effect as the number of standard deviations away from site mean. Note that only predictors identified for at least one constituent during the selection of predictors are shown on the X-axis.*

**Influences of drought on TSS**

5.  During the modelling process, the authors note, that a long-term drought influenced TSS concentration, which is a really interesting observation. However, I do not understand why a model is required for this analysis. Wouldn't simple statistics (such a t-test or Mann-Whitney-U-Test) have done the same job? I don't see that this is a special result of this model application.

*Firstly, the main purpose of the paper is to develop the modelling framework, not examine hypotheses about drought. We use this analysis as an illustration of our model capacity. We agree that simple statistics would indicate trends/changes over time, but the interpretation is limited to only changes in concentrations without further indication on potential causes. Specifically, with simple trend statistics we would be able to identify changes of TSS concentrations during the drought, but not able to tell whether such changes are due to decrease in streamflow or other more complex processes. In contrast, using the models developed in this study, we were not only able to*

*identify changes in TSS concentrations, but also able to suggest that these systematic changes are not due to changes in any of the key controls of sediments (e.g. streamflow) since drought, but instead, related to a shift in the relationships between sediment concentrations and its key controls (e.g. streamflow) during different periods – this reveals much more understanding compared with simple trend statistics.*

*To clarify this, we will add brief discussions in Section 4.3 (Potential impacts of long-term drought on water quality dynamics) to compare and contrast our analysis to simple trend analyses, and to highlight the additional understanding obtained through our approach.*

**Meaning of factors**

6. Since the model is a (multidimensional) statistical model, the explaining variables (factors) not necessarily contain process-based meaning for the target water quality parameters. For example, the water temperature is an explaining variable for temporal variability of TSS (Table S6), which is not really clear to me. In L.15-17 it is stated that the paper addresses the key controls (factors) explaining water quality variability, but an in-depth analysis and discussion is missing in the text. I would encourage the authors to even discuss the factors in more detail or to change the focus of the paper.

*As in our response to your Comment #1, we acknowledge that the focus of this paper is not to identify the key controls for spatial and temporal variabilities of stream water quality and understand their effects – which have been addressed in the two preceding companion papers (Lintern et al., 2018b; Guo et al., 2019). The effects of key controls on water quality have been presented in detail and discussed extensively in these two preceding papers and are therefore not repeated in this study.*

*To avoid the confusion which this comment reflects, we will revise the Introduction to better clarify the focus of this study and how this study differs with the preceding ones, along with revision on other parts of the manuscripts.*

*As also mentioned in response to your Comment #1, this study developed new understanding on how the temporal drivers of water quality vary spatially, which is a key component of spatio-temporal predictive capacity. To highlight this, we will add new results and discussions on the key factors relating to the spatial variability in temporal effects.*

*Regarding the example that you mentioned (the mechanisms via which water temperature influences the temporal variability of TSS), we have provided extensive discussion on possible mechanisms in Guo et al., 2019, along with those for other key temporal drivers identified. In short, the strong impacts of water temperature may be due to the high correlations between water temperature and air temperature; while higher air temperature (warmer periods) can be further associated with enhanced source and mobilization processes, which potentially lead to: (1) greater soil desiccation and soil erodibility, (2) more intense agricultural activities that occur during warmer periods such as tillage, or (3) lower plant canopy cover in drier and warmer months. Due to the different study focuses, these are not further discussed in this study.*

**Specific comments**

7. 1-2: The title "A predictive model for spatio-temporal variability in stream water quality "suggests a generic model for different sites and different water quality parameters. However, the described model is very site-specific. Thus, I would suggest to change the title to a more site-specific one, probably including the region or similar, including the applied method.

*We would like to clarify that the models developed in this study are not completely site-specific, but were integrated space-time models that are capable to predict across 102 sites over a 200,000km$^2$ region at once. The model structures were informed by previously obtained understanding on both the catchment- and regional-scale water quality variability and their key controls from the two preceding companion papers (Lintern et al., 2018; Guo et al., 2019). This data-driven modeling framework is transferable to any other parts of the world.*

*Adding locations or study region to the paper title is likely causing misunderstanding that this paper describes a case study of existing modelling approach, which would in turn greatly hamper the communication of key contributions of this study. Therefore, we politely disagree with the reviewer on adding study locations to the paper title. However, to improve clarity, we propose to add the phrase 'data-based' in our title to suggest that an empirical model is introduced:*

*'A data-based predictive model for spatio-temporal variability in stream water quality'*

8. 71: Change "…quality can not…" to "…quality not…"

*We will implement this change as suggested.*

9. 76: Change "…model built…" to "… model was built…"

*We will implement this change as suggested.*

10. 76-78. It is stated that the model was constructed and published in two previous papers. Please elaborate on the additional information this paper provides.

*As explained in our responses to your Comments #1 and #6, this paper presents the first spatio-temporal model developed over a large geographical region across multiple catchments. We will clarify this better in the Introduction. In addition, we will also adjust the earlier parts of the Introduction to focus more on the knowledge gap relevant to this study (i.e. developing spatio-temporal predictive capacity), instead of those that are relevant to the preceding studies (obtaining new understanding).*

*As also highlighted previously, this study obtained new understanding on how the key controls of temporal variability of water quality vary spatially, and thus developed spatio-temporal predictive capacity where the two preceding papers have not achieved. New results and discussions on the spatial variability in temporal effects will be added to support the new findings.*

11. 79: It is stated, that this study aims at bridging the gap between fully distributed and statistical models. Well, what is this model if not a statistical model? Probably, it was meant to bridge the gap between fully/semi-distributed and lumped models.

*Thank you. We meant to say that the model bridges the gap between fully-distributed physically based models (which are driven by equations representing physical processes e.g. SWAT) and data-driven statistical models (which are fully relying on observations e.g. black-box ANN type models). We will adjust the phrase here during revision.*

12. 154-156. During the Box-Cox transformation of the data, the high sampling values lose their significance, especially for goodness-of-fit calculations. This effect can be seen after back-transformation (figure S13), which results in low Neff values. Thus, how is the statement "poor water quality conditions…were our primary concerns…" compatible to the fact that the data was transformed?

*By saying 'poor water quality conditions (i.e., high constituent concentrations) were our primary concerns to model' (L155), we are referring to higher concentrations with respect to the below-LOR data. Our modelling was not specifically focusing on the representing the extremely high values, but rather focusing on the large-scale patterns of water quality. Such modelling focus required data to be transformed so that the modelling was not overly sensitive to extreme values.*

13. 159. Insert a blank between "as each"

*We will implement this change as suggested.*

14. 186. "… via a Spearman correlation analysis" (note the typo "analyses"). Please add the correlation coefficients and the p-values in the supplement.

*We will correct the typo and add the Spearman correlation results in the SI, as suggested.*

15. 246. "…in Sect. 4.2." Isn't it section 4.1?

*Thank you for identifying this, we will make the correction.*

16. 265. Fix "… is also show…"

*We will correct this as suggested.*

17. 414. Fix "For examples, …"

*We will correct this as suggested.*

18. 418 Fix "adjscent"

*We will correct this typo.*

19. 449-451. In the beginning, this paper aims at introducing a model. In this lines, the reader has the impression, that the main aim of this paper is the analysis of drought on TSS concentrations. Please think about the focus of the paper.

*As in our responses to your Comments #1, #6 and #10, we will revise the Introduction and relevant sections in the Discussion to better highlight the key study objective. We will also update the Conclusion accordingly to ensure that the paper is coherent and focused.*

20. 466-469: "1) collection … in the model". These are not a results/conclusions of this study. Data frequency was not evaluated in this study.

*This sentence intends to summarize the key areas of improvement for this modelling framework which have been identified in the Discussion section instead of study results – as seen in the phrase 'to further enhance the performance of the current models, we recommend that future… (L465)'. We have provided summaries of the key results in previous sections of the Conclusion (L453-365).*

21. 469-470. "These improvements will be very helpful…" How?

*The models that we developed are very useful to provide insights on the overall patterns of water quality variation and potential key controls of these variation, and thus inform the development of mitigation strategies. Therefore, our models are likely more beneficial to support mid- to long-term management, planning and policy making. Our model capacity is likely enhanced by increasing availability of high-frequency monitoring data, since they are likely providing better representation of the temporal variability. However, these data might also have extremely high variability e.g. due to unknown point sources and measurement noises, which brings new challenges for the statistical*

*modelling framework. Considering these, we have decided to revise this recommendation as an open question on the opportunities and challenges that our modelling framework will face when presented with more high-frequency monitoring data. We will also revise relevant sections in the Discussion accordingly.*

---

## Author Comment (AC3) · 16 Oct 2019

**Responses to Reviewer Comments on "A predictive model for spatio-temporal variability in stream water quality" (RC3)**

Our proposed manuscript revisions are underlined.

**General comments:**

1. The study describes a Bayesian statistical model of selected water quality variables in 102 catchments. The model successfully described both the spatial and temporal variability of certain variables, and performed quite well at describing the site-specific means for all variables. Based on the results, the model can serve as a valuable prediction tool in the calibration region (and potentially adapted elsewhere too).

   The main issue with the manuscript is that the otherwise valuable work is presented in an unsuitable (and constantly evolving) context. The title appropriately focuses on the main element of the study, the model and emphasised predictions as the primary field of utilisation. In the Abstract the motivation for the study is summarised as: "To address this [knowledge gap compromising present water quality models], we developed a Bayesian hierarchical statistical model to analyse the spatio-temporal variability in stream water quality across the state of Victoria, Australia." This shifts from predictions to analysis and promises that the model will cover knowledge gaps presumably by revealing so far unknown relations between water quality and its drivers. Interestingly, this objective is not featured in the Introduction. There it reads: "Our approach aims to bridge the gap between fully-distributed water quality models and statistical approaches to provide useful information for catchment managers, especially for largescale water quality assessments." This alters the context again, now the model is meant to be a "missing link" between very detailed (deterministic) models and simple statistical tools and the reason is to serve catchment managers. These context shifts do not help to assess the values of the study and generate expectations that are fulfilled later.

*Thank you very much for your comprehensive review and contribution of valuable ideas. We would like to clarify that:*

1) *'Revealing unknown relations between water quality and its drivers' has been covered in our previous two papers. Specifically, Lintern et al. (2018b) investigated the key catchment characteristics that are related to spatial variability of catchment water quality; Guo et al. (2019) investigated the key controls for temporal variability of water quality at each catchment.*
2) *The core objective of this study is to develop statistical models that can predict spatial and temporal variabilities of catchment water quality. In achieving this spatio-temporal predictive capacity, we have developed new understanding on how the temporal drivers of water quality vary spatially, which has not been explored in the two preceding studies.*
3) *The main model objective is to improve the predictive power for these variabilities and thus allow catchment managers to better plan/manage water quality changes across both space and time.*

*We will thoroughly revise the Introduction (and other parts accordingly) to ensure that the main study objective, research questions and study implications are clearly communicated, and that individual results are mapped to the corresponding research questions. We will also revise the abstract to better capture the full story. In addition, to support point 2) above, we propose to include more results and discussions on how temporal relationships vary spatially.*

2. Unfortunately, none of the above alternative contexts is completely followed in the Results and Discussion. The Results consist almost exclusively of performance indicators calculated and plotted in transformed scale. The Discussion focuses on the effects of the drought period on model performance and future development directions without mentioning potential major obstacles and pitfalls (gathering more detailed data and developing more detailed models is an idealistic recipe). The manuscript would greatly benefit from following a clearly defined logical structure, objectives and featuring topics that are truly relevant for the work. Performance indicators should not occupy all the Results section. There is much more to show about the model, especially considering the ideas that show up in the present Introduction and Abstract. The potential topics include:

*While confirming that our core study objective is to develop statistical models that are capable to predict spatial and temporal variabilities of catchment water quality, we agree that there are more dimensions to present/discuss on the capability of the models and their practical implications for catchment managers. We appreciate your suggestions on potential topics and we provide responses to individual ones as following:*

2.1 Untransformed comparison of measured and modelled time series for selected catchments

*Investigating model performance with time series is an excellent idea. We propose to investigate the model capacities to capture trends and changes within water quality time-series, together with discussions on their relevance to catchment managers.*

*We would like to focus model evaluation at the transformed scale, since this is what the model was calibrated for, and would thus provide the most informative assessment of model performance (as also explained in more details in our response to your Comment #43). We will add these to Section 4.1 to justify the use of transformed data for model evaluation.*

2.2 The needs of catchment managers with respect to predictions and how this model fulfils them (If it does so. If not, management should not be emphasised so much).

*We believe that our proposed investigations to address your Comment #2.1 (model capacities to capture water quality trends) is a good example of the management utility of our models, because water quality trends and changes are of great interests for catchment management. Our models linked water quality variations over time with the hydro-climatic conditions and vegetation cover of catchments (i.e. controlling temporal variabilities) which means that the model would be able to capture some trends/changes that are 'expected' (e.g. due to droughts or major changes in vegetation cover); on the other hand, catchment managers can use these models to identify 'unexpected' trends (e.g. due to additional discharge points and major land use changes which are not included in our model), and prompt further investigation. These insights are very beneficial to catchment management particularly in regional/national scales, as well as long-term catchment planning and policy making. We will strengthen relevant discussions on the existing results and the proposed additional results (as in response to Comments #2.1, 2.3, 2.4 and 2.6) to better highlight the management implications of our models.*

2.3 Key controls and mechanisms governing water quality. What do we learn from this study compared to Guo et al. 2019 and Lintern et al 2018a, 2018b?

*As highlighted in our response to your Comment #1, this study does not focus on identifying key controls for spatio-temporal variabilities in water quality, as these have already been extensively*

*analyzed and discussed in our previous companion studies (Guo et al. 2019 and Lintern et al 2018b).* *To address this comment, we will revise the Introduction thoroughly to better clarify this different focus of this study with the two preceding studies.*

*In addition, we note here that understanding and predicting differences in the influence of controls on temporal variability between catchments, as discussed in response to your Comment #1, is a unique contribution of this paper compared with the two preceding papers. This will be better clarified in the revised manuscript.*

**2.4 The grade of intrinsic randomness (and its compatibility with management), predictability of water quality variables.**

*We interpret the comment as requesting assessment of natural variability in water quality that is explainable through deterministic predictors. Unexplained variability has been summarized in our model performance evaluation (Section 3.1).* *To address this comment, we will strengthen the discussion on the potential components of unexplained variability that we have already presented, and further explore the percentages of spatial and temporal variabilities that remain unexplained. These additional results and discussions will better illustrate the model ability to explain water quality changes and the relevant limitations.* *These results are informative to catchment management, because they highlight how much variability is explainable over space and time, and thus, the modelling error we expect when predicting long-term conditions across space, and when predicting over time at individual sites.* *We will add relevant discussions to highlight the management implications.*

**2.5 Model limitations: implicit assumptions, conditionality on the calibration set and the present layout of calibration units (what would happen if the model was calibrated on merged catchments?)**

*In section Section 4.1, we have presented extensive discussions on the key model limitations due to 1) the linear statistical model structure, 2) the impacts of below-LOR records and 3) data transformation. Furthermore we have also discussed model limitations due to lack of representation of of biogeochemical processes.*

*We agree that calibration dataset and model transferability are typical limitations of statistical models,* *which we will add to the above Discussion.* *On a related note, we also propose to increase the number of cross-validation replicates from the current 5 to 50. We believe that this would give us a more comprehensive summary of model limitation to calibration dataset.*

*We are unclear on the interpretation of your comment 'what would happen if the model was calibrated on merged catchments' and thus provide two possible interpretations here.*

1) *If your comment refers to the differences between a model calibration to individual catchments versus one using all catchments merged as a single one, we are unsure about value of modelling on merged catchments. This is because that we have already used a joint model calibration across all 102 catchments (instead of site-specific calibration) for our Bayesian hierarchical models. The key benefit of the Bayesian hierarchical modelling structure that we applied is its capacity to include varying temporal relationships across catchments, which we identified as a critical consideration when exploring temporal variability of water quality in large regions (as seen in Guo et al. 2019). In contrast, modelling on merged catchments is unable to represent how temporal variability differs across catchments.* *We believe that this specific comment on the calibration on*

*merged catchments can be resolved by improving the description and justification of the Bayesian hierarchical modelling approach in the relevant Method section (Section 2.1).*

2) *If your comment is referring to the impact of nested catchments in our models, we would like to clarify that most of the 102 catchments that we used in this study were independent (as seen in Figure 1 in the manuscript). Therefore, our dataset is not suitable to answer this question.*

> 2.6 Spatial and temporal distributions of validation errors, their relationship with model development alternatives.

*We are currently running 50 replicates of cross-validation for the model developed for each constituent. Once completed we will investigate the spatial and temporal distribution of errors from this more comprehensive validation and update the results section accordingly.*

**Specific comments:**

3. Lines 15-16: In my opinion it is not the lack of understanding, but the lack of information. The effects of many key controls on water quality are well understood, albeit in an isolated, idealised context. It is clear, for example what certain polluting sources (like a WWTP effluent, a plot of arable land, etc.) do, how different landcover types affect the transport of pollutants along a specified pathway. The problem with the modelling of stream water quality on the (sub)catchment scale is that numerous key factors and controls act together and in practice there is no hope to get relevant information on all/most of them. That's why detailed and dynamic models fail on all components except those that behave quite simply and are not affected by too many factors. The challenge of modelling is to include the relevant factors AND the necessary information about them. So I would rephrase the sentence to mention that despite the long history of research there are too many key controls and very high complexity in both space and time compared to the available information.

*Thank you for the thoughts. We agree that lack of information is a critical issue, because reliable information is the basis for gaining new understanding and/or validating existing understanding. However, we also cannot ignore important limitations in the current understanding of water quality behavior across multiple catchments in large regions, which agrees with what you have summarized – 'The effects of many key controls on water quality are well understood, albeit in an isolated, idealised context'. Certainly, at some catchments we have much better understanding of locally specific water quality mechanisms, which is supported by detailed data and local knowledge (i.e. information). However, this understanding is limited in transferability to other catchments as well as to inform the development of water quality models in other catchments. In addition, current understanding also tends to be focused on characteristics such as land use rather than natural catchment characteristics. These limitations are especially important when the interest is in a large geographical region across multiple catchments. For example, from conceptual understanding we would expect surface flow to enhance transport of sediments, but we have not well understood: a) the relative importance of surface flow effects compared with other key factors of water quality e.g. sub-surface flow and other climatic conditions etc., and how all the key controls interact with each other; and b) the varying extents to which surface flow influences sediment concentration between catchments.*

*As in the above example, developing such large-scale water quality models across catchments involves the identification of the key explanatory variables at larger scales, which would be ideally developed from more extensive information, but often only limited data exist. Therefore, a key innovation of our two preceding studies is to sift through many potential explanatory variables that we have from conceptual understanding to identify the more important ones for building a parsimonious predictive model at large scales (Lintern et al., 2018b; Guo et al., 2019). As a step forward, this study illustrated*

*good ability to represent spatio-temporal variability in water quality can be achieved based on understanding developed with limited information, at a regional scale of over 200,000 km² and across more than 100 catchments.*

*Therefore, we suggest that lack of information and lack of understanding should both be discussed as the key limitations to modelling catchment water quality, especially at a regional scale and across catchments (as the focus of this study).* *To address this comment, we propose to add more elaboration in the Introduction on the tradeoff between having good understanding and at a large scale, as the two critical requirements for modelling water quality at a regional scale. We believe that these revisions will help to clarify the knowledge gap that we address i.e. the need for better modelling capacity at large scales.*

4. Line 16: Even if there would be a lack of understanding (which I doubt, see previous comment), how would this issue be addressed by a Bayesian statistical model? Statistical models build on covariance instead of causal relations and therefore are rarely suitable for modelling conditions that are different from the calibration dataset in any significant aspect âAˇT which is the primary objective of most modelling exercises.

*Firstly, we believe there is a lack of both information and understanding, as explained in our response to your Comment #3.*

*We completely agree with you that our modelling approach does not improve our understanding of causality at all, but it still allows us to make better predictions, which is the aim of the paper as we clarified in our response to your Comment #1. Bayesian hierarchical approach enables us to build better empirical models that allow for differences in parameter relationships to exist for individual catchments. This is a key advantage for modelling over large geographical regions across multiple catchments which physically-based models struggle to achieve.*

*We believe that our proposed updates in the Introduction in response to Comments #1 and #3 would better clarify the key study objectives and provide more evidence to support the knowledge gaps, specifically via:*

- *Clarifying the key study objective as to develop statistical models that can predict spatial and temporal variabilities of catchment water quality (Re Comments #1).*
- *Providing more discussion on the tradeoff between having good understanding and at a large scale, as the two critical requirements for modelling water quality in a regional scale (Re Comments #3).*

*These changes which will also provide better justification for applying the Bayesian hierarchical model, which helps to address this comment.*

*Regarding your last comment on modelling different conditions, we believe that it is challenging for all fitted models (including calibrated process-based models) to predict well for conditions that are different from the calibration dataset.*

5. Line 20: Please mention how FRP relates to the more commonly known Soluble Reactive Phosphorus (SRP).

*FRP (Filterable Reactive Phosphorus) is defined as 'Reactive Phosphorus for a filtered sample to a defined filter size (e.g. RP(<0.45 μm))', which is equivalent to SRP (Soluble Reactive Phosphorus) when the same filter size is referred to (Jarvie et al., 2002).*

- *Jarvie, H. P., Withers, J., and Neal, C.: Review of robust measurement of phosphorus in river water: sampling, storage, fractionation and sensitivity, Hydrology and Earth System Sciences, 6, 113-131, 2002.*

*We use the term FRP following the terminology being used in the overall research project which this study belongs to. It is also the terminology used in the water quality database which we extracted the study datasets from (i.e. Victoria Water Measurement Information System, available at: http://data.water.vic.gov.au/; the sampling method and terminology definition for this dataset are documented in the Victorian Water Quality Monitoring Network and State Biological Monitoring Programme (1999):*

- *Australian Water Technologies: Victorian Water Quality Monitoring Network and State Biological Monitoring Programme: Manual of Procedures, 1999.*

*To avoid confusion, we will clarify the naming convention of FRP along with the more commonly used terminology in literature (SRP), when FRP is first introduced in the manuscript (L121).*

6. Line 21: The abbreviation of "NOx" is not the best choice, as this is a widely known name of the air pollutant group of gaseous nitrogen oxides. Why not "NOi" or something else?

*NOx refers to nitrate-nitrite ($NO_3^- + NO_2^-$) in our study, and this definition has been widely used in water quality research, e.g.,:*

- *Bunn, S., Abal, E., Smith, M., Choy, S., Fellows, C., Harch, B., Kennard, M., and Sheldon, F.: Integration of science and monitoring of river ecosystem health to guide investments in catchment protection and rehabilitation, Freshwater Biology, 55, 223-240, 2010.*
- *Eyre, B. D., and Pepperell, P.: A spatially intensive approach to water quality monitoring in the Rous River catchment, NSW, Australia, Journal of Environmental Management, 56, 97-118, https://doi.org/10.1006/jema.1999.0268, 1999.*
- *Bruland, G. L., Hanchey, M. F., and Richardson, C. J.: Effects of agriculture and wetland restoration on hydrology, soils, and water quality of a Carolina bay complex, Wetlands Ecology and Management, 11, 141-156, 2003.*

*We prefer to keep the term NOx to maintaining consistency with the overall research project and related papers. NOx is the terminology that has been used in the water quality database which we extracted the study datasets from (i.e. Victoria Water Measurement Information System, available at: http://data.water.vic.gov.au/, the terminology and relevant definitions are provided in Victorian Water Quality Monitoring Network and State Biological Monitoring Programme (1999):*

- *Australian Water Technologies: Victorian Water Quality Monitoring Network and State Biological Monitoring Programme: Manual of Procedures, 1999.*

7. Lines 21-22: Yes, the model described variation, but above an improvement of understanding is promised.

*As explained in response to your Comments #1 and #2.3, improving understanding is not the key focus of this study, but instead, we focused on developing models to predict spatial and temporal variabilities in stream water quality. We will work throughout the abstract and the manuscript (mainly the Introduction) to improve clarity of the study objective.*

8. Lines 29-30: How would a statistical model include those mechanisms that govern non-conservative constituents? Such a development would indeed be a major step forward, but it is definitely not trivial.

*In a statistical modelling framework, this could be achieved by considering additional predictors that are related to the key processes that affect the non-conservative constituents and biogeochemical processes (e.g. DO, channel habitat condition, microbial activity in soils etc.) without major changes of the model structure. Another option is to use non-linear structures that attempt to characterize the processes more directly. This sentence within the abstract intends to provide only a brief introduction of potential model improvements. To address this comment, we will add more details to the relevant discussions in Section 4.1 (Implications for statistical water quality modelling).*

9. Line 32: High frequency data often reveal phenomena that are typically not parts of models and therefore model performance further declines.

*Great point. High-frequency data can be helpful, but only to the point where they do not require much more complicated model structures to account for the fine scale temporal structure, otherwise these higher frequency data will contain temporal variation patterns that are not explainable by the driving data that we have. To avoid confusion, we will delete this statement from the abstract. We will elaborate more on the utility of high-frequency data in our modelling framework in Section 4.2, and provide more comprehensive discussions on the benefit/loss that we can get from using high-frequency data with the current model structure.*

10. Line 33: Besides the classical landuse, agricultural activities (ploughing, fertiliser/pesticide application, livestock handling practices, etc.) would need to be known too.

*This is an excellent point, which we are also planning to include in future model improvements. However, considering landuse and land management activities at the large-scale that we modelled for requires an extensive amount of good quality datasets that are currently not available. We expect such lack of information to be improved with novel data collection and/or systematic interviewing approaches in the future. To address this comment, we will include land management with some brief examples in this sentence in the abstract and expanding relevant discussions in Section 4.2 (Implications for water quality monitoring programs).*

11. Lines 40-42: Unpredictable variability does not preclude management. Robust measures can address issues without having to predict the full dynamics. It is well known that the elimination of pollution sources and artificial hydrological factors improves water quality. If the statement in lines 40-42 was true, water quality management would not exist yet.

*This is good point that practical management decisions are often made with low predictive capacity. However, here we were not aiming to criticize such management practices, but to highlight how management would benefit from better understanding and prediction of variabilities. The fact that we are able to manage water quality with limited prediction capacity does not suggest that improving modelling capacities is an unnecessary effort.*

*To better clarify this, we propose to revise the sentence:*

*From: 'However, our ability to manage and mitigate water quality impacts is hampered by the variability in water quality both across space and time, and our inability to predict this variability'*

*To: 'Effective management and mitigation of water quality are key to reduce these impacts. High variability in water quality both across space and time challenges assessment of management options.*

*Improving modelling frameworks to help predict and interpret this variability provides better tools for such assessments.'*

12. Lines 42-46: This is a bit lengthy description of the high variability in both space and time. Please consider compressing.

*We agree that this is a long description that might not be necessary for experts in this field. However, considering the broad readership of HESS, we believe that it is necessary to provide all these details. These are particularly helpful for the readers to learn the background and to understanding reasoning behind the spatio-temporal structure that we used to model water quality.*

13. Lines 46-51: Briefly, there are allochthonous and autochthonous emissions and both are subject to transport. Please consider compressing.

*We will condense this while mentioning the three key processes, which we consider as important background information for the broad readership of HESS as a multi-disciplinary journal (as also explained in our last response).*

14. Lines 55-59: This listing is somewhat odd. Emission dynamics are completely missing, others are a bit over-detailed and supported with arbitrary references (is the importance of temperature only known since Robert and Mulholland, 2007?).

*Thank you, we will improve the emphasis on emission dynamics, specifically on water quality variation due to changes in land use and land management etc. We will also reduce the discussion on hydro-climatic factors and support it with more appropriate references.*

15. Lines 60-62: This sentence contradicts the abstract statement (lines 15-16). Water quality modeling faces high epistemic uncertainty, unpredictable variability stems rather from an information gap than the lack of understanding. And what do you mean here by "larger scales"? And please include why effective policy and mitigation need information on variability.

*We will re-phrase the sentence to highlight that current understandings remain largely at a conceptual level and/or are specific to a catchment, as explained in our response to your Comment #3.*

*Modelling capacity at 'larger scales' refers to the ability to model across multiple catchments over large geographical regions. We will better clarify this by highlighting 'multiple catchments' in this sentence.*

*Better ability to predict variability in large scales would inform policy and mitigation via multiple pathways, such as: a) informing hot-spots; b) identifying trends/changes in water quality and attribute them to potential causes; c) identifying unexplained variability and thus potential future improvements needed in monitoring and modelling. We will include these discussions to illustrate how management can benefit from better abilities to model water quality variability.*

16. Lines 66-69: It would be worth to mention that most statistical models have weak explanatory and predictive power and therefore it is difficult to use them for designing management interventions.

*Thank you, we will include this in building up the knowledge gap.*

17. Lines 71-72: Please check and fix this sentence, by e.g. deleting "can" or any other way.

*We will delete 'can'.*

18. Lines 74-80: After mentioning management so many times above, one would expect a brief summary about the requirements of managers against water quality models plus a sentence in the objectives on how the current model would fulfil these.

*Good suggestion. We will thoroughly revise the manuscript and make sure that the key requirements for the model to benefit management are mentioned in the Introduction, and further strengthened by relevant sections in the Results and Discussion.*

19. Line 103: Please fix "Beyesian".

*This will be corrected during revision.*

20. Line 112: Please delete "however". Either you describe data processing or not. The present formulation suggest that you don't want to describe it, but later âAˇT reluctantly ˇ âAˇT still do so. ˇ

*We will delete 'however'.*

21. Line 132: Please briefly mention the forms and indicators of landuse considered among the drivers, because these are non-trivial.

*As stated in L136, details of all potential predictors are provided in Table S1 in the supplementary materials. There are 50 potential predictors that we included in the predictor selection process, so they are not individually introduced in the main text. In addition, as understanding water quality spatial variability are not key focuses of this study (as in our responses to your Comment #1) we would keep the descriptions of the relevant approach brief, more details have been presented in Lintern et al. (2019b)*

22. Line 143: You mean "area-specific streamflow"? Streamflow also has the unit of volume/time.

*Yes, streamflow has the unit of volume/time. We will replace the original phrase 'streamflow (mm d$^{-1}$)' here with 'catchment-average runoff (mm d$^{-1}$)' to avoid confusion.*

23. Lines 144-149: How did you convert 2D climatic data to soil moisture? This must have included a complete soil hydrological model, but no hints are given in the main text.

*The 2D climate dataset was provided by the AWRA project by the Australian Bureau of Meteorology (Frost et al., 2016). It included the average percentage volumetric water contents for the root zone (at 1m depth) and the deep zone (deeper than 1m). We will add these details to better clarify the data information during revision.*

- *Frost, A. J., Ramchurn, A., and Smith, A.: The bureau's operational AWRA landscape (AWRA-L) Model, Bureau of Meteorology, 2016.*

24. Lines 156-157: Low flow days often mean the periods of concern with regard to water quality. What was the case here?

*Please note what we removed were not low-flow days, but days with zero (no) flow – during which it was impossible to take water quality samples. This is clearly communicated in L156: 'Water quality records corresponding to days with zero flows were also excluded from further analyses'.*

25. Lines 162-166: This means that you conditioned the transformation on the dataset. Since the predictive nature of the model is emphasised, please explain the procedure of including new catchments. What to do when the new data suggest a different transformation parameter?

*To model new catchments within the study region, we would expect that they follow the same statistical relationships as reflected in our models and thus the transformation parameters (along with other model parameters) to remain the same. However, we still recommend assessment of the statistical properties of the new input datasets (i.e. the key factors controlling spatial and temporal variabilities) and the water quality datasets. The calibrated model can only be applied directly if the statistical properties of the new dataset are similar to those of the calibration dataset.*

*For new catchments out of the regions, we do not recommend direct application of the calibrated models (including parameter values), since they would best represent the key water quality controls only for the calibrated region. It would be possible to apply this modelling approach in a new region to inform water quality prediction, which however, requires extensive selection of key predictors and model calibration, as what we have addressed with this study and the two preceding ones (Lintern et al., 2018b; Guo et al., 2019).*

*For either cases, if the new data suggest a transformation parameter that is substantially different to that in our model, then we recommend re-calibration of the model.*

*We will discuss these more in Section 4.1 regarding future applications of this modelling framework.*

26. Lines 172-174: A random forest approach could have been an alternative for the selection process.

*This is true, but it is a choice of approach, rather than the only approach. The predictor selection processes were developed in our two previous companion studies (Lintern et al., 2018b; Guo et al., 2019), from which the key spatial and temporal controls were already selected. These models presented in this study were developed using those key controls previously identified, without any additional predictor selection processes. We also believe that this comment would be addressed by our proposed revision to address your Comment #1 and #2.3, which involve a through revision of the Introduction to help better clarifying the focus of this study.*

27. Lines 179-183: Aren't these results? Since management is emphasised in the introduction, how would you reflect on the final set of key factors? Climate is close to impossible to manipulate, temperature, soil moisture and streamflow are difficult. Why no direct human factors other than landuse?

*These are not results from this study but instead, from our two previous companion studies (Lintern et al., 2018b; Guo et al., 2019). As explained in response to your Comments #1 and #2.3, the two companion studies focused on identifying key factors that influence spatial and temporal variabilities in stream water quality, whereas this study focuses on developing models to predict spatial and temporal variabilities in stream water quality. We have had extensive discussions in the two companion studies on each key control identified and the potential implications for catchment management, which were thus not repeated in this study. As proposed previously (responses to your Comment #1 and #2.3), we believe that a through revision of the Introduction will help clarifying the study focus better and thus addressing this comment too.*

28. Lines 194-196: What is the rationale behind the half-normal prior? What is the advantage compared to an exponential? The half-normal suggests that relatively small standard deviations are equally likely, while the exponential prioritises as small std. deviation as possible. Please justify your choice.

*When no a-priori knowledge on the distribution of a parameter is available, the prior distribution should be as minimally informative. Gelman (2006) demonstrated that a Gamma prior on precision*

*among exchangeable units (which we consider as the equivalent of using an exponential prior in this context) is actually highly informative and can skew results. His recommendation was the half-normal uninformative prior distribution for the standard deviation term in a linear Bayesian hierarchical model. We will add these justifications during revision.*

- *Gelman, A.: Prior distributions for variance parameters in hierarchical models (comment on article by Browne and Draper), Bayesian Anal., 1, 515-534, 10.1214/06-BA117A, 2006.*

29. Lines 212-214: This is a rather extreme test, why do you expect the model to describe the below-LOR data, after excluding all of them from the calibration dataset. Would a good fit mean that below-LOR data follow the same rules as above-LOD data do?

*We agree that this is an extreme test, but it provides a useful perspective in model performance assessment. Due to the exclusion of below-LOR data for our model calibration, readers may question how much the model performance would be affected by including the below-LOR data.*

*We agree with your interpretation that if inclusion of the below-LOR data leads to a good fit, then the models calibrated to above-LOR data is transferable to below-LOR data too (i.e. they follow the same rules). We will add this interpretation here to better highlight the utility of this specific performance evaluation.*

30. Line 217: The verb "suggested" sounds weird to me here.

*Presumably you are questioning the validity of using 'suggest' together with a quantitative measure. To address this, we will replace 'suggested' with 'quantified'.*

31. Lines 238-240: FRP is a subset of TP. TP has complicated relations to TSS. The FRPTP relationship is governed by several (fast) biochemical processes simultaneously. Consequently, it is no surprise that FRP is hard to model without considering all these intricate interactions. By the way, a negative NSE suggests that the model entirely failed to capture any of the real dynamics (negative NSE means that a constant model at the mean would perform better).

*We agree with your opinions. However, please note that this is the Results section where we refrain from providing extensive discussions. Later in the Discussion section (Specifically 4.1), we have commented on the poor performance of FRP and have specifically discussed the model limitation for representing biochemical processes for FRP.*

32. Figures 2-3: It would be great to see some visualisation beyond 1:1 plots in transformed space (of unknown transformation parameters unless one digs them up from elsewhere).

*Please note that all transformation parameters have been presented in Tables S3 and S4 in the Supplementary Information (which have been introduced in L168, Section 2.2).*

*As in response to your Comment #2, we propose to present more results to summarize different aspects of model performance, and also to illustrate model utilities that are useful for catchment management. Specifically, we propose to include the following topics:*

1) *Modelled and observed time-series at selected catchments, to illustrate model capacity to predict trends and changes over time;*
2) *Proportions of spatial and temporal variabilities explained for each constituent, to illustrate the importance of the two variability components for each constituent, and how they can be explained by our models;*

33. Lines 268-269: This sentence is not necessary, the section title tells the same.

*Thank you, we will delete the redundant sentence.*

34. Lines 269-270, 273: Please delete the "Note that . . . in Sect. .3.1." sentence and add "We exclude the FRP model from the analysis due to its poor performance (section 3.1)." into Line 273 after "monitoring sites.".

*We will address this in during the manuscript revision.*

35. Tables 1-2: These tables are all about calibration indicators, and not the subject of the model. These could be moved to the SI. Why not showing something about the factors? The introduction promised filling some knowledge gaps yet we do not learn about anything except performance indicators (and later the influence of drought on them in table 3).

*As explained in response to your Comments #1 and #2.3, the two companion studies focused on identifying key factors that influence spatial and temporal variabilities in stream water quality, whereas this study focuses on developing models to predict spatial and temporal variabilities in stream water quality. We have had extensive discussions in the two companion studies on each key control identified and the potential implications on catchment management, which we would not repeat in this study. As proposed in these responses, we believe that a through revision of the Introduction will help clarify the focus of this study better. In addition, as responded to Comment #1, we will include more results to summarize on how the temporal variations of water quality vary spatially, as this is a new finding that has not been reported in preceding studies.*

*Also, as in response to your Comment #2, we propose to present more results to summarize different aspects of model performance, and also to illustrate model utilities that are useful for catchment management. These results would be more aligned with the key objective of this study. Specifically, we propose to include the following topics:*

*1) Modelled and observed time-series at selected catchments, to illustrate model capacity to predict trends and changes over time;*
*2) Proportions of spatial and temporal variabilities explained for each constituent, to illustrate the importance of the two variability components for each constituent, and how they can be explained by our models;*
*3) Extending the existing model cross-validation from 5 replicates to 50 replicates, to provide a more comprehensive summary on model sensitivity to calibrated dataset.*

36. Line 324: The Results section is over, yet the roles of "key controls", the proportions of "inherent randomness" both remain untold. The primary value of such a model is its information content, which is embodied in the relationships that turn inputs to outputs using the parameters. Model performance indicators are important too, but in a secondary sense: they help to assess the quality of information that can be obtained from the model. Here the reader learns about the model performance in various cases, yet the lesson can't be learnt. What governs the different water quality variables? Are there covariations between the variables? Are certain models similar to others? Are errors clustered in certain situations? Which environmental factors influence the variables, how sensitive are they to the most important one? Etc.

*As explained in responses to your Comments #1, #2.3 and #36, the two companion studies focused on identifying key factors that influence spatial and temporal variabilities in stream water quality, whereas this study focuses on developing models to predict spatial and temporal variabilities in stream water quality. In these two papers we have extensively discussed the following topics: the key controls of water quality, there individual roles, interactions and how they can inform management. Therefore, we are not repeating or adding new discussions regarding key controls of spatial and temporal variabilities in water quality. As proposed in these previous responses, we believe that a through revision of the Introduction will help in clarifying this better.*

*On the other hand, as mentioned in the response to Comment #1, while developing this spatio-temporal model in this study, we have obtained new understanding on how the temporal drivers of water quality vary spatially, which has not been explored in the two preceding studies. To address this, we propose to include more results and discussions on how temporal relationships vary spatially.*

*Regarding your comment on showing model performance in various cases, we believe this can be addressed with the few additional results that we propose to add to summarize different aspects of model performance (as in response to your Comment #2). Specifically, we propose to include the following topics:*

*1) Modelled and observed time-series at selected catchments, to illustrate model capacity to predict trends and changes over time;*
*2) Proportions of spatial and temporal variabilities explained for each constituent, to illustrate the importance of the two variability components for each constituent, and how they can be explained by our models;*
*3) Extending the existing model cross-validation from 5 replicates to 50 replicates, to provide a more comprehensive summary on model sensitivity to the calibrated dataset.*

*We believe these proposed revisions on the Introduction, Results and Discussion could improve the clarification of the study objective as well as the coherence of the entire paper.*

37. Lines 333-334: Would be more positive to start with the opportunities and afterwards with limitations.

*Thank you. We will summarize the key model capabilities, contributions and opportunities at the start of Section 4.1.*

38. Lines 334-335: Or when the variability is high and explanatory power is weak. Very low FRP values could be much better simulated given that the model knows all the influencing factors and processes.

*The LOR issue is the first factor that we identified to influence model performance in Section 4.1. We have discussed the point you raised (i.e. model limitations on representing processes for FRP) in the subsequent discussions from L360.*

39. Line 336: This can also be by chance. TKN and EC are "more conservative" than the others, and have much weaker relations to sediment.

*Good point. Since this paragraph focuses on the impact of LOR data on model performance, we will add some discussions on how model performance can be affected by the degree of non-conservativeness of different constituents later in Section 4.1 (from L360).*

40. Lines 347-359: It is true that transformation increases the distance between distinct values close to the numerical resolution of data, which violates the linearity assumption. But when

you do not transform, linearity is violated by default (as one of the aims of transformation is to reduce nonlinearity). Besides the alternative model structures mentioned, a practical solution is to perturb the data with random small values (small fraction of numerical resolution), which dissolves the discrete bands of the low values without significantly altering the data. This is basically the same as "measurement noise" beyond the resolution of the time-series.

*Thank you for the interesting idea, we will include this in the discussion.*

*We note that one practical issue with this approach would be the requirement of 'data replicates' to reach some convergence of the calibrated model – since different perturbations of the raw data would lead to different calibrated models, particularly where the 'categorical issue' occupy high proportions of the data (e.g. TSS and FRP).*

41. Lines 360-361: Yes, this was obvious from the start. That's why the "positioning" of the model study is not optimal. The applied methodology tested whether temporal / regional differences could be replicated by a simple statistical model that lacks any mechanistic background. The exposition of knowledge gaps, management-relevant factors, general predictive power for ungauged catchments create expectations that simply cannot be fulfilled by this model. A lot of mechanistic knowledge is available for these water quality variables, no single bit of this knowledge is reflected by the model structure. A more realistic context would have been to investigate the overarching patterns in this region of Victoria, emphasising that the model only considers emissions only implicitly, through landuse, which in turn assumes similar human activities in the same landuse type. The results are completely in line with previous experiences, more conservative and less sediment-related variables are easier to predict than the others. The model can be a valuable predictive tool, but only in the region of calibration and only for those water quality variables, for which have the model performed acceptably.

*Firstly, we'd like to clarify again that this study focuses on developing integrated models to predict spatial and temporal variabilities in stream water quality, which we will better emphasize in the revised manuscript (as detailed in responses to your Comments #1, #2.3 and #36). We will also improve clarification in the Introduction, that the key knowledge gap this study addressed was the lack of statistical modelling approaches that are suitable for large-scale application (as we responded to your Comment #3).*

*We also propose to present more results on the model capabilities, to strengthen the key study objectives and to better highlight the values of these models to catchment management (see our responses to your Comment #2). Potential topics include:*

1) *Modelled and observed time-series at selected catchments, to illustrate model capacity to predict trends and changes over time;*
2) *Proportions of spatial and temporal variabilities explained for each constituent, to illustrate the importance of the two variability components for each constituent, and how they can be explained by our models;*
3) *Extending the existing model cross-validation from 5 replicates to 50 replicates, to provide a more comprehensive summary on model sensitivity to the calibrated dataset.*

*We believe that the abovementioned revisions will improve the alignment of the key knowledge gaps, the study objectives and the current results presented.*

*Furthermore, although our models consist of parsimonious linear relationships between water quality variables and their predictors as opposed to physically-based models governed by more complex equations, our model structures are informed by plausible conceptual relationships between potential predictors and water quality variables. Therefore, they should not be considered as 'simple statistical models that lack any mechanistic background'. The potential predictors of the models were determined following an extensive literature review and consultation with our industrial partners who are all actively working on catchment management (as described in L129-133, the consultation has not been described in the current manuscript but will be added during revision).*

*These potential predictors then went through an exhaustive predictor selection processes to extract key predictors that explain the most variabilities in water quality, which were detailed in our two companion studies (Lintern et al., 2018b; Guo et al., 2019). In the two studies we have also reviewed extensive literature to confirm the plausibility of those results from a process perspective. Considering all these process-informed decisions in model development, together with a rather comprehensive Bayesian hierarchical structure that we applied over a large region (>200,000 km²) across multiple catchments, we identified a clear misunderstanding here to consider these models as 'simple statistical models that lack any mechanistic background'. For the same reasons, we also politely disagree the comment that 'A lot of mechanistic knowledge is available for these water quality variables, no single bit of this knowledge is reflected by the model structure'. To avoid further misunderstanding on this, we will highlight the use of process-based evidences when introducing the predictor selection process in Method.*

*This study is the first time that water quality in the study region (and even world-wide) has been modelled over such a large geographical extent with a statistical approach. We believe that even though the performances of these models are 'as expected', the modelling experiences and understanding on capability of large-scale statistical water quality models will provide very useful contribution to existing studies, as these have been predominantly focused on physical models that operate at catchment scales. We would also add these points to the Discussion to highlight the study contributions.*

42. Lines 364-369: Making the model more detailed can potentially lead to a dead end. Non-linear statistical model structures may perform a bit better, but need more data for a meaningful calibration and still often lack the mechanistic background, and are much more complicated numerically. Adding descriptions of different mechanisms to the model either moves it towards a deterministic direction, which is a wrong way for this spatial and temporal scale because data will anyway appear to be at least partly random due to the lack of information on all relevant drivers, or leads to a stochasticdynamic model, which is extremely complicated and difficult to calibrate.

*Thank you for sharing the valuable opinions. Within a statistical modelling framework, a most feasible option would be to include additional predictors that are related to the key processes that affect the non-conservative constituents (e.g. DO, channel habitat condition etc.). Alternatively, non-linear structures can also be used to characterize the processes more directly. However, as discussed in our response to Comment #3, we also need to be aware of the trade-off between the complexity of model for detailed process representing versus the spatial of scale that the model is capable to present. To address this comment, we will add some examples to illustrate potential improvement of this modelling of biogeochemical processes within a statistical modelling framework, while also noting this trade-off between model scale and complexity.*

43. Lines 372-373: If this was an issue, why don't we learn about the "real-world" (=nontransformed) model accuracy earlier? The NSE values and the figures are all in transformed space, so it is difficult to judge what these mean for the practice.

*Since the model was calibrated in a transformed scale, we believe that the transformed scale is also most relevant and informative for performance assessments as we presented. We will add these justifications to Section 4.1.*

*Our transformed models focus more on proportional errors instead of absolute errors, since the latter is less important at high concentrations in practice. We will better clarify this around Line 375-377 along with the revision proposed for the comment below.*

44. Lines 375-377: I don't understand this example. Completely usual floods often bring much more sediments in almost pristine mountain catchments. Why would such an event be an alarm for management?

*Agreed, we will amend this explanation for proportional changes.*

45. Lines 377-379: How? This should have been the main topic if the logical line of the Introduction was followed. How strong is the predictive power of the calibrated models considering practical needs? Are they suitable for real forecasting either for the far future or for shorter periods during operative management?

*As in the response to your Comment #2, we propose to present more results to summarize different aspects of model performance, and also how these model capabilities can be useful for catchment management, as well as planning and policy making. Specifically, we propose to include the following topics:*

1) *Modelled and observed time-series at selected catchments, to illustrate model capacity to predict trends and changes over time;*
2) *Proportions of spatial and temporal variabilities explained for each constituent, to illustrate the importance of the two variability components for each constituent, and how they can be explained by our models;*
3) *Extending the existing model cross-validation from 5 replicates to 50 replicates, to provide a more comprehensive summary on model sensitivity to calibrated dataset.*

*We believe that this comment can be addressed once we have illustrated the value of our models to catchment management with these additional results and relevant discussion.*

46. 384-386: The references are "too new" for this statement in its present general form. Commercial solutions for online monitoring with <10 minute resolution is available for turbidity (proxy for TSS), temperature, EC, chlorophyll, dissolved oxygen since at least 20 years. Nutrient sensors are indeed newer, yet they are often not sensitive enough to yield meaningful data in surface waters (unless they are heavily polluted).

*Good point, but we also acknowledge that these sensors are only made more accessible (i.e. cheaper with improved mass production) recently, for more wide application in practices. To address this comment, we will add more well-established literature to the list.*

47. Lines 386-388: How would you apply remote sensing in stream networks? Except for larger rivers (and of course, lakes and reservoirs), these water surfaces are difficult to analyse because the number of "clean" pixels without any terrestrial or littoral influence is very low or even zero.

*Although available information extractable from remote-sensing are limited only to larger rivers, which will certainly involve further development before operational uses. However, current remote sensing data could still provide valuable information which allows us to augment the temporal resolution of existing monthly data. For example, it is possible to make spatial inferences once a network structure is developed from the larger rivers. In addition, for small streams we might also be able to extract information from available drone-based monitoring data with much higher resolutions (e.g. cm scale pixels). To address this, we propose to add more details on potential approaches to use remote sensing data in our modelling framework in this discussion (Section 4.2)*

48. Lines 390-391: There may be better (older, original) references for this. This is known since at least 30 years.

*We will add more appropriate literature to the list.*

49. Lines 391-397: Please remove, this is too case-specific.

*Yes, these are all examples from Australia. However, we believe that this is a useful quick summary of data availability, as well as highlighting opportunities in some major regions for future water quality modelling in Australia. Due to the study region, we expect this study to attract many Australian readers who might find such information particularly useful. We believe that this comment could be addressed by broaden the examples to more international examples.*

50. Lines 398-399: This is the exact reason why models fail despite the rather solid understanding of mechanisms (and this is a data or information gap and not a knowledge gap). Relevant, representative, and accurate data on such activities is close to impossible to obtain, even for smaller regions or shorter periods. Therefore, the temporal and spatial variability of these contribute to apparent "inherent randomness" and undescribed variance (the difference of NSE from 1) and weaken the predictive power of models. At the moment the solution to this issue remains an open question even for the past/present, not to mention the potentially changing practices of the future.

*We completely agree and appreciate your concerns on the challenges with obtaining good information, while also acknowledge that much of these 'ideal' information are not currently available, especially at the modelling scale that we focused on (i.e. regional, across multiple catchments). We agree that this lack of information is not a trivial point and is thus worth highlighting here as a priority for future monitoring; and it is very likely that such lack of information can only be achieved in the future with novel data collection approaches. To address this comment, we will include more examples on land use and land management activities that are relevant to water quality, and the need of monitoring these potentially via improved data collection and surveying approaches.*

*On the other hand, as in our response to your Comment #3, we believe that modelling capacity is limited by the lack of both information and understanding. We would like to highlight that current understanding of water quality remains largely at a conceptual level when focusing at a catchment scale. In contrast, understanding is still highly limited to enable large-scale water quality modelling, regarding e.g., the relative importance of key controls for water quality, how these key controls interact and how they vary across space and time etc. While information is limited, it is still possible to advance these understandings to build better modelling capacities, as we illustrated and considered as a key novelty of this study. As also mentioned in the response, there is a trade-off between having good understanding and being representative for large regions. We will add relevant discussions on this as well.*

51. Lines 422-423: Direct livestock input may increase concentrations during drought.

*Thank you, we will add this point to the discussion with supporting references.*

52. Lines 438-443: As the results of this study showed, this would be a hard job without implementing at least a few mechanistic features in the model. However, more features would require more data, potentially beyond the scope of the presented dataset.

*The existing dataset would be useful to reveal many aspects of the proposed analyses. One way to conduct this is to assess the rainfall and streamflow time-series at individual catchments to identify specific periods of droughts (which tends to vary across catchments, see Saft et al. (2015)). We could then assess how the strengths and directions of statistical relationships between water quality and its key controls change over droughts. This analysis has not been performed as part of the study due to the different focus. However, we will briefly discuss possible approaches to address this comment.*

- *Saft, M., Western, A. W., Zhang, L., Peel, M. C., and Potter, N. J.: The influence of multiyear drought on the annual rainfall-runoff relationship: An Australian perspective, Water Resources Research, 51, 2444-2463, doi:10.1002/2014WR015348, 2015.*

53. Lines 455-457: This is a crucially important sentence. I would add explicitly that the model is not only bound to the period, but also to the region for which calibration took place.

*Agreed, we will add this in the revised manuscript.*

54. Supplementary material: Figures could be structured better graphically. When 4x4 panel units are to be seen, please structure the figure so that the units get obvious. Please indicate the contents in the subfigure title. Print Box-Cox or log-sinh transformation parameters on figures or in the caption, because without knowing the strength of transformation it is difficult to judge the quality of fit.

*Thank you for the suggestions. We will implement these in the revised manuscript.*

---

## Author Comment (AC4) · 16 Oct 2019

**Responses to Reviewer Comments on "A predictive model for spatio-temporal variability in stream water quality" (RC1)**

Anonymous Referee #1

*Our proposed manuscript revisions are underlined.*

This manuscript presents a Bayesian-based approach to analyze spatio-temporal variability in stream water quality. The approach is demonstrated with an application to a large set of monitoring data in Australia. Overall, I think the manuscript is well written and will become a worthwhile contribution to the hydrological community. The proposed method also has the potential of being applied to monitoring data elsewhere. I do have some major and specific comments for the authors, which I hope can help improve the manuscript. I recommend its publication after the following comments are addressed.

*Thank you very much for your comprehensive review and recognition of the study contribution. We provide detailed responses to your comments in the subsequent sections.*

**General comments:**

1. On model applications: I recommend the authors to add a separate sub-section to provide some guidelines to potential users of the proposed approach, including at least the computer running time of the model, the required no. of stations and required no. of water-quality samples for running the model, as well as approaches to evaluate if the model does a reasonable job.

*We agree with the reviewer and we will add these recommendations for future users of this modelling framework to Section 4.1 (Implications for statistical water quality modelling) of the revised manuscript.*

2. On calibration/validation analysis: The authors randomly selected 80% of the sites for calibration and used the remaining 20% for validation, and repeated this validation process for five times for each constituent, in order to evaluate the sensitivity of the model to the monitoring sites. Could you justify the use of five times for each constituent? If this cannot be easily justified, I recommend the authors to increase the replicates from five to a larger number (say 30 or 50). The results may be summarized as boxplots instead of Table 2, which can provide an overall evaluation of the model's ability to capture the dynamics of the different constituents.

*We agree that increasing the number of this calibration/validation runs would provide a more comprehensive understanding of model robustness to calibration datasets. We are currently running this cross-validation process with 50 replicates, and we will update the relevant results and discussions in the revised manuscript.*

3. On the below-LOR data: The authors argue that the model performance is related to the proportions of below-LOR data. The results appear to support the argument that model works better when the proportion of below-LOR data is low. Can you further prove this? The authors may quantify the proportion of below-LOR data for each monitoring site and conduct a separate analysis for sites of low proportions vs. sites of high proportions (perhaps 50% of sites for each group?) and see if the performance varies significantly between the two groups. This analysis may be implemented for each constituent.

*Thank you for the interesting idea. Since our focus is to explain performance variation across constituents, we believe that a more informative analysis to support our argument (below-LOR data impacted model performance) is as follows:*

1) *For each of the constituents which has the highest proportions of below-LOR issue (e.g. TSS, FRP and NOx), calibrate the corresponding model by leaving out 10% sites which have the highest proportions of below-LOR data.*

2) *Use a 'control' constituent which are less influenced by the below-LOR issue (e.g. EC) and re-calibrate the model with the same 10% sites removed as in 1).*

3) *Compare the change in model performance from the full model between the focused constituent and the control constituent. If the former shows clear increase in performance whereas performance for the latter remains similar to the full model, then that is supporting our argument that our model performance for constituents with higher proportions of below-LOR data will be affected.*

*We will investigate these results and expand the discussing accordingly.*

4. On monitoring data: In this pilot application of the proposed approach, water-quality variability is modeled based on monthly monitoring data. First, I think the authors have made a good point that high-temporal-resolution data can further strength the model capacity to explain temporal variability in water quality. Second, I think the approach's ability to reasonably capture that variability based on just monthly monitoring data is a big strength of the proposed approach. After all, a lot of the monitoring records at many locations are based on a monthly sampling scheme. This aspect should be more emphasized. Third, how about high-flow sampling? Many monitoring programs supplement regular sampling with targeted stormflow sampling to capture concentration variability during storm events (e.g., Chanat et al., 2016; Zhang et al., 2017). It is widely acknowledged that sediment and particulate constituents are heavily affected by storms. However, I cannot find any discussion of this aspect in the manuscript. Would you expect the models to be further improved if the monitoring data contain targeted stormflow samples? References: âAˇ c Chanat et al. ´ (2016) (URL: http://dx.doi.org/10.3133/sir20155133) âAˇ c Zhang et al. (2017) (URL: ´ https://doi.org/10.1016/j.jhydrol.2016.12.052)

*Thank you very much for sharing these great discussion points. To address this, we propose to add more discussions in Section 4.2 (Implications for water quality monitoring programs) to:*

1) *Emphasize that the strength of our model in being able to predict ST variation in monthly data across large region, as many large water quality datasets are composed of monthly samples;*

2) *Discuss a) the common limitation regarding lack of high-flow (event) sampling data, and also b) transferability of the statistical model structure to event-data (where re-calibration is required). In general, monthly monitoring data (like what we used in this study) can typically well represent ambient water quality concentrations across the flow duration curve; however, they would be limited in representing and predicting conditions during events, which becomes a greater issue in estimating total load.*

*We cannot directly compare these two sampling schemes, since event-based data are not available for this particular monitoring dataset. However, we have reasonable confidence in the model capability to capture variations in flow-weighted event mean concentrations, which we obtained from a parallel study of us in the Great Barrier Reef region in Queensland (Australia). Specifically, we applied a similar spatial-temporal statistical modelling framework to an event-based water quality sampling dataset and achieved satisfactory performance. The study is currently under review with Water Resources Research.*

5. On key controlling variables: Table S5 and Table S6 may be combined to a single table and moved to the main text. I think this information is critical and deserves to be placed in the main text.

*Agreed, we will move Tables S5 and S6 to the main text. In addition, since the second column of Table S6 (which summarizes the key factors relating to the spatial variability in temporal effects) are new findings in this study, we will provide more interpretations and discussions on these results.*

**Specific comments:**

6. The term "filterable reactive phosphorus (FRP)" may be replaced with "soluble reactive phosphorus (SRP)". I think the latter is more widely used.

*Thank you for raising this point, and we agree that SRP is more widely used than FRP in the water quality field. However, the term 'FRP' has been used by the State Government of Victoria where all our water quality data were accessed from (i.e. Victoria Water Measurement Information System, available at: http://data.water.vic.gov.au/). We would like to keep consistent terminology, and thus to keep the term FRP throughout this manuscript. To avoid confusion, we will clarify the naming convention of FRP and relate it with the more commonly used terminology in the literature (SRP), when FRP is first introduced in the manuscript (L121).*

7. L46: Add a few more references to support the argument "differ significantly".

*We will add more recent references to support this argument, e.g.,*

- *Chang, H.: Spatial analysis of water quality trends in the Han River basin, South Korea, Water Research, 42, 3285-3304, https://doi.org/10.1016/j.watres.2008.04.006, 2008.*
- *Varanka, S., Hjort, J., and Luoto, M.: Geomorphological factors predict water quality in boreal rivers, Earth Surface Processes and Landforms, 40, 1989-1999, 10.1002/esp.3601, 2015.*

*We will also replace 'significantly' with 'substantially' to avoid confusion with 'statistically significant' here (and also for other similar occurrences throughout the paper).*

8. L56: Provide some specific examples on "other catchment conditions". One could be antecedent condition, which is heavily discussed in the manuscript. In this regard, Zhang et al. (2017) (URL: https://doi.org/10.1016/j.jhydrol.2016.12.052) provides a study on how antecedent conditions affect the estimation of riverine constituent concentrations. This is also relevant to your discussion at L430.

*Thank you for the recommendations. We will elaborate more on the impact of 'other catchment conditions' with the supporting literature.*

9. L103-L107: These sentences can be removed. I think the subsection titles are already very clear.

*We'd like clarify the paper structure as much as possible for the readers' benefit with these overview sentences. To address this comment while maintain clarity, we propose to move these sentences to start of Section 2 (before Section 2.1) – which is a more suitable place to have an overview of the entire Method section.*

10. Figure 1: Use a different color or a larger font for the dots to make them more clear.

*We will revise this figure to improve visualization.*

11. L130: Add a few more references to support the argument "widely known to influence water quality condition".

*We will add more recent references to support this argument, e.g.,*

- *Giri, S., and Qiu, Z.: Understanding the relationship of land uses and water quality in Twenty First Century: A review, Journal of Environmental Management, 173, 41-48, https://doi.org/10.1016/j.jenvman.2016.02.029, 2016.*
- *Heathwaite, A. L.: Multiple stressors on water availability at global to catchment scales: understanding human impact on nutrient cycles to protect water quality and water availability in the long term, Freshwater Biology, 55, 241-257, 10.1111/j.1365-2427.2009.02368.x, 2010.*

12. L131: "literature review" is vague. Could you briefly describe how it was conducted?

*We will add more details on this process, which involved both an extensive review of published literature (focusing on key controls which affect spatial variability of water quality, see Lintern et al., 2018a) and a consultation with all project partners via a project scoping workshop.*

- *Lintern, A., Webb, J. A., Ryu, D., Liu, S., Bende-Michl, U., Waters, D., Leahy, P., Wilson, P., and Western, A. W.: Key factors influencing differences in stream water quality across space, Wiley Interdisciplinary Reviews: Water, 5, e1260, doi:10.1002/wat2.1260, 2018.*

13. L164: I do think one or two references should be provided for "Box-Cox transformation" to help readers. The meaning of the parameter lambda should be also briefly described.

*We will add more details on the transformation approach, equation and parameterization with supporting literature.*

14. L352: This ranking is roughly consistent with particular constituent vs. dissolved constituent. Any comment in this regard?

*We believe that you are referring to our finding that the ranking of the 'categorical' issue most heavily affected FRP (dissolved), followed by TSS (particulate), TP (mostly particulate), NOx (dissolved), TKN (mostly particulate) and EC (dissolved) in a decreasing order. We find it difficult to relate this ranking to the form of constituent (i.e. particulate or dissolved), as there is no distinct pattern of which form of constituent has more 'categorical' issue.*

15. L366: The authors list here some processes for N. How about processes for P?

*We will expand this discussion onto alternative phosphorus pathways (e.g. P desorption and adsorption) in catchments.*

16. L206: What is the "Rhat" value? Please clarify.

*Rhat is a summary statistic on the convergence of the Bayesian models implemented in package rstan, which indicates the differences in the estimated model parameters between and within the independent Markov chains (4 chains used in this study, as in L204). Rhat>>1 indicates that the chains have not mixed well (i.e., the between- and within-chain estimates are not consistent) and a value of below 1.1 is often recommended to check convergence (Stan Development Team, 2019). We will add these clarifications in the revised manuscript.*

- *Stan Reference Manual Version 2.20: https://mc-stan.org/docs/2_20/reference-manual-2_20.pdf, access: 28/09/2019, 2019.*

**Editorial comments:**

17. L71: Fix usage of "...not only...but also..." In addition, "limits" should be "limit".
18. L76: The model built... –> The model was built...
19. Equation 3 and Equation 4: For the betas, consider using subscript instead of dash.
20. L180: "General speaking" –> "Generally speaking"
21. L317: Fix "a results of"
22. L382: Fix "oppourtunities"
23. L417: Fix "droguht"
24. L420: Similarly to –> Similar to Comments on the SM:
25. Supplementary Materials lack of "title-page" information.
26. Table S4: Change "lambda" to its Greek form.

*We appreciate your valuable comments and we will address all these in the revised manuscript.*

---

## Author Response (AR1)

**Responses to Comments on "A predictive model for spatio-temporal variability in stream water quality" (Editor)**

Editor comments published on 22 Oct 2019

Thank you for responding to the four reviews of your manuscript. I appreciate the serious manner of how you have provided answers to comments and suggest that you revise the manuscript accordingly.

Nevertheless, a few aspects deserve more attention than what you have proposed. I list them below and recommend that you pay them due attention during the revision of the manuscript as well.

Thank you for your decision. We have thoroughly revised the manuscript as proposed in our previous responses to the four reviewers. We also carefully considered your additional comments and have revised the manuscript accordingly. We provide specific responses to each of your comment as below (with specific revisions shown in underlined text).

**Transformation of the data:**

- 1. Several reviewer comments questioned aspects of how using the transformed concentration values (R2: comment 12, R3: comment 2.1, 32, Rev. 3, comment 3.2). Your arguments to avoid comparing observations and simulations in the original space is not fully convincing: you argue that it was best to evaluate model performance in the transformed space because (e.g., response to Rev. 2, comments 2.1, 43) it was most informative and because absolute errors were less important in practice.
  - 1.1 First, you don't provide an argument WHY it should be most informative in the transformed space. Actually, inspection of Fig. S13, reveals obvious model biases (even for the site-specific average concentrations, if I interpret the figure caption correctly). A careful look at Fig. 2 and 3 show similar deficiencies (e.g. systematic underestimation of high concentrations for TSS, TP, FRP, and NOx. However, these deviations are much less conspicuous than in the transformed space. This holds especially true because some of the chosen transformations are very non-linear making it very difficult to have a sense for the actual meaning of the transformed values. Additionally, inspection of Fig. S13 for EC suggests that there might be two populations of catchments: one population is very well represented by the model (close to the 1:1 line), while the second is definitely off. This can hardly be seen with the transformed data. Do the catchment being off share some commonality?

We agree with you that the untransformed plot can better help us to understand absolute model errors so we should discuss some results and implications of this. However, we also acknowledge that since the model was developed in a transformed space, performance evaluations in the transformed space would allow us to best explore a wide range of factors that can influence model performance (e.g. the LOR issue – now referred to as the 'detection-limit issue' in the revised manuscript, the limitation in simulating non-conservative constituents, and any changes in model performance across different monitoring sites and periods used for model calibration).

To resolve this comment, we first improved the justifications in Section 2.2 (Model performance and sensitivity analyses) on why model performance assessments are presented in a transformed scale.

- L297: Since the model was calibrated in a Box-Cox transformation scale (see justification in Section 2.1.2), the Box-Cox transformation scale was used for model evaluation to enable a clear investigation on the influences of a wide range of factors that can influence model performance.'

We also moved Fig. S13 to the main text to better clarify the back-transformed model performance – which becomes Fig. 5 and placed after the transformed model performance is shown in Fig. 4. Along with the figure we have added corresponding explanations on how the model performance is limited by back-transformation, as:

Figure 1. Back-transformation of the model simulations to the measurement scale emphasizes lack of fit for the highest concentrations, illustrated by simulated against observed site-level mean concentrations of each constituent in a back-transformed scale. The 95% lower and upper bounds of all posterior simulations shown in vertical grey lines. The NSE for each constituent is also shown and red dash lines show the 1:1 lines.

- L421: 'At the back-transformed scale, the model shows greater biases for sites with higher concentrations (approximately the highest 10% sites for each constituent) (Fig. 5). This is not surprising as the model was fitted to a Box-Cox transformed space that reduces focus on high values and increases the focused on low values. This compromised its ability to represent sites with unusually high concentrations. The implications of the model having higher predictive capacity in the transformed scale is further discussed in Section. 4.1.'
- 1.2 Second, the relevance of absolute errors is probably very context-specific. In some situations, practitioners do care about high concentrations and model uncertainty was important to them.

We agree with you that absolute values and high concentrations can be important to practitioners in many cases. We have added discussions the following discussions in Section 4.1 to clarify this:

- L577: 'As previously noted, our model was developed in a Box-Cox transformed scale to ensure the validity of the statistical assumptions (see details on data transformation in Sect. 2.1.2), which shows limited performance for high constituent concentrations when simulations are back-transformed to the measurement scale (Figs. 4 and 5). However, our model approximately represents proportional changes in water quality, which can thus help managers to understand proportional changes to inform practical catchment management. Footnote: All Box-Cox transformation parameters for water quality constituents are approximately 0 (Table S4), which means that the transformations are similar to a log transformation.'

Regarding your concern on uncertainty, we have added the 95% uncertainty bounds to the back- transformed model performance shown in Figure 5 (previously as Figure S13), as well as the all relevant plots showing model performance in the transformed space (Figures 4, 6, 7 and 8). Note that for clear visualization, we did not add the uncertainty bands in Figure 2 which compares all model simulations with corresponding observations. The model uncertainty is generally very small, and no clear pattern of heteroscedasticity is observed.

1.3 It is important to note that a systematic model deficiency (e.g., under or overestimation in a certain concentration range) is not alleviated by transforming the data. However, it allows for better fulfilling distributional assumption for making statistical inference. Therefore, the transformations make sense. However, to proper and transparently present the model performance and the effects of transformations, more information needs to be provided (as also suggested by the reviewers):

**Thank you, we respond to each of your specific suggestion as below.**

- 1.3.1 Provide information on how you have determined the optimal log-sinh and Box-Cox parameters (L. 161, 164). What was the optimality criteria and how did you assess optimality (manual calibration, visual inspection of quantile plots etc.)?
   We have added the equation of each transformation (Eqs. 7 and 8), with further details on approaches to determine the transformation parameters in Section 2.1.2:
- L216: 'The GA package in R (Luca Scrucca, 2019) was used to identify the log-sinh transformation parameters (a and b) for each spatial explanatory variable that minimized the data skewness (i.e. symmetry is maximized) across all 102 catchments.'
- L223: 'For each variable, the optimal Box-Cox transformation parameter  $\lambda$  was identified using the car R package and a maximum likelihood-like approach. We first identified the optimal Box-Cox parameter  $\lambda$  using the data at each site (i.e. 21-year time-series). The averaged  $\lambda$  across all sites was then used to transform the data across all catchments together. This transformation approach ensured that all sites used a consistent transformation parameter.'

In addition, we also summarized the assessment of the quality of transformation, as:

- L227: 'All transformation parameters used are summarized in Tables S3 and S4 in the Supplementary Material. The transformation process has greatly improved the data symmetry and thus suitability for use in a linear model (the quality of the transformations was assessed via visual inspection in Lintern et al., 2018b; Guo et al., 2019; and summarized in Figures S2, S4 and S6 in the Supplementary Material).'
- 1.3.2 Provide information on the distribution per site and constituent in Tab. S4. You may also consider to plot the respective distributions in the SI.

We have specified the between-site variation by the standard deviation of the lambda in Table S4.

We have added Figures S1-S6 in the Supplementary Materials for the distributions of (a) the raw data and (b) the transformed data for each of the six water quality constituents, all 50 potential spatial predictors and all 19 potential temporal predictors.

1.3.3 Complement Fig. 2 – 5 with the regression lines between observations and simulations and provide the slope estimates (including uncertainty).

Thank you for the suggestions. Firstly, we have added the 95% uncertainty bounds to the back-transformed model performance shown in Figure 5 (previously Figure S13), as well as the all relevant plots showing model performance in a transformed space (Figures 4, 6, 7 and 8, although note that Figure 2 was not imposed by uncertainty bounds for clear visualization). The model uncertainty is generally very small with no visible patterns of heteroscedasticity.

However, we are unsure about the value added by having regression lines between observations and simulations in these plots, because:

- 1) These regression lines can potentially affect the visualization of the 1:1 lines which are currently shown which we believe are sufficient, more relevant and more direct visual aids for assessing over-/under-estimation of the model we developed.
- 2) These regression lines are also not directly related to our models. Different to what these regression lines show (i.e. relationship between observed and simulated concentrations), our models described relationships between WQ constituent concentrations with catchment landscape characteristics and temporal hydro-climatic and vegetation conditions. Adding regression lines between simulations and observations may thus confuse readers when interpreting the model evaluations.

**1.3.4 Clarify whether Fig. 3 and Fig. S13 correspond to the same data.**

We have moved Fig. S13 to the main text (now as Fig. 5, after presenting the transformed results, as now in Fig. 4). These back-transformed results are introduced as:

- L422: 'At the back-transformed scale, the model shows greater biases for sites with higher concentrations (approximately the highest 10% sites for each constituent) (Fig. 5). This is not surprising as the model was fitted to a Box-Cox transformed space that reduces focus on high values and increases the focused on low values. This compromised its ability to represent sites with unusually high concentrations. The implications of the model having higher predictive capacity in the transformed scale is further discussed in Section. 4.1.'
- 1.3.5 Include one figure in the main text comparing observations and model results in backtransformed form. This could be Fig. S13 or a time series that you have mentioned several times (e.g., response 2.1 to Rev. 4).

This is addressed in our response to your last Comment #1.3.4

**Further editorial comments:**

2. L. 27: You focus here on improving the model fit for low concentrations. However, Fig. 2 and 4 suggest that the model is deficient in the low and the high concentration ranges. These systematic deviations should be addressed. If my interpretation was wrong, please provide a convincing argument why to put emphasis on the low concentrations. The argument mentioned above about the practical relevance that was less for high concentrations is not convincing. This very much depends on the actual context and some practitioners may be much more interested in high concentrations. Note that L. 154 – 155 would support this view as well.

Thank you for raising this issue. We acknowledge that the transformation issue has limited model capacity to predict absolute values for high concentrations. However, this is less of a concern is the model focuses on predicting proportional changes (e.g. as presented in a Box-Cox transformed scale) and when the interest in large-scale patterns instead of individual catchments. To address your comment and better highlighting the scope of this model, we first revised the abstract and the main text to highlight the model limitation on simulating absolute values. In the abstract we added:

- L22 (abstract): 'The model is best used to predict proportional changes in water quality in a Box-Cox transformed scale, but can have substantial bias if used to predict absolute values for high concentrations.' We also revised the discussion on the implication of the transformation impacts on model performance to better clarify the limitations and recommended model usage for management as:

- L577: 'As previously noted, our model was developed in a Box-Cox transformed scale to ensure the validity of the statistical assumptions (see details on data transformation in Sect. 2.1.2), which shows limited performance for high constituent concentrations when simulations are back-transformed to the measurement scale (Figs. 4 and 5). However, our model approximately represents proportional changes in water quality, which can thus help managers to understand proportional changes to inform practical catchment management.'

We have also revised the justification to remove the below LOR (now referred to as the detection limit (DL) data) in the Method section. Specifically, we removed an inaccurate statement that the below-LOR data were removed from analyses because our model focused on the high concentrations:

- L207: '...This was because the uncertainty in values below the DL would be amplified after transformation, which would influence the subsequent model fitting. Furthermore, those undetectable low concentrations were of less interest for management purposes. Water quality records corresponding to days with zero flows were also excluded from further analyses.'
- 3. The data presented in the main text (e.g., Fig. 3 − 5) refer to site-specific mean concentrations across space. Of course, Fig. 4 and 5 represent such mean concentrations for different periods. But there is no information on how well temporal dynamics are captured at shorter time scales. Strengthening this temporal aspect as you mention several times is important.

We have added evaluations of the model capacity to represent temporal variability by adding the following results and interpretations:

- Fig. 3, which shows the proportions of spatial and temporal variability within total observed variability, as well as the model performance in explaining each component of variability. These results indicate that the model performs much better in capturing spatial variability compared to the temporal variability.

Figure 3. Observed spatial and temporal variabilities as proportions of the total variability (total width of each bar, 100%). The dashed line differentiates temporal variability (left side) with spatial variability (right side), and the darker colours highlight the proportions of spatial and temporal variabilities that are explainable by the model. All values were estimated in Box-Cox transformed space.

- Fig. 6, which shows the simulated and observed temporal variability for each constituent, at the catchment where the model performs the best. These results further illustrated that the model largely underestimated temporal variability across all constituents, but is generally capable to represent long-term trend (except for FRP).

---

## Editor Decision (ED1)

Editor Decision

HESS-2019-342 **A predictive model for spatio-temporal variability in stream water quality**

Christian Stamm, 22.10.2019

Dear Dr. Guo,

Thank you for responding to the four reviews of your manuscript. I appreciate the serious manner of how you have provided answers to comments and suggest that you revise the manuscript accordingly.

Nevertheless, a few aspect deserve more attention than what you have proposed. I list them below and recommend that you pay them due attention during the revision of the manuscript as well.

**Transformation of the data:**

Several reviewer comments questioned aspects of how using the transformed concentration values (R2: comment 12, R3: comment 2.1, 32, Rev. 3, comment 3.2). Your arguments to avoid comparing observations and simulations in the original space is not fully convincing: you argue that it was best to evaluate model performance in the transformed space because (e.g., response to Rev. 2, comments 2.1, 43) it was most informative and because absolute errors were less important in practice.

First, you don't provide an argument WHY it should be most informative in the transformed space. Actually, inspection of Fig. S13, reveals obvious model biases (even for the site-specific average concentrations, if I interpret the figure caption correctly). A careful look at Fig. 2 and 3 show similar deficiencies (e.g. systematic underestimation of high concentrations for TSS, TP, FRP, and NOx. However, these deviations are much less conspicuous than in the transformed space. This holds especially true because some of the chosen transformations are very non-linear making it very difficult to have a sense for the actual meaning of the transformed values. Additionally, inspection of Fig. S13 for EC suggests that there might be two populations of catchments: one population is very well represented by the model (close to the 1:1 line), while the second is definitely off. This can hardly be seen with the transformed data. Do the catchment being off share some commonality?

Second, the relevance of absolute errors is probably very context-specific. In some situations, practitioners do care about high concentrations and model uncertainty was important to them.

It is important to note that a systematic model deficiency (e.g., under or overestimation in a certain concentration range) is not alleviated by transforming the data. However, it allows for better fulfilling distributional assumption for making statistical inference. Therefore, the transformations make

sense. However, to proper and transparently present the model performance and the effects of transformations, more information needs to be provided (as also suggested by the reviewers):

- Provide information on how you have determined the optimal log-sinh and Box-Cox parameters (L. 161, 164). What was the optimality criteria and how did you assess optimality (manual calibration, visual inspection of quantile plots etc.)?
- Provide information on the $\lambda$ distribution per site and constituent in Tab. S4. You may also consider to plot the respective distributions in the SI.
- Complement Fig. 2 – 5 with the regression lines between observations and simulations and provide the slope estimates (including uncertainty).
- Clarify whether Fig. 3 and Fig. S13 correspond to the same data.
- Include one figure in the main text comparing observations and model results in back-transformed form. This could be Fig. S13 or a time series that you have mentioned several times (e.g., response 2.1 to Rev. 4).

**Further editor comments:**

- L. 27: You focus here on improving the model fit for low concentrations. However, Fig. 2 and 4 suggest that the model is deficient in the low and the high concentration ranges. These systematic deviations should be addressed. If my interpretation was wrong, please provide a convincing argument why to put emphasis on the low concentrations. The argument mentioned above about the practical relevance that was less for high concentrations is not convincing. This very much depends on the actual context and some practitioners may be much more interested in high concentrations. Note that L. 154 – 155 would support this view as well.
- The data presented in the main text (e.g., Fig. 3 – 5) refer to site-specific mean concentrations across space. Of course, Fig. 4 and 5 represent such mean concentrations for different periods. But there is no information on how well temporal dynamics are captured at shorter time scales. Strengthening this temporal aspect as you mention several times is important.
- In this context, I am not fully convinced of your argument not to discuss in some more details how the model simulates the drought effects (see Fig. R3). If you consider the results solid in Fig. 4 and 5 enough to be presented in the manuscript you have also to demonstrate what makes the difference in the parameters for different periods. This is simply reporting your findings. It is subsequently fair enough to critically mention that an over-interpretation isn't warranted because of model deficiencies.

Sincerely

Christian Stamm

Editor HESS

---

## Author Response (AR2)

*(Please see our responses in blue)*

Comments to the Author:

Editor Decision

Dear Dr. Guo,

Thank you for revising the manuscript and providing extensive responses and explanations of the changes. You have addressed the basic concerns. There remain a few technical issues that should be addressed:

*Thank you so much for your consideration, our responses to your questions and corresponding revisions are specified below.*

- Fig. S1: The bins for the x-axis are selected such that the figures are not informative (e.g., all TSS observations fall into the same category). Please adapt the scale and bins accordingly.

*We have revised Fig. S1 to better visualize the highly skewed data. Specifically, for each constituent the plot only show up to the 99th percentile of all records, and the maximum value is highlighted in the corresponding panel title (see below).*

[Figure]

**Figure S1. Distribution of the raw water quality data across all catchments. Each panel shows one constituent with only the above-DL data. To help visualizing the highly skewed data, the top percentile of data for each constituent were not plotted, while the maximum value was shown in the corresponding panel title.**

- Tab. 5 (and the respective text): For TSS, FRP, NOx the full model performs worse than the minimum of all 50 partial models. Does this not indicate that there is an incompatibility of the data in space and time that cannot fully resolved by the model? Or what is your interpretation? Should this not be briefly mentioned in the text?

*Thank you so much for raising this issue. After careful checking of the results, we found that in the previous version of Table 5, the full-model performances were based on only the above-DL data, whereas the performances of the 50 partial models (cross-validation) were based on all (calibration or validation) data including both the above- and below-DL values. So, these two model evaluations were not in comparable scales in the previous manuscript.*

*During correction we have updated the 'full model performance' in Table 5 with that for all data including the below-DL records (see below), so that the full models NSEs and now more comparable with the cross-validation NSEs.*

**Table 5. Comparison of model performances (as NSE) of the full model (Column 2) and the 50 partial models (Columns 3 to 5) with each calibrated to 80% randomly selected monitoring sites. Columns 3 to 5 summarize the mean, minimum and maximum NSE values across the 50 runs, where for each constituent, the top row showing calibration performance and the bottom row showing the validation performance (i.e. at the 20% sites that were not used for calibration).**

| Constituent | Full model | 50 CV mean | 50 CV min | 50 CV max |
| --- | --- | --- | --- | --- |
| **TSS** | 0.397 | 0.413 | 0.376 | 0.439 |
| | | 0.382 | 0.292 | 0.513 |
| **TP** | 0.445 | 0.461 | 0.427 | 0.501 |
| | | 0.411 | 0.151 | 0.575 |
| **FRP** | 0.199 | 0.168 | 0.067 | 0.232 |
| | | 0.129 | -0.078 | 0.272 |
| **TKN** | 0.630 | 0.654 | 0.622 | 0.670 |
| | | 0.622 | 0.468 | 0.691 |
| **NOx** | 0.382 | 0.453 | 0.414 | 0.489 |
| | | 0.397 | 0.258 | 0.563 |
| **EC** | 0.886 | 0.893 | 0.882 | 0.903 |
| | | 0.875 | 0.809 | 0.924 |

*After this revision, the only result that still needs further attention is for NOx, where the cross-validation performances are still clearly better than that of the full-model. We think this might be related to differences in the proportions of below-DL data used to evaluate the full model and the partial models. We have presented detailed explanation in the main text as:*

- *L471: "Note the slightly higher calibration performance for the partial models of NOx compared to the full model. This seems to be related to the generally lower percentages of below-DL data in the 50 randomly-chosen partial calibration datasets (14.1%-17.9%) compared to the full dataset (17.3%) – we further discuss the impacts of below-DL data on model performance in Section 4.1."*

*In consistent with Table 5, the 'full model performance' in Table 6 (which compares the full model performance with the calibration/validation performance of three sub-periods) was also updated with inclusion of the below-DL values. Further, in consistent with these, we have also updated the corresponding model performances summary in the Abstract with inclusion of the below-DL values:*

- *L19: "Apart from FRP, which is hardly explainable (19.9%), the model explains 38.2% (NOx) to 88.6% (EC) of total spatio-temporal variability in water quality."*

- Table 6: The caption is wrong (Table 3).

*This has been revised.*

Sincerely

Christian Stamm

Editor HESS

*Again we thank you for your time and effort in handling this manuscript. We believe that the manuscript has been greatly benefited from the review process.*

[revised manuscript text omitted]

**nineteen potential temporal predictors. Values in bracket show the standard deviation of individual site-**
**level $\lambda$.**

| Water Quality Constituent | $\lambda$ |
|---|---|
| TSS | -0.249 (0.287) |
| TP | -0.058 (0.181) |
| FRP | -0.836 (1.056) |
| TKN | 0.141 (0.342) |
| NO$_x$ | 0.107 (0.305) |
| EC | -0.024 (0.921) |
| **Temporal predictors** | **$\lambda$** |
| Rainfall (mm) | 0.106 (0.041) |
| Rainfall on previous day (mm) | 0.108 (0.028) |
| Averaged rainfall over previous 3 days (mm) | 0.157 (0.022) |
| Averaged rainfall over previous 7 days (mm) | 0.220 (0.025) |
| Averaged rainfall over previous 14 days (mm) | 0.192 (0.046) |
| Averaged rainfall over previous 30 days (mm) | 0.116 (0.075) |
| Streamflow (mm d$^{-1}$) | -0.015 (0.225) |
| Streamflow on previous day (mm d$^{-1}$) | -0.027 (0.207) |
| Averaged Streamflow over previous 3 days (mm d$^{-1}$) | -0.032 (0.207) |
| Averaged Streamflow over previous 7 days (mm d$^{-1}$) | -0.030 (0.2) |
| Averaged Streamflow over previous 14 days (mm d$^{-1}$) | -0.021 (0.198) |
| Averaged Streamflow over previous 30 days (mm d$^{-1}$) | -0.004 (0.195) |
| Dry spell length in the past 14 days (days) | 0.257 (0.089) |
| NDVI for the month | 3.715 (1.998) |
| Water temperature (°C) | 0.357 (0.269) |
| Air temperature (°C) | 0.231 (0.244) |
| Evaporation (mm) | 0.019 (0.13) |
| Root zone soil moisture (%) | 0.913 (0.648) |
| Deep soil moisture (%) | 0.357 (0.269) |

**Table S5. The key temporal predictor for each water quality constituent, and the two key factors that are**
**mostly closely related to the spatial variation of each temporal predictor (see Section 2.3 in the main text**
**for detailed selection process). The corresponding Spearman's correlation coefficients (R) are also shown**
**in the last column.**

| Constituent | Key factors that affect temporal variability | Key factors that affect spatial variability in temporal effects | Spearman's R |
|---|---|---|---|
| TSS | Same-day streamflow | Annual rainfall | 0.722 |
| | | Hottest month maximum temperature | -0.575 |
| | 7-day antecedent streamflow | Annual runoff | -0.536 |

| | | Mean elevation | -0.465 |
|---|---|---|---|
| | Water temperature | Daily flow standard deviation | 0.204 |
| | | Total catchment length | 0.177 |
| | Soil moisture root | Percentage area with saline aquifers | 0.507 |
| | | Hottest month maximum temperature | 0.495 |
| | Soil moisture deep | Maximum distance upstream to dam wall or reservoir | -0.275 |
| | | Percentage area covered by grassland | -0.24 |
| TP | Same-day streamflow | Annual rainfall | 0.695 |
| | | Hottest month maximum temperature | -0.556 |
| | 30-day antecedent streamflow | Erosivity | -0.675 |
| | | Percentage cropping area | 0.626 |
| | Water temperature | Percentage agricultural area | 0.382 |
| | | Percentage area used for roads | 0.274 |
| | Soil moisture root | Percentage pasture area | 0.564 |
| | | Hottest month maximum temperature | 0.557 |
| | Soil moisture deep | Percentage area underlain by mixed igneous bedrock | -0.23 |
| | | Maximum distance upstream to dam wall or reservoir | -0.21 |
| FRP | Same-day streamflow | Percentage agriculture area | 0.392 |
| | | Percentage area underlain by mixed igneous bedrock | 0.314 |
| | Water temperature | Total catchment length | -0.28 |
| | | Coldest quarter mean temperature | 0.232 |
| | Soil moisture deep | Percentage area used for roads | -0.21 |
| | | Percentage aea covered by woodland | 0.204 |
| TKN | Same-day streamflow | Annual rainfall | 0.713 |
| | | Hottest month maximum temperature | -0.618 |
| | 30-day antecedent streamflow | Erosivity | -0.823 |
| | | Percentage cropping area | 0.694 |
| | NDVI | Mean_7daylowflow | 0.42 |
| | | Maximum distance upstream to dam wall or reservoir | -0.366 |
| | Water temperature | Coldest quarter rainfall | -0.386 |
| | | Maximum distance upstream to dam wall or reservoir | 0.374 |
| | Soil moisture root | Warmest quarter mean temperature | 0.6 |
| | | Percentage pasture area | 0.588 |
| | Soil moisture deep | Hottest month maximum temperature | -0.274 |
| | | Warmest quarter mean temperature | -0.269 |
| NOx | Same-day streamflow | Total storage capacity of dams in catchment | -0.493 |
| | | Mean soil TN content | 0.458 |
| | 30-day antecedent streamflow | Coldest quarter rainfall | -0.413 |
| | | Hottest month maximum temperature | 0.396 |
| | NDVI | Percentage area covered by woodland | -0.442 |

| | | Maximum elevation | -0.428 |
|---|---|---|---|
| | Water temperature | Percentage area underlain by mixed igneous bedrock | 0.266 |
| | | Percentage urbanized area | -0.2 |
| | Soil moisture root | Annual temperature | 0.44 |
| | | Warmest quarter average temperature | 0.338 |
| | Soil moisture deep | Percentage horticulture area | 0.341 |
| | | Wettest quarter rainfall | -0.334 |
| EC | Same-day streamflow | Percentage area covered by grassland | -0.347 |
| | | Percentage area covered by woodland | -0.317 |
| | 14-day antecedent streamflow | Percentage area covered by forest | 0.324 |
| | | PerForest_Ext | 0.276 |
| | Water temperature | Coldest month minimum temperature | -0.328 |
| | | Mean catchment slope | 0.28 |
| | Soil moisture root | Mean 7-day low flow | 0.33 |
| | | Average soil TN content | 0.303 |
| | Soil moisture deep | Maximum elevation | 0.366 |
| | | Percentage area covered by woodland | 0.312 |

[Figure]

**Figure S1. Distribution of the raw water quality data across all catchments. Each panel shows one constituent with only the above-DL data. To help visualizing the highly skewed data, the top percentile of data for each constituent were not plotted, while the maximum value was shown in text.**

[Figure]

Figure S2. Distribution of the transformed water quality data across all catchments. Each panel shows one constituent with only the above-DL data.

[Figure]

**Figure S3. Distribution of the raw data for catchment characteristics included as potential spatial predictors in the model.**

[Figure]

**Figure S4. Distribution of the transformed data for catchment characteristics included as potential spatial predictors in the model.**

[Figure]

**Figure S5. Distribution of the raw data for hydro-climatic and vegetation variables included as potential temporal predictors in the model.**

[Figure]

**Figure S6. Distribution of the transformed data for transformed (Box-Cox) hydro-climatic and vegetation variables included as potential temporal predictors in the model.**

[Figure]

Figure S4. The two key factors that are mostly closely related to the spatial variation of each temporal predictor of each water quality constituents, as highlighted in the coloured cells (see Section 2.3 in the main text for detailed selection of the two key factors). Colours indicate the corresponding Spearman's correlation coefficients (R) from -1 (red) to 1 (blue).

[Figure]

**Figure S5.** Effects of streamflow across catchments against the two most important catchment landscape characteristics, for each constituent (see Section 2.3 in the main text for detailed selection of the two key factors). Red dash lines indicate the zero levels, and thus differentiate positive and negative streamflow effects

[Figure]

**Figure S6. Annual average residuals of the models for TSS, TP and FRP, as % of long-term average. All values are presented in a Box-Cox transformed scale.**

[Figure]

**Figure S7. Annual average residuals of the models for TKN, NOx and EC, as % of long-term average. All values are presented in a Box-Cox transformed scale.**

[Figure]

**Figure S8. Comparison of the TSS model performance, as the simulated against observed site-level mean concentrations across three different calibration/validation periods for calibrations on the pre-drought (1994-1996), drought (1997-2009) and the post-drought (2010-2014) periods, respectively, see Section 2.4 for details of the calibration and validation approach.**

[Figure]

Figure S9. Comparison of the TP model performance, as the simulated against observed site-level mean concentrations across three different calibration/validation periods for calibrations on the pre-drought (1994-1996), drought (1997-2009) and the post-drought (2010-2014) periods, respectively, see Section 2.4 for details of the calibration and validation approach.

[Figure]

**Figure S10. Comparison of the FRP model performance, as the simulated against observed site-level mean concentrations across three different calibration/validation periods for calibrations on the pre-drought (1994-1996), drought (1997-2009) and the post-drought (2010-2014) periods, respectively, see Section 2.4 for details of the calibration and validation approach. Note that the unstable performance can be resulted by the poor performance for the full model, see Section 3.1.**

[Figure]

**Figure S11. Comparison of the TKN model performance, as the simulated against observed site-level**
**mean concentrations across three different calibration/validation periods for calibrations on the pre-**
**drought (1994-1996), drought (1997-2009) and the post-drought (2010-2014) periods, respectively, see**
**Section 2.4 for details of the calibration and validation approach.**

[Figure]

Figure S12. Comparison of the NO$_x$ model performance, as the simulated against observed site-level mean concentrations across three different calibration/validation periods for calibrations on the pre-drought (1994-1996), drought (1997-2009) and the post-drought (2010-2014) periods, respectively, see Section 2.4 for details of the calibration and validation approach.

[Figure]

**Figure S13. Comparison of the EC model performance, as the simulated against observed site-level mean concentrations across three different calibration/validation periods for calibrations on the pre-drought (1994-1996), drought (1997-2009) and the post-drought (2010-2014) periods, respectively, see Section 2.4 for details of the calibration and validation approach.**

[Figure]

**Figure S14. Effects of the seven key predictors for the spatial variability in TSS across 102 sites,**
**summarized by the posterior mean of the calibrated parameter values for each predictor, to the**
**pre-, during- and post-drought periods (differentiated by colour). The seven key predictors are,**
**from left: hottest month maximum temperature, percentage catchment area as grassland,**
**percentage catchment area as shrub, percentage catchment area as cropping land, maximum**
**catchment elevation, percentage catchment area made up of valley bottoms, and average soil**
**clay content.**